# Interferon-induced PARP14-mediated ADP-ribosylation in p62 bodies requires the ubiquitin-proteasome system

Rameez Raja [ID][1], Banhi Biswas[1], Rachy Abraham [ID][1], Yiran Wang[1], Che-Yuan Chang[1], Ivo A Hendriks [ID][2], Sara C Buch-Larsen[2], Hongrui Liu [ID][1,3], Xingyi Yang[1], Chenyao Wang[4], Hien Vu[1], Anne Hamacher-Brady [ID][1], Danfeng Cai[1] & Anthony K L Leung [ID][1,5,6,7 ✉]

## Abstract

Biomolecular condensates are cellular compartments without enveloping membranes, enabling them to dynamically adjust their composition in response to environmental changes through post-translational modifications. Recent work has revealed that interferon-induced ADP-ribosylation (ADPr), which can be reversed by a SARS-CoV-2-encoded hydrolase, is enriched within a condensate. However, the identity of the condensate and the responsible host ADP-ribosyltransferase remain elusive. Here, we demonstrate that interferon induces ADPr through transcriptional activation of PARP14, requiring both the physical presence and catalytic activity of PARP14 for condensate formation. Interferon-induced ADPr colocalizes with PARP14 and its associated E3 ligase, DTX3L. These PARP14/ADPr condensates contain key components of p62 bodies—including the selective autophagy receptor p62, its binding partner NBR1 and the associated protein TAX1BP1, along with K48-linked and K63-linked polyubiquitin chains—but lack the autophagosome marker LC3B. Knockdown of p62 disrupts the formation of these ADPr condensates. Importantly, these structures are unaffected by autophagy inhibition, but depend on ubiquitination and proteasome activity. Taken together, these findings demonstrate that interferon triggers PARP14-mediated ADP-ribosylation in p62 bodies, which requires an active ubiquitin-proteasome system.

**Keywords** Interferon; Condensates; ADP-Ribosylation; p62; Ubiquitin-Proteasome System
**Subject Category** Post-translational Modifications & Proteolysis

## Introduction

Biomolecular condensates are broadly defined as cellular compartments that are not enveloped by membranes (Banani et al, 2017; Alberti and Dormann, 2019; Shin and Brangwynne, 2017; Spannl et al, 2019). Some, such as nucleoli, are constitutively present, while others, including DNA repair foci and stress granules, form in response to changes in cellular conditions. As components can freely diffuse in and out of these condensates, this constant exchange allows the condensates to rapidly adjust their composition over time, adapting to a changing environment. However, the mechanisms governing condensate composition remain a significant gap in the field.

A condensate formation can be regulated by post-translational modifications, such as ADP-ribosylation (ADPr) (Gupte et al, 2017; Dasovich and Leung, 2023; Lüscher et al, 2021; Suskiewicz et al, 2023)—the addition of one or more ADP-ribose units onto proteins, regulated by a family of 17 human ADP-ribosyltransferases commonly known as PARPs. Within this family, four PARPs add poly(ADP-ribose) (PAR), 11 add mono(ADP-ribose) (MAR), and two are catalytically inactive (PARP9 and PARP13) (Lüscher et al, 2021; Vyas et al, 2014). PAR is critical for the structural integrity and function of DNA repair foci, stress granules, and nucleoli in human cells, while MAR is essential for forming *Sec* bodies under amino acid starvation in *Drosophila* cells (Leung, 2020; Aguilera-Gomez et al, 2016).

Recently, a novel class of ADPr-enriched condensates (hereafter ADPr condensates) was identified in the cytoplasm of human lung A549 epithelial cells during the search for coronavirus antivirals (Russo et al, 2021). Notably, these condensates are induced by type I and II interferons (IFNα, β, and γ) (Russo et al, 2021) and can be reversed by a SARS-CoV-2 macrodomain with hydrolase activity to remove MAR (Dasovich et al, 2021; Rack et al, 2020; Alhammad et al, 2020). Within these condensates, ADPr regulation is mediated by the PARP9/DTX3L heterodimer (Russo et al, 2021). However, as PARP9 is catalytically inactive as an ADP-ribosylation writer

[1]Department of Biochemistry and Molecular Biology, Bloomberg School of Public Health, Johns Hopkins University, Baltimore, MD 21205, USA. [2]NNF Center for Protein Research, Copenhagen N DK-2200, Denmark. [3]XDBio Graduate Program, Johns Hopkins School of Medicine, Baltimore, MD 21205, USA. [4]BeiGene Institute, Shanghai R&D Center, Shanghai 200131, China. [5]Department of Molecular Biology and Genetics, Johns Hopkins University, Baltimore, MD 21205, USA. [6]McKusick-Nathans Department of Genetic Medicine, Johns Hopkins University, Baltimore, MD 21205, USA. [7]Department of Oncology, School of Medicine, Johns Hopkins University, Baltimore, MD 21205, USA. ✉E-mail: anthony.leung@jhu.edu

(Vyas et al, 2014; Zhu et al, 2022), the specific enzyme responsible for MAR addition remains unidentified. This study delineates the identity of the interferon-induced PARP(s) involved in MAR addition, determines whether these ADPr condensates are part of a known class or a new cytoplasmic structure, and investigates the requirements for ADPr condensation.

Here, we report that these interferon-induced ADPr condensates are regulated by the MAR-adding ADP-ribosyltransferase PARP14, which shares the same genomic loci as PARP9 and DTX3L—all of which are transcriptionally activated by interferons (Juszczynski et al, 2006). Intriguingly, PARP14 functions as a dual-activity enzyme, also possessing a macrodomain with hydrolase activity similar to that observed in SARS-CoV-2 (Đukić et al, 2023; Torretta et al, 2023; Delgado-Rodriguez et al, 2023). Notably, ectopic expression of a PARP14 macrodomain mutant, deficient in ADP-ribosylhydrolase activity, leads to high levels of ADP-ribosylation and forms ADPr condensates, colocalizing with the mutant (Đukić et al, 2023). However, it was unclear if this occurs with the endogenous wild-type protein. This study demonstrates that PARP14 colocalizes with interferon-induced ADPr condensates, depending on its transferase activity.

In addition to PARP14's transferase activity, we also identified that forming these ADPr condensates depends on p62, also known as sequestosome1 (SQSTM1) (Berkamp et al, 2021; Komatsu, 2022; Moscat and Diaz-Meco, 2012). p62 serves as a central hub for signaling pathways and directs ubiquitinated proteins toward degradation via autophagy. Upon interacting with ubiquitinated proteins, p62 condenses in vitro and forms p62 bodies within cells (Sun et al, 2018; Zaffagnini et al, 2018). These p62 bodies, heterogeneous in size, are present in a wide range of cell lines from various tissue origins, including both normal and cancerous types (Bjørkøy et al, 2005; Moscat and Diaz-Meco, 2012). Diverse structures containing p62 and ubiquitin have been noted in liver cancers, as well as in various neurodegenerative diseases, such as Parkinson's, Alzheimer's, and Huntington's disease (Kuusisto et al, 2001; Zatloukal et al, 2002; Nagaoka et al, 2004; Tan et al, 2008). Therefore, elucidating how alterations in the composition of p62 bodies affect their function in cell signaling or protein degradation could open novel therapeutic opportunities (Kehl et al, 2019).

In this work, we established that the ADPr-containing p62 bodies formed upon interferon treatment are distinct from the canonical ones. While these bodies contain ubiquitin (Tan et al, 2008; Zaffagnini et al, 2018; Jakobi et al, 2020), they lack the autophagy marker LC3B and are not affected by autophagy inhibition; instead, they depend on an active ubiquitin-proteasome system. These compositional changes in p62 bodies likely reflect dynamic adaptations to the immune environment, facilitated in part by PARP14-mediated ADPr.

# Results

## Interferon-induced cytoplasmic ADPr condensates depend on PARP14

To investigate the molecular mechanisms behind ADPr condensate formation, we initially examined their response to interferon. Among the three interferons tested, IFNγ was more potent than IFNα and β (Fig. EV1A), hence chosen for further study. ADPr condensates formed as early as 10 h after IFNγ exposure (Fig. 1A). To assess if ADPr condensate formation depends on continuous or transient IFNγ exposure, cells were exposed to IFNγ for various durations before switching to IFNγ-free media for the remaining time of the 24-h observation period (Fig. EV1B). Remarkably, just 1 h of IFNγ exposure was enough to induce ADPr condensate formation (Fig. EV1B), indicating that continuous IFNγ presence is not necessary after initial exposure.

The requirement for a 1-h IFNγ exposure and the ensuing 10-h delay before ADPr condensate formation indicates the possible involvement of a transcription program triggered by IFNγ. To determine whether ADPr condensate formation relies on transcription, we treated cells with the inhibitor Actinomycin D. Indeed, upon transcriptional inhibition, ADPr condensates were no longer observed (Fig. 1B).

As some PARPs are interferon-stimulated genes, we assessed the mRNA expression of all 17 human PARPs after IFNγ treatment. PARP9, PARP12, and PARP14 were significantly upregulated at 6 and 24 h post-treatment, with PARP14 showing the largest increase in mRNA expression levels (Fig. EV1C,D). A similar upregulation was observed after just 1 h IFNγ exposure (Fig. EV1E, upper panel). The significant upregulation of PARP14, a MAR-adding transferase, aligns with previous data indicating that the MAR-degrading SARS-CoV-2 Mac1 macrodomain can remove the ADPr signal within the condensates (Dasovich et al, 2021; Rack et al, 2020; Alhammad et al, 2020). Western blot analyses demonstrated a corresponding induction of PARP14 protein post-IFNγ treatment (Figs. 1C and EV1E, lower panel). Immunostaining further revealed these ADPr condensates are also enriched with PARP14 (Fig. 1D). These results collectively highlight the critical role of IFNγ-induced transcription, particularly involving MAR-adding PARP14, in the formation of ADPr condensates.

To determine if the presence of PARP14 is crucial for ADPr condensate formation, we employed genetic depletion strategies, including siRNA, shRNA, and CRISPR/Cas9, to reduce PARP14 levels (Figs. 1E,F and EV1F–K). In all cases, these IFNγ-induced ADPr condensates were no longer observed with PARP14 depletion (Figs. 1F, EV1I,K). Notably, the knockdown of PARP9 or PARP12—both of which are also upregulated by IFNγ—did not affect ADPr condensate formation (Fig. EV1L–O), indicating that PARP14 primarily drives this condensation.

Additionally, specific degradation of PARP14 proteins using the proteolysis targeting chimera (PROTAC) inhibitor RBN012811 also led to the disappearance of ADPr condensates within just 1 h (Fig. 1G,H) (Schenkel et al, 2021; Wigle et al, 2021; Wong et al, 2023). This effect was not observed with the negative control, RBN013527, an N-methylated analog (Fig. 1G,H) (Wigle et al, 2021). Taken together, the physical presence of PARP14 is essential for ADPr condensate formation.

## PARP14 catalytic activity is required for ADPr condensate formation and co-condensation with PARP14

As an orthogonal approach to determine which PARP activity is necessary for ADPr condensate formation, we screened a panel of PARP inhibitors (Figs. 2A and EV2A–F). This panel included inhibitors targeting different PAR-adding ADP-ribosyltransferases: Olaparib (PARPs 1/2), XAV939 (PARPs 1,2,5a/b), and MAR-adding ADP-ribosyltransferases: OUL-35 (PARP10), ITK6

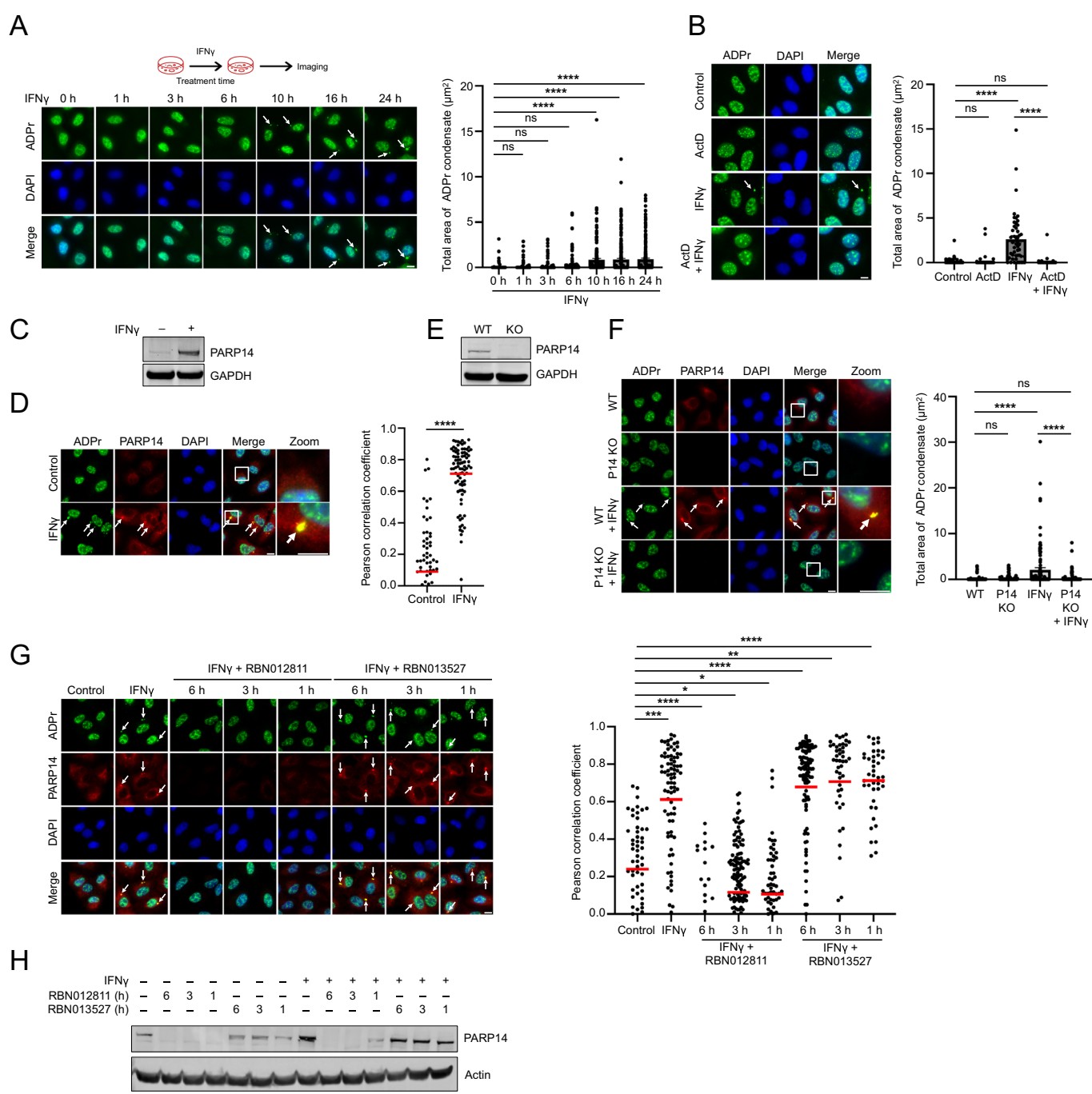

**Figure 1. Interferon-induced cytoplasmic ADPr condensates depend on PARP14.**

(A) A549 cells were treated with IFNγ (500 IU/ml) for the indicated time points, stained with pan-ADPr binding reagent MABE1016, and monitored for ADPr condensate formation, n = 314–613 cells. (B) ADPr condensate formation was analyzed in cells pretreated with Actinomycin D (ActD; 0.5 μg/ml) for 1 h, followed by 24-h treatment with or without IFNγ, n = 143–243 cells. (C) PARP14 protein levels were measured in cells treated with or without IFNγ for 24 h by western blot. (D) PARP14 and ADPr condensate colocalization was assessed in control and IFNγ-treated cells after 24 h, n = 71-86 condensates. (E) Western blot showing PARP14 protein levels in A549 wild-type (WT) and PARP14 knockout (KO) cells. (F) ADPr condensate formation was analyzed in A549 WT and PARP14 KO cells treated with or without IFNγ for 24 h, n = 213–456 cells. (G) ADPr and PARP14 colocalization was analyzed in cells treated with or without IFNγ overnight, followed by treatment with either PROTAC RBN012811 (1 μM) targeting PARP14 or its negative control analog RBN013527 (1 μM) for the indicated time points, n = 43–109 condensates. (H) Western blot analysis of PARP14 protein levels corresponding to panel (G). White arrows indicate the position of condensates, and white boxes represent the zoomed-in regions. The total area of condensates and Pearson correlation coefficient were analyzed by CellProfiler. Mean ± SEM, ns not significant, *p < 0.05, **p < 0.01, ***p < 0.001, ****p < 0.0001. (A, F, G) One-way ANOVA; (B) Two-way ANOVA; (D) Unpaired t-test. All p values are provided in Dataset EV2. Data were representative of three biological replicates. Scale bar, 10 μm. Source data are available online for this figure.

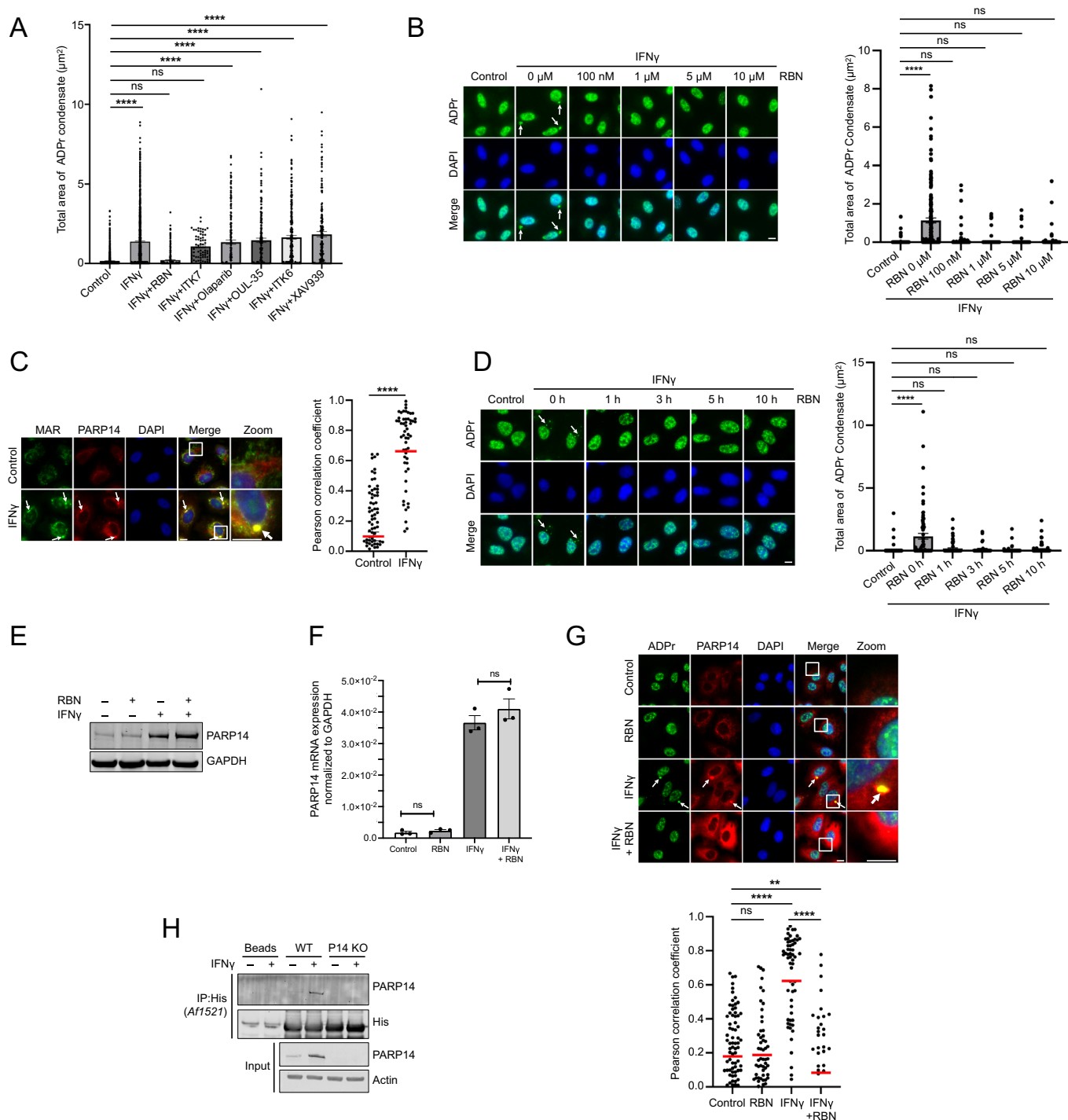

(PARP10), ITK7 (PARP11), and RBN012579 (PARP14) (Schenkel et al, 2021; Kirby et al, 2018; Venkannagari et al, 2016; Kirby and Cohen, 2019). Consistent with our investigations on transcriptional profiles and genetic depletion, inhibitors capable of inhibiting PARP14's catalytic activity—specifically, RBN012579 (hereafter RBN) and ITK7 (a potent PARP11 inhibitor with inhibitory effects on PARP14 that are weaker than RBN) (Schenkel et al, 2021; Kirby et al, 2018)—suppressed ADPr condensate formation (Figs. 2A,B and EV2A,B,G). In alignment with the requirement for MAR-adding

PARP14, these condensates were enriched with MAR, while PAR was not detected (Figs. 2C and EV2H). These findings underscore the crucial role of PARP14—both its physical presence and MARylation activity—in the formation of ADPr condensates.

Given RBN's greater specificity and lower required dose for effective inhibition than ITK7 (Figs. 2B and EV2G), we chose RBN for our in-depth PARP14 inhibition analyses. Notably, RBN significantly reduced the formation of ADPr condensates with as brief as 1-h treatment (Fig. 2D). Although RBN treatment resulted

**Figure 2. PARP14 catalytic activity is required for ADPr condensate formation and co-condensation with PARP14.**

(A) Quantification of ADPr condensates in A549 cells pretreated with different PARP inhibitors for 1 h prior to 24-h IFNγ treatment, and compared to untreated control. 10 μM RBN, 10 μM ITK7, 10 μM Olaparib, 3 μM OUL-35, 10 μM XAV939, and 10 μM ITK6, $n = 164$–3284 cells. (B) ADPr condensate formation was analyzed in cells pretreated with different doses of RBN for 1 h, followed by 24-h IFNγ treatment, and compared to untreated control, $n = 419$–690 cells. (C) MAR (HCA-355) and PARP14 colocalization was assessed after 24-h treatment with or without IFNγ, $n = 57$–91 condensates. (D) ADPr condensate formation was analyzed in cells treated with IFNγ overnight, followed by RBN treatment (10 μM) for the indicated time points, and compared with untreated control, $n = 87$–208 cells. (E) PARP14 protein levels were measured in cells pretreated with RBN (10 μM) for 1 h with or without 24-h IFNγ treatment. RBN was maintained throughout the experiment. (F) PARP14 mRNA levels were measured after 6 h of IFNγ treatment either alone or in the presence of RBN (10 μM) by RT-qPCR, and compared to untreated control. (G) Colocalization of ADPr and PARP14 was assessed in cells pretreated with either DMSO control or RBN (10 μM) for 1 h, with or without 24-h IFNγ treatment. RBN was maintained throughout the experiment, $n = 45$–88 condensates. (H) A549 WT and PARP14 KO cells were treated with or without IFNγ for 24 h and subjected to immunoprecipitation using the pan-ADPr binding reagent (MABE1016) overnight. This reagent is a His-tagged recombinant *Af1521* macrodomain that binds to ADP-ribosylated proteins. Ni-NTA resin was used to pull down His-tagged MABE1016, followed by western blot analyses. White arrows indicate the position of condensates, and white boxes represent the zoomed-in regions. The total area of condensates and Pearson correlation coefficient were analyzed by CellProfiler. Mean ± SEM, ns not significant, *$p < 0.05$, **$p < 0.01$, ***$p < 0.001$, ****$p < 0.0001$, unless otherwise stated. (A, G) Two-way ANOVA; (B, D, F) One-way ANOVA; (C) Unpaired *t*-test. All *p* values are provided in Dataset EV2. Data were representative of three biological replicates. Scale bar, 10 μm. Source data are available online for this figure.

in elevated PARP14 protein levels without altering mRNA levels (Fig. 2E,F), PARP14 was no longer observed in its condensate form (Fig. 2G), indicating that the mere physical presence of PARP14 is not sufficient for its localization along with ADPr.

Because PARP14 was pulled down by ADP-ribose binding Af1521 macrodomain following IFNγ treatment in wild-type cells, but not in PARP14 knockout cells (Fig. 2H), PARP14 might be present in an ADP-ribosylated state within these condensates. Taken together, these observations suggest that the continuous presence of both ADPr and PARP14 in condensate form depends on the catalytic activity of PARP14.

## p62 is required for condensation of ADPr and PARP14 upon IFNγ treatment

Next, we explored whether these cytoplasmic condensates, enriched with ADPr and PARP14 in an IFNγ-dependent manner, represent a known or novel cellular structure. We examined the colocalization of the IFNγ-induced ADPr signal with markers of various organelles (ER, Golgi, lysosome, mitochondria) and biomolecular condensates (stress granules, p-bodies), as well as macromolecular complexes such as immunoproteasome and proteasome (Fig. EV3A,B). Among all structures tested, the ADPr signal did not colocalize with any markers, with the sole exception being the selective autophagy marker p62 (Fig. 3A). p62 can self-assemble into condensates in vitro and is crucial for forming "p62 bodies" within cells (Sun et al, 2018; Zaffagnini et al, 2018). As expected, MAR and PARP14 were also detected in p62 bodies upon IFNγ treatment (Figs. 3B and EV3C).

Super-resolution Airyscan imaging further confirmed the colocalization of p62 bodies with ADPr (Fig. 3C,D). These condensates were slightly, but significantly, larger than those that did not contain ADPr (Fig. 3E), accounting for ~20% of p62 bodies following IFNγ treatment (Fig. S3D-E). Closer examination revealed sub-compartments within these condensates (Fig. 3F; Movie EV1), each exhibiting an elongated morphology with a less-than-spherical shape and a high solidity, indicating a compact structure with minimal indentations (Figs. 3G and EV3F,G). ADPr exhibited the largest volume, with overlapping regions involving PARP14 and p62 at different 3D spatial locations (Fig. 3F,G).

Colocalization of p62, PARP14, and ADPr was also observed in the melanoma cell line A375 following IFNγ treatment (Fig. EV3H), suggesting that this phenomenon is not restricted to lung cells.

Given that p62 bodies exist constitutively in unstressed conditions (Bjørkøy et al, 2005) (Fig. 3A), these findings suggest that PARP14 is localized to p62 bodies upon IFNγ treatment when ADP-ribosylation occurs.

Following IFNγ treatment, p62 bodies increased in the overall size distribution (Figs. 3E and EV3I). Fluorescence recovery after photobleaching (FRAP) analyses of similarly sized p62 bodies revealed that IFNγ treatment increased the mobile fraction of p62 within the bodies (Figs. 3H,I and EV3J). However, it did not alter the exchange dynamics between p62 bodies and the cytoplasm. Taken together, IFNγ treatment modulates the size, composition, and physical properties of p62 bodies.

To further investigate the role of p62 in ADPr condensates, we generated p62 knockdown clones from A549 cells using lentiviral transduction (Fig. EV3K). p62 knockdown resulted in the loss of IFNγ-induced ADPr signal and PARP14 in the form of cytoplasmic condensates (Fig. 3J,K), despite PARP14 protein levels remaining unchanged in the presence of IFNγ (Fig. EV3L). Therefore, p62 is required for the condensation of ADPr and PARP14 upon IFNγ treatment.

## PARP14-mediated MARylation facilitates p62 co-condensation

To investigate the interaction between p62 and PARP14, we immunoprecipitated p62 and probed for PARP14. p62 and PARP14 were associated with each other in untreated conditions, but more PARP14 was pulled down by p62 following treatment with IFNγ (Fig. 4A). As the protein and mRNA levels of p62 remained unchanged upon IFNγ treatment (Fig. 4A–C), the increased association is likely attributable to the upregulation of PARP14 expression (Fig. 4A). This association prompted us to inquire whether p62 could be a substrate of PARP14 for MARylation. We observed an increased MARylation of p62 upon IFNγ treatment; however, this increase was abrogated in PARP14 knockout cells (Figs. 4D and EV4A), where p62 bodies were still present (Fig. 4E). Notably, the MARylation signal on p62 immunoprecipitates persisted under denaturing conditions (Fig. EV4B). Collectively, these data suggest that p62 undergoes increased MARylation by PARP14 upon IFNγ treatment.

Importantly, upon treatment with the PARP14-specific inhibitor RBN, the association between PARP14 and p62 was not disrupted (Fig. 4A), indicating the association was not dependent on

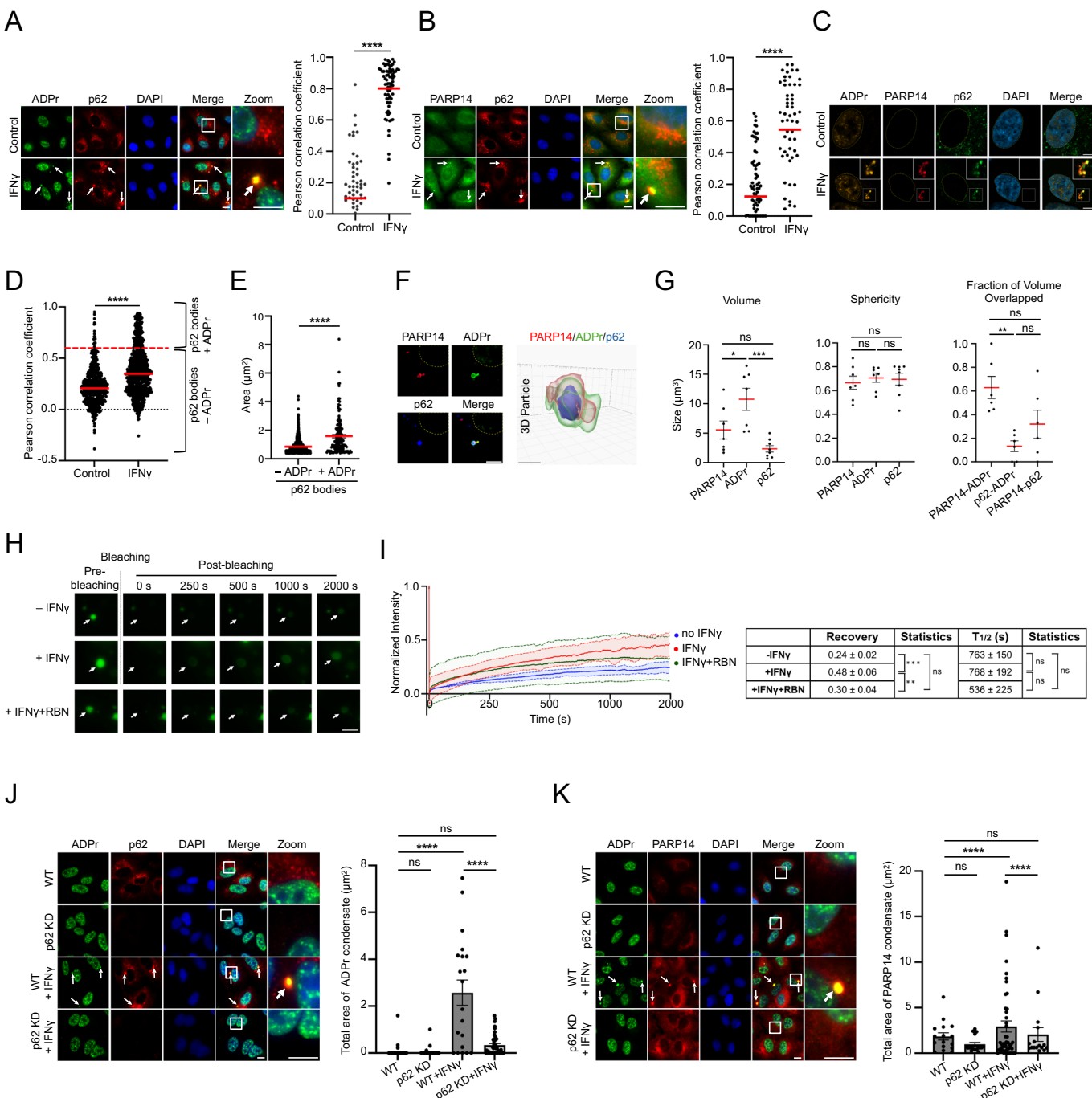

ADP-ribosylation. While the mRNA and protein levels of p62 remained unchanged with RBN treatment (Fig. 4C), the association was further increased, likely due to the increased PARP14 protein levels (Schenkel et al, 2021) (Fig. 4A). Yet, the ADPr and PARP14 signals were no longer observed in p62 bodies (Figs. 4F,G and EV4C), suggesting that the catalytic activity of PARP14, rather than its association with p62, is the prerequisite for ADPr condensation.

FRAP analyses further revealed that the mobile fraction of p62, in the presence of IFNγ and the PARP14 inhibitor RBN, was restored to levels comparable to those observed before IFNγ treatment, with no

apparent change in recovery time (Figs. 3H,I and EV3J). These findings suggest that PARP14, likely through its MAR-adding activity, alters the binding interactions within p62 bodies following IFNγ treatment.

To determine whether IFNγ is required for the co-condensation of PARP14-mediated ADPr and p62, we investigated the effects of a PARP14 mutant deficient in ADP-ribosylhydrolase activity (Fig. 4H). Previous studies demonstrated that transient transfection of this mutant into U2OS cells leads to the formation of cytoplasmic ADPr condensates colocalizing with PARP14, independent of IFNγ treatment (Đukić et al, 2023). Notably, a subset of

◄

**Figure 3.  p62 is required for condensation of ADPr and PARP14 upon IFNγ treatment.**

(A, B) Colocalization of p62 with (A) ADPr and (B) PARP14 was analyzed in cells with or without 24-h IFNγ treatment, n = 56–88 condensates. (C) Super-resolution imaging of ADPr, p62, and PARP14 colocalization in cells, with or without 24-h IFNγ treatment, was performed using a Zeiss Airyscan detector. Scale bars = 5 μm. (D) ADPr and p62 colocalization was assessed, with or without 24-h IFNγ treatment, n = 466–714 condensates. (E) The size of p62 bodies, with or without ADPr, was compared, n = 133–1878 condensates. (F) 3D representation of ADPr condensate (right) and their colocalization with PARP14 and p62, generated using the surface detection module (Imaris) from original images (left). Scale bar, 1 μm (right) and 5 μm (left), n = 7 condensates. (G) ADPr-containing p62 bodies were characterized for volume, sphericity (1.0 = perfect sphere), and a fraction of volume overlap among ADPr, PARP14, and p62; n = 6-8 condensates. (H, I) FRAP analyses of A549 cells transiently transfected with GFP-p62 with or without 24-h IFNγ treatment. 10 μM RBN was given for 6 h before FRAP analysis. Table summarizing FRAP analyses without IFNγ, with IFNγ, and with both IFNγ and RBN. n = 3 condensates for without and with IFNγ, n = 6 condensates for with both IFNγ and RBN. Scale bar, 5 μm. Refer to the Methods section for the calculation of mobile fractions and T½. (J, K) ADPr and PARP14 condensate formation were analyzed in wild-type (WT) and p62 knockdown (KD) A549 cells, with or without 24-h IFNγ treatment, n = 53–173 cells for panel (J), n = 24–39 cells for panel (K). White arrows indicate the position of condensates, and white boxes represent the zoomed-in regions. The total area of condensates and Pearson correlation coefficient were analyzed by CellProfiler. Mean ± SEM, ns not significant, *p < 0.05, **p < 0.01, ***p < 0.001, ****p < 0.0001, unless otherwise stated. (A, B, D, E) Unpaired t-test; (G, I) One-way ANOVA; (J, K) Two-way ANOVA. All p values are provided in Dataset EV2. Data were representative of three biological replicates. Scale bar, 10 μm, unless otherwise indicated. Source data are available online for this figure.

condensates—particularly the larger ones—that contain both PARP14 and ADPr showed strong colocalization with p62 (Fig. 4H). Treatment with RBN under these conditions resulted in the disappearance of ADPr/PARP14 condensates while p62 bodies remained (Fig. 4I), further indicating that ADPr enriched in p62 bodies depends on the transferase activity of PARP14. Consistently, PARP14 and ADPr condensates were not observed when expressing a PARP14 mutant deficient in transferase activity (Fig. EV4D). These findings suggest that elevated PARP14-mediated ADP-ribosylation, whether induced by IFNγ or resulting from reduced hydrolase activity, can lead to its co-condensation with p62 bodies.

## ADPr-enriched p62 bodies contain ubiquitin but lack autophagy marker LC3B

p62, a multifunctional scaffolding protein, comprises multiple domains that facilitate protein interactions critical for cell signaling and protein degradation (Fig. 5A) (Berkamp et al, 2021; Sun et al, 2018; Moscat and Diaz-Meco, 2012). Inflammatory NF-κB signaling is mediated by the N-terminal self-associating PB1 domain, a ZZ-type zinc finger motif, and a TRAF6-binding domain (TB), whereas antioxidant signaling NRF2 activation is mediated through the Keap1-interacting region (KIR). These signaling domains are linked by an intrinsically disordered region that interacts with the autophagosome via LC3 (LIR). This LC3 interaction, coupled with the ubiquitin-associated (UBA) domain at p62's C-terminus, targets ubiquitinated proteins for autophagy, positioning p62 as a selective autophagy receptor.

We initially explored the activation of NF-κB and NRF2 signaling pathways by IFNγ and whether this activation is PARP14-dependent. Activation of the IFNγ signaling pathway involves the phosphorylation of STAT1 at tyrosine 701. STAT1 phosphorylation remained robust at 6 and 24 h in A549 PARP14 knockout cells or in wild-type parental lines treated with the PARP14 inhibitor RBN (Fig. EV5A,B). This pattern was also observed in p62 knockdown cells (Figs. 5B and EV5C), suggesting that the IFNγ signaling pathway is not dependent on p62 or PARP14.

No significant NRF2 signaling activation was observed with or without IFNγ treatment: its downstream targets, NQO1 and HO1, were not induced, regardless of PARP14 levels or activity, as evidenced respectively by genetic depletion and RBN inhibition (Fig. EV5A,B). Similarly, NF-κB-induced genes, such as IL6 and OAS1, maintained consistent expression levels despite the presence

of the PARP14 inhibitor (Figs. 5C and EV5D). Taken together, these results suggest that under IFNγ stimulation, the regulation of NF-κB and NRF2 signaling activation in these cells does not depend on the physical presence and enzymatic activity of PARP14.

Next, we investigated the relationship between ADPr condensates and two major mechanisms of protein degradation mediated by p62. Normally, autophagy initiation is suppressed by the mTOR pathway (Kim and Guan, 2015). However, when inhibited with the mTOR inhibitor Torin 1, autophagy is induced, leading to increased autophagosome formation marked by LC3B on the membranes (Fig. EV5E,F), which facilitates the recruitment of p62 and ubiquitinated proteins. However, in contrast to Torin 1-induced autophagy, the ADPr-containing p62 bodies induced by IFNγ treatment notably lacked LC3B (Figs. 5D and EV5G).

Despite this difference, these p62 bodies that contain ADPr and PARP14 were enriched in ubiquitinated proteins (Figs. 5E,F and EV5H–K), including both K63 and K48 linkages (Tan et al, 2008; Zaffagnini et al, 2018; Sun et al, 2018), with the latter being known for its role in protein degradation. This colocalization with ubiquitin signals is similarly observed in canonical p62 bodies, which can be induced by puromycin (Fig. EV5I–K), an amino acid analog that terminates protein synthesis and triggers polyubiquitination of these premature polypeptides (Kehl et al, 2019; Clausen et al, 2010).

Additionally, we detected the presence of NBR1 (Fig. 5G)—a p62 homolog and binding partner—which not only promotes the formation of p62 bodies but also serves as a receptor for ubiquitinated cargo cooperatively (Turco et al, 2021; Zaffagnini et al, 2018; Rasmussen et al, 2022; Jakobi et al, 2020). Besides NBR1, we also observed colocalization with another autophagy receptor TAX1BP1 (Figs. 5H and EV5L,M), but not with the downstream autophagy initiation component FIP200 (Turco et al, 2021) (Figs. 5I and EV5N), suggesting that these condensates selectively associate with certain autophagy receptors without recruiting the autophagosome formation machinery. These data indicate that ADPr is enriched in p62 bodies that contain PARP14, NBR1, TAX1BP1, and ubiquitin but lack LC3B and FIP200.

To globally identify which proteins are increasingly associated with p62 upon IFNγ treatment, we immunoprecipitated p62 from both untreated and IFNγ-treated samples and subjected them to mass spectrometry (MS)-based proteomics analysis (Figs. 5J–L and EV5O–Q; Dataset EV1). Our analysis revealed proteins differentially associated with p62 in untreated versus

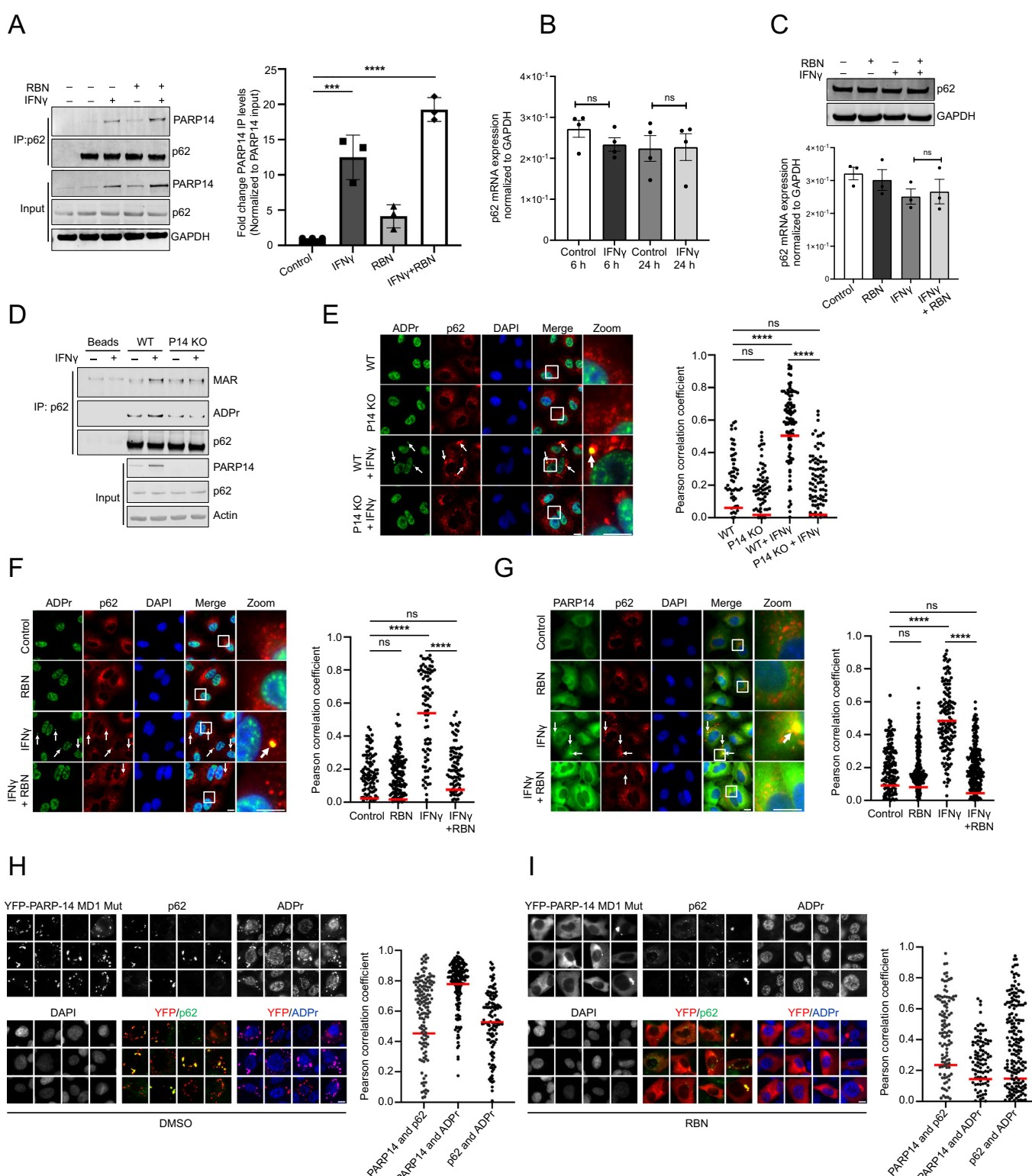

IFNγ-treated samples, distinct from those pulled down by the IgG control (Figs. 5J and EV5O). Specifically, we observed an increased association of p62 with PARP14 and two other proteins known to co-express with PARP14—PARP9 and DTX3L. In contrast, associations with other known binding partners, NBR1 and TAX1BP1, remained relatively unchanged (Fig. 5K).

## Ubiquitin-mediated proteasome activity is required for PARP14 and ADPr co-condensation with p62 bodies

The presence of ubiquitin but not LC3B suggested that autophagy might not be central to ADPr regulation in this subclass of p62 bodies. Supporting this premise, neither knockouts of key

**Figure 4.  PARP14-mediated MARylation facilitates p62 co-condensation.**

(A) A549 cells were treated with a different combination of IFNγ and RBN (10 μM) for 24 h, and immunoprecipitates of p62 were probed for p62 and PARP14 and quantified on the right panel. (B) p62 mRNA levels were measured by RT-qPCR after 6-h and 24-h treatment with or without IFNγ, $n = 4$ biological replicates. (C) p62 protein levels were measured in A549 cells treated with different combinations of RBN (10 μM) and IFNγ, same as in Fig. 2E. Lower panel showing corresponding mRNA levels of p62, either mock-treated or treated with RBN (10 μM), IFNγ, or both, quantitated by RT-qPCR. (D) Immunoprecipitates of p62 were probed for MAR (HCA-355), ADPr, and p62 from wild-type (WT) and PARP14 knockout (KO) A549 cells after 24-h IFNγ treatment. (E) ADPr and p62 colocalization was analyzed in WT or PARP14 KO cells after 24-h treatment with or without IFNγ, $n = 92$-160 condensates. (F, G) Colocalization of p62 with (F) ADPr, $n = 79$-247 condensates or (G) PARP14, $n = 204$-344 condensates, was analyzed in cells pretreated with or without RBN (10 μM) for 1 h before 24-h treatment with or without IFNγ. (H, I) U2OS cells were transiently transfected with YFP-PARP14 macrodomain 1 (MD1) mutant and treated with (H) DMSO control, $n = 168$-175 condensates, or (I) RBN (10 μM), $n = 165$-283 condensates, for 6 h and analyzed for ADPr condensate formation and PARP14 and p62 colocalization. White arrows indicate the position of condensates, and white boxes represent the zoomed-in regions. The total area of condensates and Pearson correlation coefficient were analyzed by CellProfiler. Mean ± SEM, ns not significant, *$p < 0.05$, **$p < 0.01$, ***$p < 0.001$, ****$p < 0.0001$, unless otherwise stated. (A–C) One-way ANOVA; (E–G) Two-way ANOVA. All $p$ values are provided in Dataset EV2. Data were representative of three biological replicates, unless otherwise stated. Scale bar, 10 μm. Source data are available online for this figure.

early-stage components (Beclin 1, ATG5) nor inhibitors targeting late-stage autophagy (Chloroquine, Bafilomycin A1) affected ADPr condensate formation (Figs. 6A–C and EV6A,B). These findings indicate that autophagy is not required for this process, prompting us to explore the proteasome pathway as a potential alternative mechanism of ADPr regulation.

To test whether active proteasome activity is required for IFNγ-induced ADPr in p62 bodies, we treated cells with proteasome inhibitor MG132. As expected, MG132 induced the formation of aggresomes, detected by the PROTEOSTAT® dye, which binds to misfolded proteins (Fig. EV6C). Notably, this dye did not colocalize with IFNγ-induced ADPr condensates (Fig. EV6C). However, proteasome inhibition eliminated the ADPr condensate signals in a time-dependent manner (Figs. 6D and EV6D). This phenomenon was further confirmed with two more selective proteasome inhibitors, Epoxomicin, and Bortezomib (Velcade; Figs. 6D and EV6E).

Furthermore, upon proteasome inhibition, PARP14 signals within p62 bodies were significantly reduced (Figs. 6E and EV6F,G), and the normally diffuse p62 signal became more prominent as condensates (Figs. 6E and EV6G). Of note, inhibition of the proteasome slightly reduced p62 and PARP14 protein levels (Fig. 6F), which may partly explain the loss of ADPr and PARP14 signals.

The requirement of active proteasome was further demonstrated in U2OS cells transiently expressing PARP14 mutant deficient in ADP-ribosylhydrolase activity. Adding MG132 resulted in the loss of the ADPr signal, even when PARP14 and p62 remained colocalized in condensates (Fig. 6G; cf. Fig. 4H).

Proteasome activity can be mediated through both ubiquitin-dependent and ubiquitin-independent pathways (Erales and Coffino, 2014; Jariel-Encontre et al, 2008; Ciechanover and Schwartz, 1998; Dikic and Schulman, 2023). The ubiquitin-proteasome system involves three major steps of ubiquitination, via E1, E2, and E3 enzymes, followed by proteasomal degradation. To test whether the required proteasome activity is indeed dependent on ubiquitination, we knocked down the ubiquitin-activating enzyme E1 (UBA1) and observed a loss of ADPr condensates (Figs. 7A and EV7A). Similarly, blocking the initiation of ubiquitination with TAK-243, a potent E1 inhibitor, resulted in the absence of ADPr condensates (Fig. 7B). However, unlike treatment with MG132, TAK-243 did not alter the levels of p62 and PARP14 (Fig. EV7B). This indicates that changes in the levels of these two key proteins are not the primary factors regulating ADPr condensation.

Among the steps of ubiquitination, E3 ubiquitin ligases play a crucial role in determining which proteins are ubiquitinated and thus marked for degradation. While both E3 ligases DTX3L and

TRIM25 were increasingly associated with p62 upon IFNγ treatment (Fig. 5K), only DTX3L colocalized with p62 and ADPr (Figs. 7C and EV7C). Consistent with the localization data, the knockdown of DTX3L disrupted the formation of ADPr condensates (Figs. 7D and EV7D).

Ubiquitin is recycled by deubiquitinating enzymes (DUBs) for reuse in signaling events. To assess whether ubiquitin recycling is necessary, we applied the DUB inhibitor PR619, which increased steady-state levels of ubiquitinated proteins (Fig. EV7E). Notably, this treatment did not disrupt condensate formation; instead, the number of IFNγ-induced ADPr condensates increased (Fig. 7E), further suggesting that ubiquitination is critical for ADPr condensate formation. Collectively, these data indicate that an active ubiquitin-proteasome system is required for the condensation of PARP14 and ADPr in p62 bodies.

# Discussions

Here, we have identified that IFNγ activates a transcription program that increases PARP14 mRNA and protein levels, forming cytoplasmic ADPr condensates (Fig. 7F). Elevated levels of PARP14-mediated ADP-ribosylation facilitate its co-condensation with known components of p62 bodies, forming compositionally related structures that neither contain autophagy markers, such as LC3B and FIP200, nor are sensitive to autophagy inhibition. This class of IFNγ-induced ADPr condensates has the following characteristics:

## ADPr condensates are transient and sensitive to PARP14 levels and activity

ADPr condensation requires both the physical presence and catalytic activity of PARP14. However, the mere presence is insufficient if its catalytic activity is inhibited. ADPr condensation is highly sensitive to both its levels and activity. Even brief interventions, such as 1 h catalytic inhibition or PROTAC-induced degradation of PARP14, can inhibit ADPr condensation. This transient nature suggests that PARP14-mediated MARylation is continually removed by endogenous degraders, potentially including the macrodomain ADP-ribosylhydrolase activity inherent to this dual-activity enzyme (Đukić et al, 2023; Torretta et al, 2023; Delgado-Rodriguez et al, 2023).

The expression of a PARP14 ADP-ribosylhydrolase-deficient mutant leads to elevated levels of ADP-ribosylation, with most

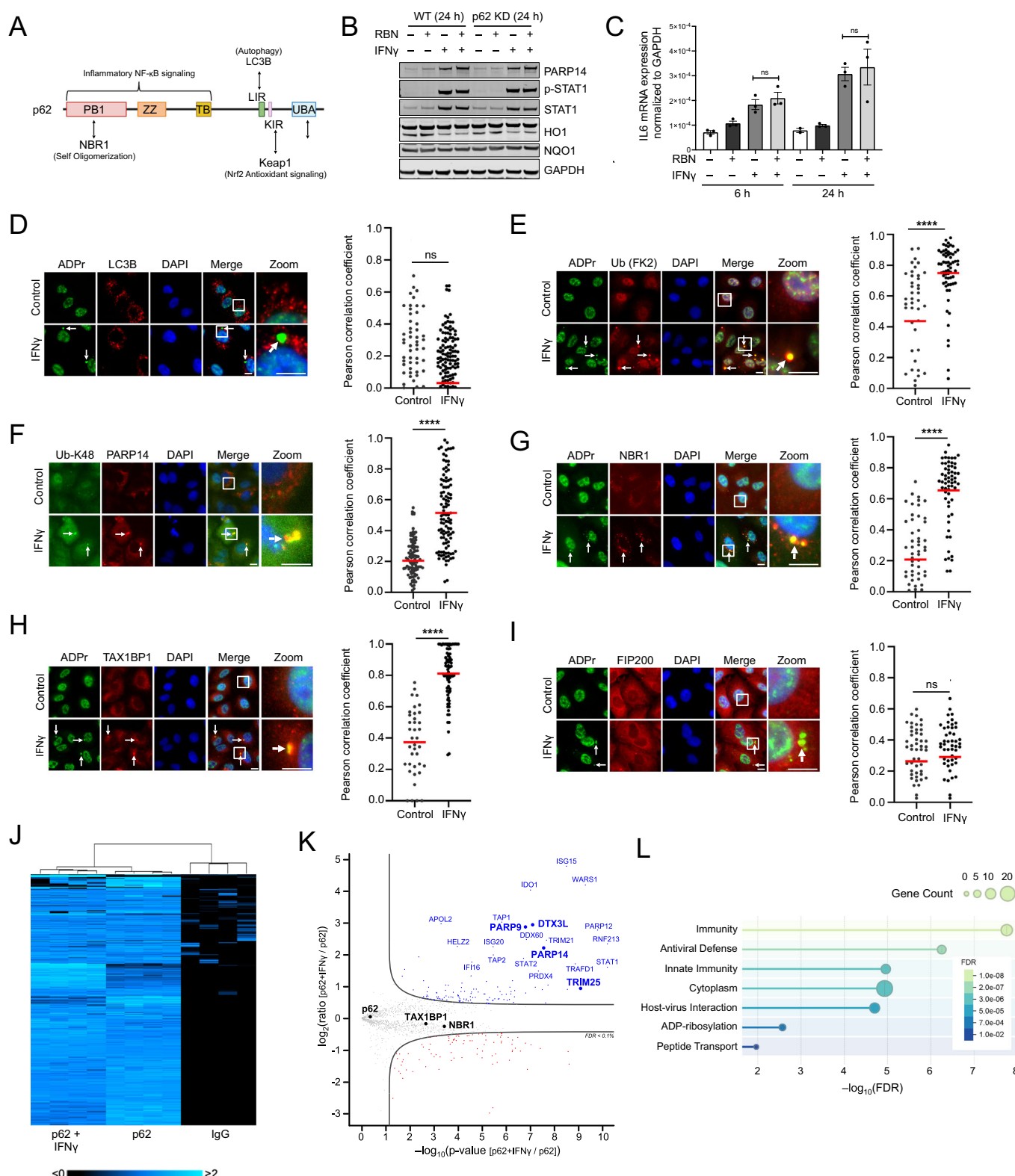

substrates showing sensitivity to PARP14 transferase inhibition (Đukić et al, 2023). This observation underscores that many hydrolase targets are also transferase substrates, and the balance of their ADP-ribosylation is tightly regulated by the opposing enzymatic activities within PARP14. Consistent with our findings, recent studies reported after our submission confirm that ADPr condensates depend on PARP14 and have identified PARP14 and p62 as ADP-ribosylated in a PARP14-dependent manner (Kar et al, 2024; Ribeiro et al, 2024; Kubon et al, 2024). Notably, both PARP14 and p62 were also shown to be substrates for its hydrolase activity

**Figure 5. ADPr-enriched p62 bodies contain ubiquitin but lack autophagy marker LC3B.**

(A) Schematic of p62 domain architecture. (B) p-STAT1, STAT1, HO1, and NQO1 levels were measured in A549 wild-type (WT) and p62 knockdown (KD) cells treated with different combinations of treatments with RBN (10 μM) and IFNγ. (C) IL6 mRNA levels were measured in A549 cells treated with different combinations of RBN (10 μM) pretreatment for 1 h and IFNγ treatment for either 6 or 24 h by RT-qPCR. (D–I) A549 cells after 24-h treatment with or without IFNγ were analyzed for colocalization between (D) ADPr and autophagy marker LC3B, n = 118–182 condensates, (E) ADPr and ubiquitination (FK2), n = 45–65 condensates, (F) PARP14 and Ub-K48, n = 102–125 condensates, (G) ADPr and NBR1, n = 60–61 condensates, (H) ADPr and TAX1BP1, n = 38–79 condensates, and (I) ADPr and FIP200, n = 58 condensates. (J) Heatmap showing proteins identified using mass spectrometry (MS)-based proteomics analysis, following immunoprecipitation of proteins with p62 from A549 cell lysates which were either untreated or treated with IFNγ for 24 h. IgG rabbit antibody was used as a control. Coloring indicates z-score, ranging from black (z < 0), to blue (z = 1), to cyan (z > 2). (K) Volcano plot of MS-identified proteins showing significant changes in the p62 interactome after 24 h of IFNγ treatment. Key p62 interactors mentioned in the text are bolded, while other interactors with increased association (log2 ratio >1.5 and −log10 p value >3) are labeled. Significance testing via two-tailed Student's t-test, corrected for multiple-hypotheses testing via permutation-based FDR control (corrected p value <0.1%); n = 4 biological replicates. (L) Term enrichment analysis showing UniProt keywords statistically enriched within the p62-specific network after IFNγ treatment. Statistical testing was conducted using the STRING database with default settings. White arrows indicate the position of condensates, and white boxes represent the zoomed-in regions. The total area of condensates and Pearson correlation coefficient were analyzed by CellProfiler. Mean ± SEM, ns not significant, *p < 0.05, **p < 0.01, ***p < 0.001, ****p < 0.0001, unless otherwise stated. (C) One-way ANOVA; (D–I) Unpaired t-test. All p values are provided in Dataset EV2. Data were representative of three biological replicates, unless otherwise stated. Scale bar, 10 μm. Source data are available online for this figure.

(Đukić et al, 2023). Since more ADP-ribosylation is observed for these and other substrates upon IFNγ treatment, cells may respond to the evolving immune environment by adjusting the enzymatic balance of PARP14—either via reducing hydrolase activity or enhancing transferase activity.

## High levels of PARP14-mediated ADP-ribosylation are sufficient to induce their condensation in a specific subset of p62 bodies

PARP14 associates with p62 prior to IFNγ treatment, and these associations increase after treatment, likely due to the increased expression of PARP14. While this association is necessary, it is insufficient for co-condensation. PARP14 and p62 continue to associate when PARP14's transferase activity is inhibited, yet they fail to co-condense, even with IFNγ treatment. Elevated protein levels typically increase condensation likelihood (Banani et al, 2017; Alberti and Dormann, 2019; Shin and Brangwynne, 2017; Spannl et al, 2019). However, despite higher PARP14 levels after RBN treatment, PARP14 condensation does not occur without ADP-ribosylation. Instead, high levels of PARP14-mediated ADP-ribosylation are required for the co-condensation of PARP14, ADPr, and p62. This process can be triggered either by IFNγ treatment or by expressing an ADP-ribosylhydrolase-deficient PARP14 mutant without IFNγ treatment.

Such co-condensation occurs in ~20% of p62 bodies: they are slightly larger and contain canonical components such as p62, its binding partner NBR1, TAX1BP1, and K48- and K63-linked polyubiquitin chains, yet notably lack the autophagosome-associated LC3B and FIP200 (Fig. 7F). These co-localizations are consistent with our proteomics data, which show that certain proteins, such as NBR1 and TAX1BP1, remain associated with p62 upon IFNγ treatment, while others, such as PARP14, PARP9, and DTX3L, become increasingly associated with p62. Notably, PARP9 and DTX3L also co-localize with ADPr-containing condensates upon IFNγ treatment (Kar et al, 2024; Ribeiro et al, 2024). Importantly, the genetic depletion of key autophagy players ATG5 and Beclin 1 does not inhibit these condensations, further supporting that these structures are distinct from canonical p62 bodies. Our FRAP analyses reveal that while the overall exchange rate of p62 with the cytoplasm remains constant, the fraction of p62 available for exchange increases. These compositional and dynamic differences align with the transient nature of these condensates,

likely facilitating their response to changes in the immune environment.

## The formation of ADPr-containing p62 bodies requires an active ubiquitin-proteasome system

Unlike canonical p62 bodies, these ADPr-containing ones are insensitive to autophagy inhibition. Instead, they are sensitive to the inhibition of ubiquitin activation and proteasome activity, indicating that their formation requires an intact ubiquitin-proteasome system. Crucially, p62 still forms condensates in the presence of proteasome or ubiquitin-activating enzyme E1 inhibitors; however, these condensates lack PARP14 and ADPr, highlighting a selective composition regulation within p62 bodies.

The requirement for an active ubiquitin-proteasome system led us to explore existing data on the crosstalk between ubiquitination and ADP-ribosylation. Recent data show that PARP9 and DTX3L bind to PARP14 and regulate its stability (Saleh et al, 2024). This stabilization, critical for increasing PARP14 levels, is independent of the proteasome (Saleh et al, 2024), implying it is unrelated to the IFN-induced ADPr condensate regulation.

Instead, the IFN-induced ADP-ribosylation depends on the ubiquitination of proteins that are destined for proteasome degradation, consistent with the enrichment of K48-linked polyubiquitin chains in these ADPr-containing p62 bodies. A potential enzyme responsible for this modification is the E3 ligase DTX3L, given its physical association and colocalization with PARP14 (Saleh et al, 2024). Consistent with this premise, the genetic depletion of DTX3L results in the loss of IFN-induced ADPr (Russo et al, 2021). Our proteasome inhibitor data now further suggest that not only the physical presence but also the enzymatic activity of DTX3L or other E3 ligases is critical, particularly their ability to ubiquitinate proteins with K48 linkages, which targets these poly-ubiquitinated proteins for degradation (Dikic and Schulman, 2023). Notably, inhibiting ubiquitination results in the loss of ADPr condensates, whereas increasing ubiquitination promotes ADPr condensate formation, suggesting that ADPr may play a downstream role in ubiquitin signaling within the proteasomal degradation pathway. Further investigation is warranted to identify these ubiquitinated protein substrates for proteasome degradation and to clarify ADPr's role in protein degradation during IFNγ treatment.

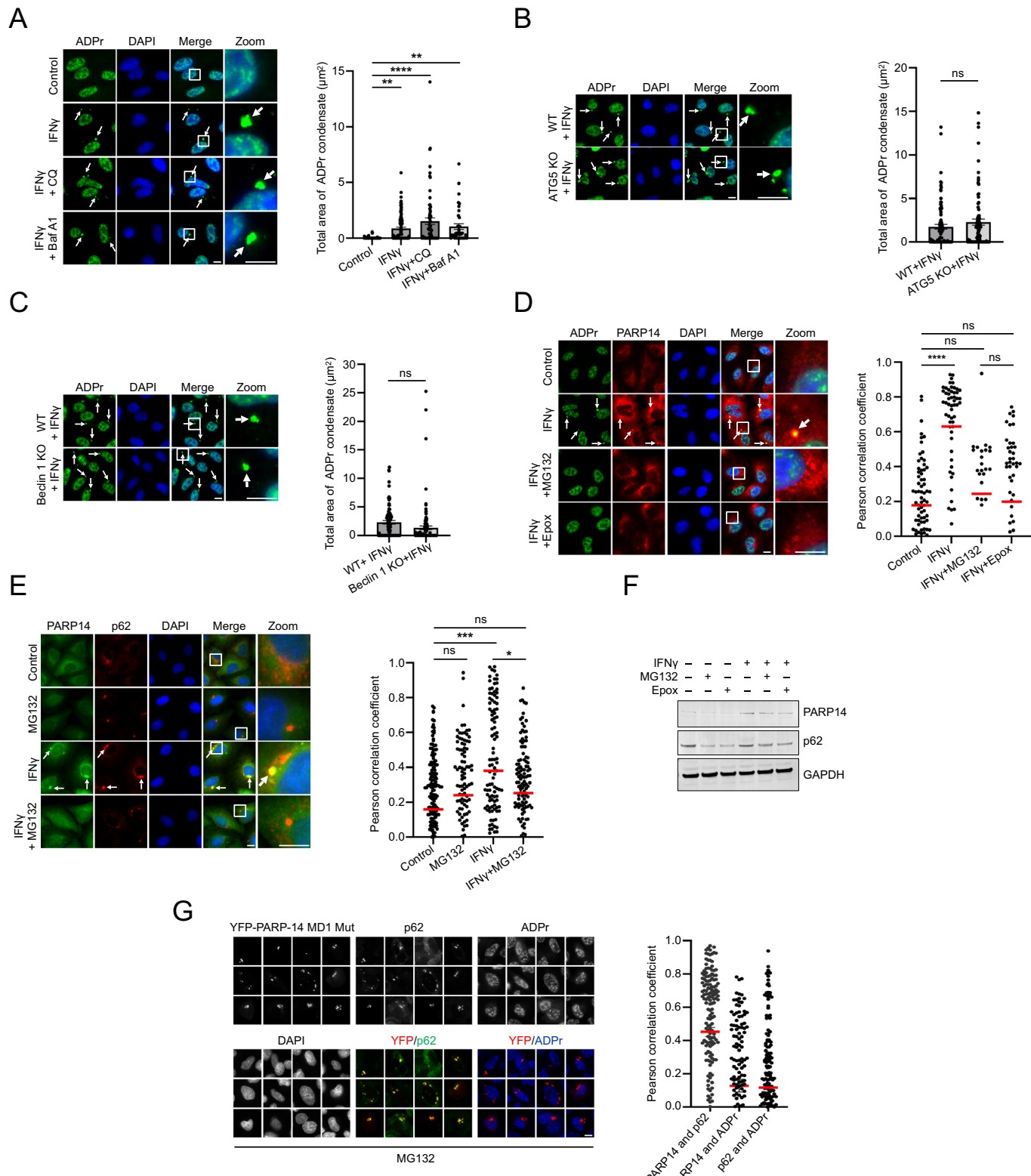

It, however, remains unclear how inhibition of different components of the ubiquitin-proteasome system leads to decreased PARP14-mediated ADP-ribosylation in p62 bodies. While MG132 slightly reduces the amount of PARP14, TAK-243 does not have this effect. Thus, factors other than the modulation of

PARP14 levels are likely contributing to the decreased ADP-ribosylation. One possibility is that the ubiquitin-proteasome system might reduce PARP14-mediated ADP-ribosylation by degrading either a repressor of PARP14 transferase activity and/or an activator of its hydrolase activity, following IFNγ treatment.

**Figure 6. Ubiquitin-mediated proteasome activity is required for PARP14 and ADPr co-condensation with p62 bodies.**

(A) ADPr condensate formation was analyzed in A549 cells treated with or without IFNγ overnight, followed by the treatment with autophagy inhibitors 100 μM Chloroquine (CQ) or 200 nM Bafilomycin A1 (Baf A1) for 6 h, $n = 68$–213 cells. (B) ADPr condensates were analyzed in either A549 wild-type (WT) or ATG5 knockout (KO) cells after 24-h treatment with or without IFNγ, $n = 244$–410 cells. (C) ADPr condensates were analyzed in either A549 WT or Beclin 1 KO cells after 24-h treatment with or without IFNγ, $n = 236$–305 cells. (D) ADPr and PARP14 colocalization was assessed in cells treated with IFNγ overnight, followed by treatment with proteasome inhibitors 10 μM MG132 or 1 μM Epoxomicin (Epox) for 6 h, and compared to untreated control, $n = 33$–79 condensates. (E) PARP14 and p62 colocalization was analyzed in cells treated with or without IFNγ overnight, followed by treatment with 10 μM MG132 for 6 h, $n = 114$–196 condensates. (F) PARP14 and p62 protein levels were measured in cells treated with or without IFNγ overnight, followed by treatment with proteasome inhibitors 10 μM MG132 or 1 μM Epoxomicin for 6 h. (G) U2OS cells were transiently transfected with YFP-PARP14 macrodomain 1 (MD1) mutant and analyzed for p62, PARP14 (YFP), and ADPr colocalization, after treatment of 10 μM MG132 for 6 h, $n = 127$–175 condensates. White arrows indicate the position of condensates, and white boxes represent the zoomed-in regions. The total area of condensates and Pearson correlation coefficient were analyzed by CellProfiler. Mean ± SEM, ns not significant, $*p < 0.05$, $**p < 0.01$, $***p < 0.001$, $****p < 0.0001$, unless otherwise stated. (A) One-way ANOVA; (B, C) Unpaired $t$-test; (D, E) Two-way ANOVA. All $p$ values are provided in Dataset EV2. Data were representative of three biological replicates. Scale bar, 10 μm. Source data are available online for this figure.

## Physiological and pathological implications

These IFNγ-induced ADPr condensates were initially identified while developing a cell-based assay to screen inhibitors of viral ADP-ribosylhydrolase activity (Russo et al, 2021). Expressing the first of the three macrodomains from SARS-CoV-2—the only one with hydrolase activity—reduces IFNγ-induced ADPr (Russo et al, 2021). Remarkably, the three-macrodomain architecture in the nsP3 gene of SARS-CoV-2 closely resembles that of PARP14, where only the first macrodomain exhibits ADP-ribosylhydrolase activity (Đukić et al, 2023; Torretta et al, 2023; Delgado-Rodriguez et al, 2023). This similarity raises the possibility that the virus exploits the host's endogenous system, targeting the same substrates and shifting the balance toward hydrolysis of PARP14-mediated ADP-ribosylation in p62 bodies. Supporting their critical role in host-virus interactions, proteins increasingly associated with p62 upon IFNγ treatment are enriched in functions related to cytokine signaling, immune responses, and defense mechanisms (Figs. 5L and EV5P,Q).

Beyond this structural mimicry, the three macrodomains are flanked by two ubiquitin-like domains—a conserved feature across coronaviruses—with the second domain followed by the papain-like protease (PLpro) domain in nsP3, which has deubiquitinating activity (Lei et al, 2018). Recent findings show that DTX3L can ubiquitinate ADP-ribosyl modifications on protein substrates (Zhu et al, 2022). Our data further reveal that an active ubiquitin-proteasome system is critical for the formation of PARP14-mediated ADPr condensates upon IFN stimulation, where the E3 ligase DTX3L also colocalizes. These emerging data suggest a complex interplay between ADP-ribosylation and ubiquitination pathways in regulating protein condensation processes during viral infection.

Besides its role in viral infection, IFNγ can also promote resistance to immunotherapy, such as those involving anti-PD-1 antibodies (Gocher et al, 2022; Wong et al, 2023; Martínez-Sabadell et al, 2022). PARP14 upregulation has been observed in melanoma cell cultures derived from patient tumors resistant to immunotherapy with elevated IFNγ signaling (Wong et al, 2023). Notably, in mouse models, restoring sensitivity to anti-PD-1 therapy can be achieved by PARP14 inhibitor RBN (Wong et al, 2023). As ADPr-containing p62 bodies are also observed in melanoma cells following IFNγ treatment—and that such ADPr/PARP14 condensation is lost upon RBN treatment—understanding how PARP14-mediated ADP-ribosylation in p62 bodies functions could open new avenues for overcoming immunotherapy resistance.

## Methods

### Reagents and tools table

| Antibodies | Source | Catalog Number | Host | Dilution WB | Dilution IF |
|---|---|---|---|---|---|
| SQSTM1/p62 (D5L7G) | CST | 88588 | Mouse | 1:1000 | 1:500 |
| SQSTM1/p62 (D10E10) | CST | 7695 | Rabbit | 1:1000 | 1:250 |
| PARP14 | Santa Cruz Biotechnology | sc-377150 | Mouse | 1:1000 | 1:250 |
| PARP14 | Ribon Therapeutics/ Genscript | U9980EK130-3 | Rabbit | | 1:250 |
| PARP14 | Abcam | Ab224352 | Rabbit | | 1:100 |
| Pan-ADPr | Millipore | MABE1016 | Rabbit | 1:500 | 1:500 |
| Mono-ADPr (HCA-355) | Bio-Rad | abD33205 | Rabbit | 1:500 | 1:500 |
| Calnexin (AF18) | Thermo Fisher Scientific | MA3-027 | Mouse | | 1:250 |
| GM130 | BD Transduction Laboratories | 610822 | Mouse | | 1:250 |
| G3BP1-H10 | Santa Cruz Biotechnology | SC-365338 | Mouse | | 1:250 |
| LAMP1 (D401S) | CST | 15665 | Mouse | | 1:100 |
| LC3B (E5Q2K) | CST | 83506 | Mouse | | 1:500 |
| LMP7 (1A5) | CST | 13726 | Mouse | | 1:50 |
| Stat1 | CST | 9172 | Rabbit | 1:1000 | |
| Phospho-Stat1(58D6) | CST | 9167 | Rabbit | 1:1000 | |
| HO1 (A-3) | Santa Cruz Biotechnology | sc-136960 | Mouse | 1:1000 | |
| NQO1 (A180) | Santa Cruz Biotechnology | SC3187 | Mouse | 1:1000 | |
| DDX6 | Sigma-Aldrich | SAB4200837 | Mouse | | 1:100 |
| GAPDH (6C5) | EMD Millipore | MAB374 | Mouse | 1:10000 | |
| β-Actin | Sigma-Aldrich | A1978 | Mouse | 1:10000 | |
| NBR1 (E6Q3F) | CST | 20145 | Rabbit | | 1:100 |
| NBR1 (5C3) | Abcam | ab55474 | Mouse | | 1:100 |
| Ub (P4D1) | Santa Cruz Biotechnology | SC8017 | Mouse | 1:1000 | 1:50 |
| Ub (FK2) | EMD Millipore | 04-263 | Mouse | | 1:500 |
| Ub-K48 | Sigma-Aldrich | ZRB2150 | Rabbit | | 1:100 |
| Ub-K63 | EMD Millipore | 05-1306 | Rabbit | | 1:200 |
| GFP | Roche | | Mouse | 1:1000 | |
| 20S Proteasome β2 | Santa Cruz Biotechnology | SC-54810 | Mouse | | 1:50 |
| Cy5-Donkey Anti-Mouse IgG | Jackson ImmunoResearch | 715-175-150 | Donkey | | 1:500 |
| Cy3-Donkey Anti-Mouse IgG | Jackson ImmunoResearch | 715-165-150 | Donkey | | 1:500 |

| Antibodies | Source | Catalog Number | Host | Dilution WB | Dilution IF |
|---|---|---|---|---|---|
| FITC-Fab Donkey Anti-Rabbit IgG | Jackson ImmunoResearch | 711-097-003 | Donkey | | 1:500 |
| Cy5-Donkey Anti-Rabbit IgG (H + L) | Jackson ImmunoResearch | 711-175-152 | Donkey | | 1:500 |
| FITC-Donkey Anti-Mouse IgG (H + L) | Jackson ImmunoResearch | 715-095-150 | Donkey | | 1:500 |
| **Reagents/Chemicals** | | | | | |
| MG132 | Sigma-Aldrich | C2211 | | | |
| Epoxomicin | MedChem Express | HY-13821 | | | |
| Bortezomib | MedChem Express | HY-10227 | | | |
| Torin 1 | MedChem Express | HY-13003 | | | |
| Rapamycin | MedChem Express | HY-10219 | | | |
| Chloroquine | CST | 14774 | | | |
| Bafilomycin A1 | MedChem Express | HY-100558 | | | |
| ITK7 | Sigma-Aldrich | SML2669 | | | |
| Olaparib | Selleckchem | AZD2281 | | | |
| XAV939 | Selleckchem | S1180 | | | |
| RBN012759 | MedChem Express | HY-136979 | | | |
| RBN012811 | Ribon therapeutics | N/A | | | |
| RBN013527 | Ribon therapeutics | N/A | | | |
| OUL-35 | Dr. Lari Lehtio | N/A | | | |
| ITK6 | Dr. Michael Cohen | N/A | | | |
| Actinomycin D | CST | 15021 | | | |
| PROTEOSTAT Aggresome Detection kit | Enzo Life Sciences | ENZ-51035 | | | |
| Puromycin | Invivogen | ant-pr-1 | | | |
| Human IFNα | PBL Assay Science | 11200-1 | | | |
| Human IFN Beta | PBL Assay Science | 11415-1 | | | |
| Human IFNγ | Sigma-Aldrich | SRP3058 | | | |
| Lipofectamine 2000 | Invitrogen | 11668019 | | | |

| Oligonucleotides and other sequence-based reagents | | |
|---|---|---|
| Gene | Forward Primer | Reverse Primer |
| PARP1 | TGAGGTCCAGCAGGCGGTGT | AGTCGTGGGGGATCAGGGTGT |
| PARP2 | ATAGCACCAAGGGGCTGGGCA | GGCACTGTACTCCCATTCAGGGTG |
| PARP3 | TGCTCCAAGACAGCAACCGCT | GACTGGCCGACCTCTCCCACA |
| PARP4 | TCGCCTGCGGTATCGGTTCT | ACTCCGGCACCACACTGGGA |
| PARP5a | AGCAGGCTACAACCGCGTGTC | GGGCACCAAGCCACCCTTGT |
| PARP5b | CTTCGCCGCAGGTTTTGGGC | GCCCCCATCATCACGTGCTTGG |
| PARP6 | ACTCCGGCACCACACTGGGA | ACTCCGGCACCACACTGGGA |
| PARP7 | AAAGGAGACCCCTCTTCCGCTCC | GTGGGAACCCCACCAAGTGTCTGTA |
| PARP8 | CTTTGCGCAGGTCGCCAAGTT | CCGCAGCAGCTTATGCGTTTTGG |
| PARP9 | TGGCGCTCGTTAGGACAGTGG | GCTCTTGAGTTGGAGGCACAGGA |
| PARP10 | GCACCGGCAGTGCCTGACAT | CCCGTAGACCGTGGCGTTGC |
| PARP11 | ACGTCAGATACCCAGTGGGGCTG | TGGTATCCGGCTGAAACATGTGC |
| PARP12 | CCTACGGCAAGGGGAGCTACTT | TCGTGTGGGTCTGCGTGTCG |
| PARP13.1 | GAGAGGGCCAGACCATCAGCCA | CCTCCTGAGGACGAAAGGTCGCA |
| PARP13.2 | GCAGCAGATGAAGAGAGGGCCA | TGAGCCCAGGGCATGAACATCT |
| PARP14 | GGCTCCGTGCCACACGTCAA | CATATGCCACAGCATTCTTTCCGGC |

| Antibodies | Source | Catalog Number | Host | Dilution WB | Dilution IF |
|---|---|---|---|---|---|
| PARP15 | TCGGAAATCAGGTGTGTCAAGAGC | TCTGTGAGGCTGTGCAGCTAGT | | | |
| PARP16 | AACCAGCTGCTGCGAGTGAAGT | GGCTCGAAGCCCTGCTCTTGG | | | |
| OAS1 | CAAGCTCAAGAGCCTCATCC | TCGGGCTTGTGTTGAAATGTGT | | | |
| IL6 | GTAGCCGCCCCACACAGA | CATGTCTCCTTTCTCAGGGCT | | | |
| GAPDH | ACCACAGTCCATGCCATCAC | TCCACCACCCTGTTGCTGTA | | | |

## Cell lines and reagents

A549 (CCL-185), HEK293T (CRL-3216), and U2OS (HTB-96) cells were purchased from the American Type Culture Collection (Manassas, VA, USA). A549 PARP14 KO (Grunewald et al, 2019) and its wild-type counterpart were received as a kind gift by Dr. Christopher S. Sullivan, University of Texas, Austin. A375 Melanoma cell line was a kind gift from Dr. Vito W. Rebecca, Johns Hopkins University. A549 shPARP14, as well as shp62 knockdown cells, were generated in the lab using lentiviral shRNA constructs TRCN0000290897 and TRCN0000007237 (Sigma). All cells were cultured in Dulbecco's modified Eagle's medium (DMEM; Gibco; Waltham, MA, USA) supplemented with 100 U/mL penicillin and streptomycin, 10% heat-inactivated fetal bovine serum (FBS; Gibco; Waltham, MA, USA) at 37 °C in 5% $CO_2$ atmosphere. IFNγ (Sigma, #SRP3058) 500 IU/ml was used for most of the experiments at indicated time points in A549 cells. The reagents and tools table details all drugs used along with IFNγ.

## Lentivirus transduction and stable cell line generation

HEK293T cells were seeded into a 10 cm dish to reach ~80% confluence by the following day. Sixteen hours after seeding, cells were transfected using Lipofectamine 2000 (Invitrogen, 11668019) according to the manufacturer's instructions with 18 µg of the following plasmid cocktail: lentiviral shRNA constructs from Sigma-Aldrich (shPARP14 TRCN0000290897 and TRCN0000296754 or shp62 TRCN0000007237 and TRCN0000007236), shDTX3L (TRCN0000073211), shPARP9 (TRCN0000052917), shPARP12 (TRCN0000004614), shRNA-UBA1 (TRCN0000004004) and pLentiCrispr-v2 constructs for Beclin 1 sgRNA: 5′CCTGGACCGTGTCACCATCC3′ and ATG5 sgRNA: 5′AACTT GTTTCACGCTATATC3′, psPAX2 (Addgene, #12260), and pMD2.G (Addgene, #12259) in a ratio of 3:2:1, respectively, to generate lentiviral particles. 48 h post-transfection, the supernatant was collected, centrifuged at 500×g, and filtered through a 0.45-µm low-protein-binding filter. The resulting virus suspension containing lentiviral particles was used to transduce A549 cells overnight. Fresh media containing puromycin (Sigma) at a concentration of 1 µg/ml was then added for selection, and single-cell clones were picked for shRNA-based genetic depletion of PARP14 and p62 in A549 cells.

To generate Beclin 1 and ATG5 CRISPR knockout (KO) cells, lentiviral particles were produced as described above and subsequently used to infect A549 cells. Following infection, cells were subjected to puromycin selection. The selected cell pool was then utilized for further experimental analyses.

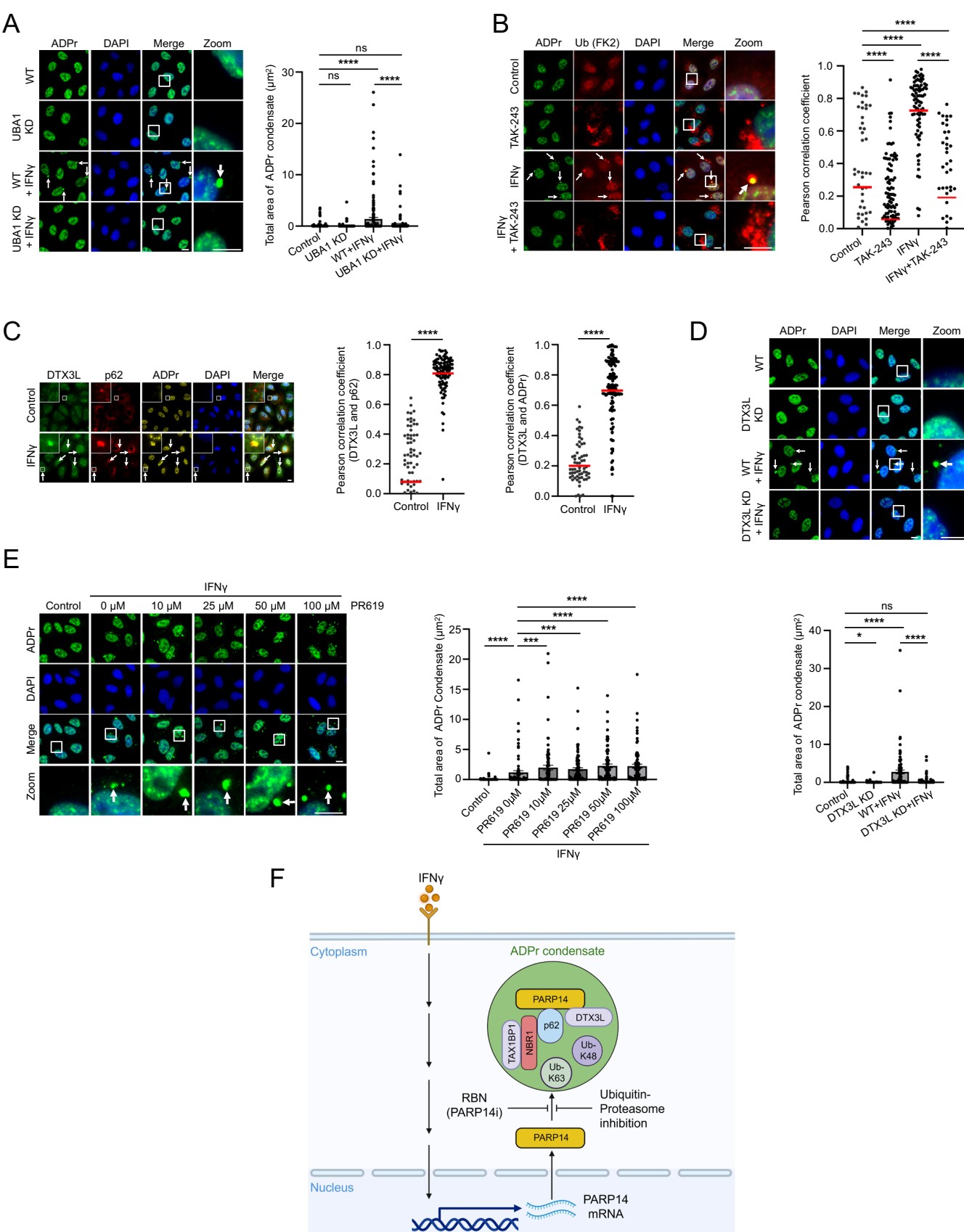

**Figure 7. Ubiquitin-mediated proteasome activity is required for PARP14 and ADPr co-condensation with p62 bodies.**

(A) ADPr condensates were analyzed in either wild-type (WT) or shRNA-UBA1 knockdown (KD) lentiviral knockdown cells after 24-h treatment with or without IFNγ, $n = 160$–313 cells. (B) ADPr and ubiquitination (FK2) colocalization was analyzed in cells treated with or without IFNγ overnight, followed by treatment with 1 µM E1 inhibitor TAK-243 for 6 h, $n = 60$–160 condensates. (C) DTX3L and p62 or ADPr colocalization was analyzed in cells treated with or without IFNγ 24 h, $n = 97$–100 condensates for the p62 panel and $n = 65$–144 condensates for the ADPr panel. (C) ADPr condensates were analyzed in either WT or shRNA-DTX3L KD lentiviral knockdown cells after 24-h treatment with or without IFNγ, $n = 228$–368 cells. (E) ADPr condensate formation was assessed in A549 cells after overnight IFNγ treatment, followed by treatment with different doses of PR619 for 8 h, and compared to untreated control, $n = 202$–424 cells. (F) Working model: IFNγ transcriptionally activates PARP14, leading to the formation of ADPr-containing p62 bodies, which require an active ubiquitin-proteasome system. White arrows indicate the position of condensates, and white boxes represent the zoomed-in regions. The total area of condensates and Pearson correlation coefficient were analyzed by CellProfiler. Mean ± SEM, ns not significant, $*p < 0.05$, $**p < 0.01$, $***p < 0.001$, $****p < 0.0001$, unless otherwise stated. (A, B, D, E) Two-way ANOVA; (C) Unpaired $t$-test. All $p$ values are provided in Dataset EV2. Data were representative of three biological replicates. Scale bar, 10 µm. Source data are available online for this figure.

## Quantitative real-time PCR

Total RNA was isolated from A549 cells under different treatment conditions using the High Pure RNA Isolation Kit (Roche), following the manufacturer's instructions. The RNA was quantified and reverse transcribed with random hexamers, using the SuperScript VILO cDNA Synthesis Kit (Invitrogen) at 25 °C for 10 min, 50 °C for 10 min, and 85 °C for 5 min. The resulting cDNA was mixed with the appropriate primers (Integrated DNA Technologies; Coralville, IA) and SYBR Green PCR Master Mix (Applied Biosystems) and then amplified for 40 cycles (30 s at 94 °C, 40 s at 60 °C, and 30 s at 72 °C) on a 7500 Fast Real-Time PCR System (Applied Biosystems). The target gene expression was normalized to GAPDH, and the relative gene expression level was calculated using the $2^{-\Delta CT}$ method. All primer sequences are listed in the Reagents and tools table.

## Immunoblotting

Cells were lysed in RIPA lysis buffer (CST, #9806) in the presence of protease inhibitors (Roche, # 11873580001) and phosphatase inhibitors (Roche, #4906845001), 10 µM Olaparib and 10 µM PDD 00017273 (PARG inhibitor) at 4 °C for 30 min and centrifuged at $21,130 \times g$ at 4 °C for 20 min. The lysates were quantitated using BCA assay (Thermo Fisher Scientific, #23225) and 20 µg of proteins were electrophoresed on SDS–PAGE gels and then transferred to PVDF membrane (Thermo Fisher Scientific, # 88518). The membranes were blocked using 5% bovine serum albumin (BSA) (Sigma, #A9647) in 1X TBST buffer (150 mM NaCl; Tris, pH 7.4; 0.1% Tween-20) for 1 h at room temperature. Blots were probed overnight at 4 °C with primary antibodies, washed with TBST (three times, 10 min each), and incubated with the appropriate secondary antibodies for 1 h at room temperature. Blots were washed again with TBST (three times, 10 min each), and signals were detected on the Odyssey CLx imaging system using Image Studio Lite software (LI-COR). Western blot image data was processed by ImageJ software. The primary and secondary antibodies used are available in the Reagents and tools table.

## Immunoprecipitation

Cells were lysed in immunoprecipitation buffer containing 20 mM HEPES (pH 7.5), 150 mM NaCl, 10 mM NaF, 1.5 mM MgCl$_2$, 10 mM β-glycerophosphate, 2 mM EDTA, 5 mM Na-pyrophosphate, 1 mM Na$_3$VO$_4$, 1% (v/v) Triton X-100, 0.2% NP-40, and protease and phosphatase inhibitors and 10 µM Olaparib and 10 µM PDD 00017273 (PARG inhibitor) at 4 °C for 30 min and centrifuged at $21,130 \times g$ at 4 °C for 20 min. The supernatant was

incubated overnight with antibodies at 4 °C, followed by incubation with Protein A/G PLUS-Agarose beads (Santa Cruz Biotechnology, #SC-2003) or Ni-NTA resin (Thermo Fisher Scientific, # 88222) for 2–4 h with rotation. The related beads were washed with IP buffer and boiled in 2x LDS buffer (Novex) for 10 min at 70 °C. For Ni-NTA pull down, imidazole was added to 2x LDS buffer at 300 mM concentration to facilitate the release of immunoprecipitated protein in 2x LDS buffer. The samples were run in SDS–PAGE gel, transferred onto PVDF membranes (Bio-Rad), and processed for western blot analysis.

## Denaturing immunoprecipitation

To analyze p62 MAR modification, A549 cells were treated with IFNγ overnight, followed by lysis in denaturing buffer containing 20 mM Tris, pH 7.5, 50 mM NaCl, 1 mM EDTA, 0.5% NP-40, 1% SDS, 0.5% sodium deoxycholate, 1 mM DTT supplemented with protease and phosphatase inhibitors along with 10 µM Olaparib and 10 µM PDD 00017273 (PARG inhibitor) at 4 °C for 30 min and centrifuged at $21,130 \times g$ at 4 °C for 20 min, followed by sonication. Lysates were then cleared by centrifugation ($21,000 \times g$, 20 min) and incubated with GFP-Trap beads (gtma-20, ChromoTek) overnight at 4 °C. For each condition, 1 mg of protein was used. After overnight binding, the GFP-Trap beads were washed three times with lysis buffer, 10 min each wash, followed by a high salt wash with lysis buffer containing 300 mM NaCl. After washing with lysis buffer, beads were boiled in 2x LDS buffer (Novex) for 10 min at 70 °C. The samples were run in SDS–PAGE gel, transferred onto PVDF membranes (Bio-Rad), and processed for western blot analysis.

## Transient transfections

U2OS cells ($5 \times 10^4$ cells per well) were transiently transfected with 300 ng of plasmid construct YFP-PARP14 MD1 Mut (Kind gift from Dr. Ivan Ahel, University of Oxford) using Lipofectamine 3000 (Invitrogen, L3000001) as per manufacturer's instructions. The media was replaced 6 h post-transfection. Cells were fixed with 4% PFA 24 h post-transfection. If inhibitors like MG132 (10 µM) and RBN (10 µM) were used, they were added 24 h post-transfection, with a treatment time of 6 h followed by cell fixation.

## Immunofluorescence

A549 cells upon IFNγ and/or drug treatment ($5 \times 10^4$ cells per well) or U2OS cells transiently transfected and/or drug treatment ($5 \times 10^4$ cells per well) in Ibidi µ-24 well plate were fixed with 4% paraformaldehyde (PFA) for 20 min. Following fixation, cells were

permeabilized using 0.2% Triton X-100 in PBS for 20 min, washed twice with PBS, and blocked with 5% BSA in PBS for 1 h at room temperature and then labeled overnight with corresponding primary antibodies. The next day, cells were washed three times with PBS, and secondary antibodies (Jackson Immunoresearch) were added. The nuclei was stained with DAPI (5 μg/ml) and washed thrice with PBS before imaging. Images were acquired using a Leica Thunder Imaging System with 20X or 40X magnification and processed with Leica LAS X software and Image J.

Colocalization data was analyzed in Fiji (ImageJ) software using Pearson Correlation. Initially, the color channels of the images were separated. Nonspecific signals were minimized by applying noise median and despeckling tools to correct the background. After these steps, the Colocalization plugin was used to analyze how much two signals overlap. p62 bodies were identified with the built-in "Analyze Particles…" command with particle size set at greater than 0.5 μm$^2$ and particle circularity greater than 0.7 and plotted.

The image analysis workflow in CellProfiler included segmentation of nuclei stained with DAPI, designated as primary objects, while cytoplasmic condensates were classified as secondary objects. For 20X images, condensates were identified within a pixel range of 1–30, and for 40X images, the range was set to 1–50. Cells with over 20 and 6 condensate structures for 40X and 20X images, respectively, were identified as outliers and assumed to be mitotic cells or having cell death characteristics. Mitotic cells have uniform ADPr signal across the cytoplasm, and condensate structures were not identified in such cells. These cells are excluded from subsequent analyses. Mean intensity, mean area, and counts were quantified for each cell, which enabled the calculation of the total area of condensates per cell.

For Pearson correlation analysis of 40X images, condensates were identified within a pixel range of 1–50 and then expanded by 15 pixels. The Pearson correlation coefficient for each expanded area was calculated with a threshold set at 50% of maximum intensity. In 63X super-resolution images, condensates were identified within a pixel range of 3–50 and expanded by 50 pixels, with the Pearson correlation coefficient similarly determined at a 50% threshold.

For morphometric analysis of ADPr condensates, parameters such as area, eccentricity, and solidity were assessed using the Size/Shape measurement module in CellProfiler. The aspect ratio was calculated as the ratio of the major axis length to the minor axis length for each region of interest.

## Fluorescence recovery after photobleaching (FRAP) analysis

FRAP experiments were performed on a Leica Thunder Imaging system equipped with an infinity scanner for photobleaching and a Leica DFC9000 GTC camera for fluorescence imaging. A549 cells ($5 \times 10^4$ cells per well) were transiently transfected with 500 ng GFP-p62 (kind gift of Dr. Terje Johansen) using Lipofectamine 2000 (Invitrogen,11668019), according to the manufacturer's instructions. Six hours after transfection, cells were treated with IFNγ overnight, followed by RBN treatment for 6 h, before performing FRAP analysis. GFP-tagged p62 condensates were photobleached using 100% 488 nm laser irradiation for 50 iterations. Time-lapse images were acquired post-bleaching every 10 s to monitor fluorescence recovery using a 63X oil objective (HC

PL APO 63X/1.40-0.60 OIL), and fluorescence intensity within the bleached ROI and a reference unbleached region was measured using ImageJ. The fluorescence intensities were normalized by calculating their ratios to the fluorescence intensity of the same ROI before photobleaching. The recovery curves were fit by a single exponential equation:

$$F(t) = F_{\min} + (F_{\max} - F_{\min})(1 - e^{-kt})$$

Where:

- $F(t)$ is the normalized fluorescence intensity at time $t$,
- $F_{\max}$ is the maximum fluorescence intensity recovered,
- $F_{\min}$ is the normalized fluorescence intensity right after photobleaching,
- $k$ is the rate constant.

The mobile fraction and half-time of recovery ($T_{1/2}$) were calculated as:

- Mobile Fraction $= F_{\max} - F_{\min}$
- $T_{1/2}$ is time $t$ when the normalized fluorescence intensity equals to 50% (i.e., $\frac{1}{2}(F_{\max} - F_{\min})$)

## Mass spectrometry (MS) analysis

Cells were lysed in immunoprecipitation buffer containing 25 mM Tris-HCl (pH 7.5), 150 mM NaCl, 10 mM NaF, 1.5 mM MgCl$_2$, 10 mM β-glycerophosphate, 2 mM EDTA, 5 mM Na-pyrophosphate, 1 mM Na$_3$VO$_4$, 1% (v/v) Triton X-100, 0.5% NP-40, 5% Glycerol, 10 μM Olaparib, and 10 μM PDD 00017273 (PARG inhibitor), at 4 °C for 30 min and centrifuged at 21,130 × $g$ at 4 °C for 20 min. The supernatant was incubated overnight with either SQSTM1/p62 (D10E10) (#7695, CST) or Rabbit IgG control (#3990, CST) antibodies at 4 °C, followed by incubation with Protein A/G PLUS-Agarose beads (Santa Cruz Biotechnology, #SC-2003) for 3 h with rotation. The beads were washed three times with IP buffer for 10 min each, followed by a stringent wash with 300 mM NaCl for 5 min. After washing, the beads were boiled in 60 μL of an elution buffer containing 2% SDS in 50 mM Tris, pH 8, for 10 min, with gentle tapping during the process, and stored at −80 °C.

Samples were thawed and processed essentially according to the Protein Aggregate Capture protocol (Batth et al, 2019). To this end, samples were supplemented with 10 volumes of acetonitrile to precipitate all proteins, after which 25 μL (dry volume) of magnetic microspheres were added. Samples were briefly mixed, beads were allowed to settle, and afterward placed in a magnetic rack to fix the beads to the side of the tubes. Beads were subsequently washed twice with 100% acetonitrile and once with 70% ethanol, after which they were dried and resuspended in 50 μL of ice-cold 50 mM TRIS pH 8.5 buffer supplemented with 2.5 ng/μL sequencing-grade modified trypsin. The beads were mixed for 10 min on ice, after which they were moved to 30 °C overnight, shaking at 1250 RPM.

Beads were removed from the digested samples, and peptides were reduced and alkylated by adding TCEP and chloroacetamide to 5 mM, followed by incubation at 30 °C for 30 min. For desalting and purification of peptides, C18 StageTips were prepared in-house, by layering four plugs of C18 material (Sigma-Aldrich,

Empore SPE Disks, C18, 47 mm) per StageTip. Activation of StageTips was performed with 100 μL 100% methanol, followed by equilibration using 100 μL 80% acetonitrile (ACN) in 0.1% formic acid, and two washes with 100 μL 0.1% formic acid. Samples were acidified to pH <3 by addition of trifluoroacetic acid to a concentration of 1%, after which they were loaded on StageTips. Subsequently, StageTips were washed twice using 100 μL 0.1% formic acid, after which peptides were eluted using 80 μL 30% ACN in 0.1% formic acid. All samples were dried to completion using a SpeedVac at 60 °C. Dried peptides were dissolved in 20 μL 0.1% formic acid and stored at −20 °C until analysis using mass spectrometry (MS).

For sample analysis, 10% of samples (2 μL) were analyzed per injection. All samples were analyzed on a Vanquish™ Neo UHPLC system (Thermo Fisher Scientific) coupled to an Orbitrap™ Astral™ mass spectrometer (Thermo Fisher Scientific). Samples were analyzed on 20 cm long analytical columns, with an internal diameter of 75 μm, and packed in-house using ReproSil-Pur 120 C18-AQ 1.9 μm beads (Dr. Maisch). The analytical column was heated to 40 °C, and elution of peptides from the column was achieved by the application of gradients with stationary phase Buffer A (0.1% FA) and increasing amounts of mobile phase Buffer B (80% ACN in 0.1% FA). The primary analytical gradient ranged from 8%B to 40%B over 23 min, followed by a washing block of 7 min. Ionization was achieved using a NanoSpray Flex NG ion source (Thermo Fisher Scientific), with spray voltage set at 2 kV, ion transfer tube temperature to 275 °C, and RF funnel level to 50%. All full precursor (MS1) scans were acquired using the Orbitrap™ mass analyzer, while all tandem fragment (data-independent acquisition; DIA) scans were acquired in parallel using the Astral™ mass analyzer. All samples were analyzed in DIA mode, with full scan range set to 300–1000 m/z, MS1 resolution to 240,000, MS1 AGC target to "500" (5,000,000 charges), and MS1 maximum injection time to 50 ms. DIA scans were executed across a scan range of 300–1000 m/z, using 4 m/z isolation windows, and fragmentation was performed using HCD with a normalized collision energy of 25. DIA fragment scan range was set to 100–1500 m/z, DIA AGC target to "200" (20,000 charges), and DIA maximum injection time to 5 ms, resulting in a duty cycle time of ~1 s.

## MS data analysis

All RAW files were analyzed with DIA-NN v1.8.1 (Demichev et al, 2020). For library-free search, the human FASTA database downloaded from UniProt on the 29th of April 2023, and DIA-NN was used to generate a spectral library in silico. To this end, the settings used were Trypsin/P digestion (default), 1 missed cleavage (default), peptide length between 6 and 60, precursor charge state between 2 and 4, precursor m/z range from 300 to 1000 m/z, and fragment ion m/z range from 100 to 1500 m/z. RAW files were first converted to DIA format, after which they were searched with DIA-NN using the spectral library described above, with default settings. "MBR" was enabled, which generates a new spectral library based on a first search, which is then used to re-analyze the data for more robust quantification.

## MS data statistical handling

Quantification of the DIA-NN output files (gene list; "report.gg_-matrix.tsv" and precursor list; "report.pr_matrix"), as well as all

statistical testing, was performed using Perseus software (Tyanova et al, 2016). The gene list was annotated with information from the precursor list, to allow more extensive filtering. Proteins were required to be identified by at least four unique peptide sequences, of which at least one proteotypic. For quantification purposes, all protein abundance values were log2 transformed, and filtered for presence in 4 of 4 replicates (n = 4/4) in at least one experimental condition. Missing values were imputed below the global experimental detection limit at a downshift of 2.0 and a randomized width of 0.2. Statistical significances of differences were tested using two-tailed Student's t-testing, with permutation-based FDR control applied. For determining which proteins were significantly enriched by IP, proteins were required to be significantly enriched over the IgG control (FDR <0.1%; s0 set to 1) in either of the two enriched conditions. For determining differences between interactors in response to IFN treatment, the FDR cut-off was <0.1% with s0 set to 0.25.

## Statistical analysis

All statistical analyses were performed using GraphPad Prism 10 software.

For qPCR analysis, error bars represent SEM from three independent experiments. The differences in statistical significance between the two groups were tested via a two-tailed t-test, and multiple groups were tested via one- or two-way ANOVA test. All values are means of SEM of the indicated independent experiments. ns >0.05; *P < 0.05; **P < 0.01, ***P < 0.001, ****P < 0.0001.

For FRAP experiments, one-way ANOVA was performed to compare different experimental conditions, and error bars represent the standard deviation. Controls, including unbleached regions within the same sample, were used to assess photobleaching and phototoxicity effects, and experiments were repeated in multiple independent trials to ensure reproducibility.

For Western blot, co-IP, and confocal experiments, unless indicated otherwise, results are representative of at least three independent experiments.

# Data availability

Data supporting this finding are available within the main text, expanded view figures, movie, dataset, and source data. Source data are provided in this paper. The mass spectrometry proteomics data have been deposited to the ProteomeXchange Consortium via the PRIDE partner repository (http://www.ebi.ac.uk/pride) with the dataset identifier PXD057944 and referee access code SLD55LoOZFD2.

The source data of this paper are collected in the following database record: biostudies:S-SCDT-10_1038-S44318-025-00421-4.

# Peer review information

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

## Acknowledgements

We thank Drs. Phillip Sharp, Morgan Dasovich, as well as the Leung Lab members for critical comments on the manuscript. Figures 5A and 7F were created with Biorender.com. We thank Ms. Sophia Cai and Mr. Piyush Rath for their assistance with image data analyses. We thank Drs. Christopher Sullivan and Anthony Fehr for sharing the A549 PARP14 KO cells and the parental lines, Dr. Vito Rebecca for A375 melanoma cells, Dr. Ivan Ahel for the YFP-PARP14 constructs, Dr. Terje Johansen for GFP-p62 plasmid, Dr. Michael Cohen for ITK6 and ITK7, Dr. Lari Lehtiö for OUL-35, and Ribon Therapeutics, Inc. for various PARP14 inhibitors. This work is supported by Bluefield Innovation (A.K.L.L.), NIH NIGMS grant R35GM142837 (D.C.), and Novo Nordisk Foundation NNF14CC0001 (I.A.H and S.C.B-L)

## Author contributions

**Rameez Raja**: Conceptualization; Resources; Formal analysis; Supervision; Validation; Investigation; Visualization; Writing—original draft; Writing—review and editing. **Banhi Biswas**: Formal analysis; Validation; Investigation; Visualization; Writing—review and editing. **Rachy Abraham**: Formal analysis; Supervision; Validation; Investigation; Visualization; Project administration; Writing—review and editing. **Yiran Wang**: Investigation; Visualization; Writing—review and editing. **Che-Yuan Chang**: Formal analysis; Visualization. **Ivo A Hendriks**: Funding acquisition; Investigation; Writing—review and editing. **Sara C Buch-Larsen**: Funding acquisition; Investigation; Writing—review and editing. **Hongrui Liu**: Formal analysis; Investigation; Visualization; Writing—review and editing. **Xingyi Yang**: Formal analysis. **Chenyao Wang**: Formal analysis. **Hien Vu**: Validation; Writing—review and editing. **Anne Hamacher-Brady**: Supervision; Writing—review and editing. **Danfeng Cai**: Supervision; Funding acquisition; Writing—review and editing. **Anthony K L Leung**: Conceptualization; Supervision; Funding acquisition; Visualization; Writing—original draft; Project administration; Writing—review and editing.

Source data underlying figure panels in this paper may have individual authorship assigned. Where available, figure panel/source data authorship is listed in the following database record: biostudies:S-SCDT-10_1038-S44318-025-00421-4.

## Disclosure and competing interests statement

The authors declare no competing interests.

# Expanded View Figures

**Figure EV1.  Interferon-induced upregulation of PARP14 is critical for cytoplasmic ADPr condensate formation.**

(A) ADPr condensates were monitored after cells treated with or without IFNα, β, γ (500 IU/ml or 1000 IU/ml) for 24 h, $n = 52$–112 cells. (B) ADPr condensate formation was monitored after cells were treated with IFNγ for the indicated time points and then replaced with fresh medium without IFNγ for a total duration of 24 h, $n = 339$–578 cells. (C) RT-qPCR analyses of all human PARP expression in cells upon treatment with or without IFNγ for 6 and 24 h. (D) PARP14 mRNA was measured by RT-qPCR in A549 cells pretreated with Actinomycin D (ActD 0.5 μg/ml) for 1 h, followed by 6-h treatment with or without IFNγ. (E) PARP9, PARP12, and PARP14 mRNA levels were measured by RT-qPCR in A549 cells treated with or without IFNγ for different time periods, followed by growing cells in media without IFNγ for a total of 24 h (same as panel S1B). Lower panel showing PARP14 protein levels measured by western blot. (F) PARP14 mRNA was measured by RT-qPCR after 48 h of siRNA transfection. (G) PARP14 protein levels were measured in A549 cells after 48 h of siPARP14 transfection. (H) PARP14 protein levels were assessed in cells transfected with si-PARP14 or control siRNA, with or without IFNγ for 24 h. (I) ADPr condensates were analyzed in either wild-type or si-PARP14 transfected cells with or without IFNγ treatment for 24 h, $n = 434$-547 cells. (J) PARP14 protein levels were measured in A549 stably transduced with shRNA against PARP14 using lentivirus. (K) ADPr condensates were analyzed in either wild-type or shPARP14 knockdown cells, after 24-h treatment with or without IFNγ, $n = 375$–449 cells. (L) PARP9 protein levels were measured in A549 wild-type (WT) or knockdown cells (KD) stably transduced with shRNA against PARP9 via lentivirus. (M) ADPr condensates were analyzed in either WT or shPARP9 KD cells, after 24-h treatment with or without IFNγ, $n = 131$–467 cells. (N) PARP12 protein levels were measured in A549 WT or KD cells stably transduced with shRNA against PARP12. (O) ADPr condensates were analyzed in either wild-type or shPARP12 knockdown cells after 24-h treatment with or without IFNγ, $n = 225$–267 cells. White arrows indicate the position of condensates, and white boxes represent the zoomed-in regions. The total area of condensates and Pearson correlation coefficient were analyzed by CellProfiler. Mean ± SEM, ns not significant, $*p < 0.05$, $**p < 0.01$, $***p < 0.001$, $****P < 0.0001$, unless otherwise stated. (A, B, F) One-way ANOVA, (C–E, I, K, M, O) Two-way ANOVA. All $p$ values are provided in Dataset EV2. Data were representative of three biological replicates. Scale bar, 10 μm. Source data are available online for this figure.

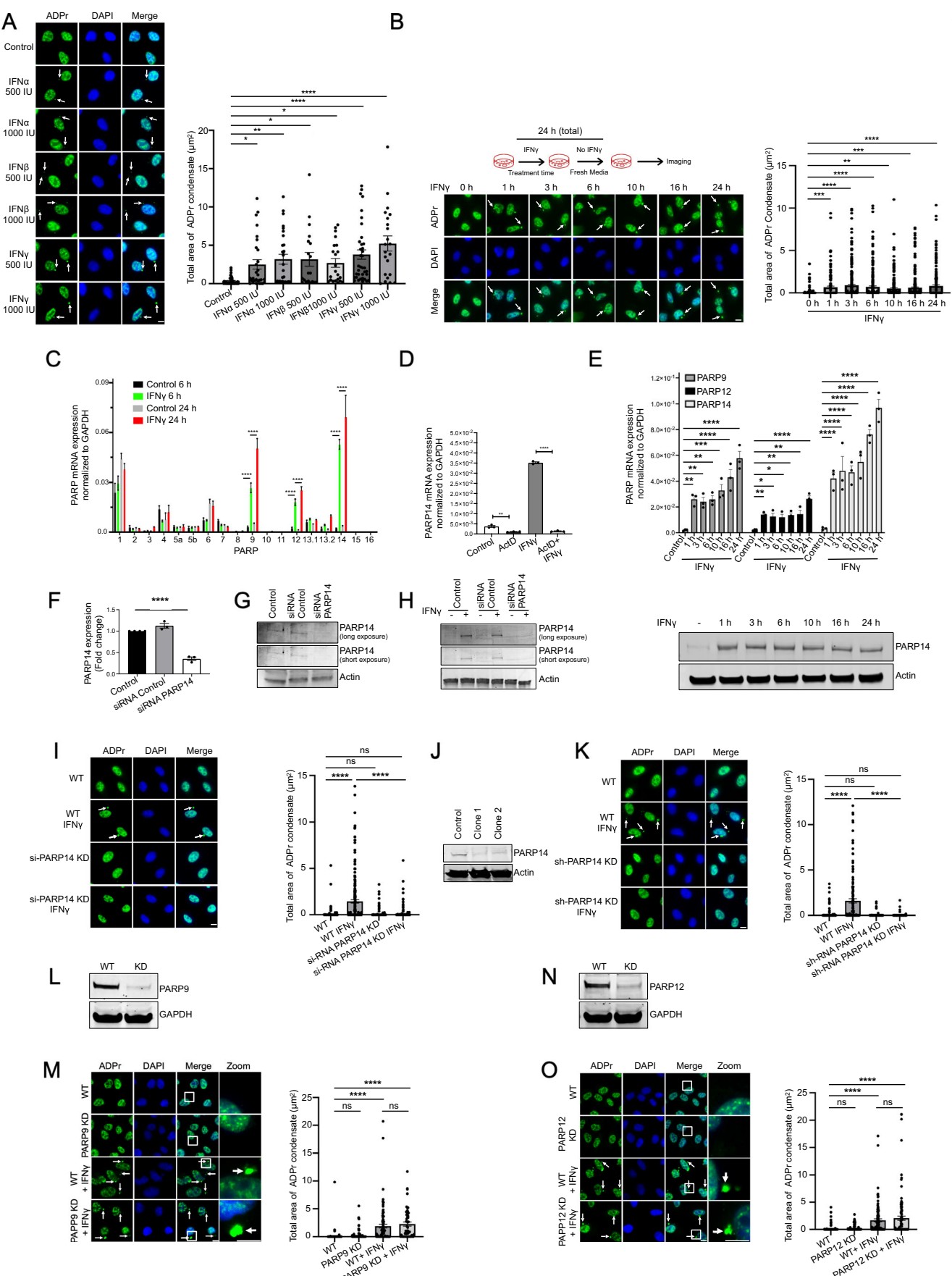

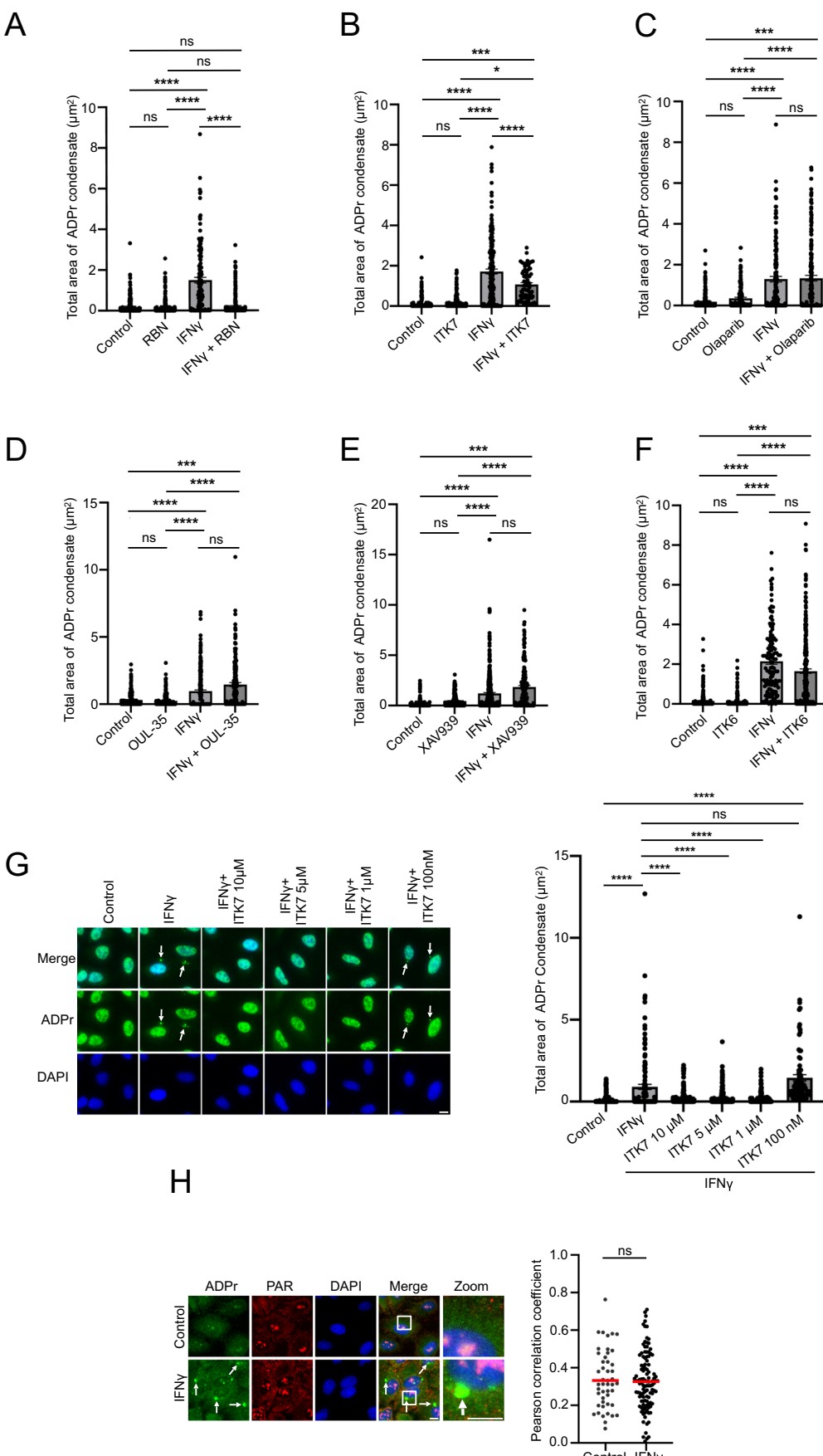

**Figure EV2.  Cytoplasmic ADPr condensate formation depends on PARP14 catalytic activity.**

Quantification of ADPr condensates in A549 cells pretreated with different PARP inhibitors for 1 h prior to 24-h treatment with or without IFNγ: (**A**) 10 μM RBN, $n = 236–607$ cells. (**B**) 10 μM ITK7, $n = 166–506$ cells. (**C**) 10 μM Olaparib, $n = 305–485$ cells. (**D**) 3 μM OUL-35, $n = 450–596$ cells. (**E**) 10 μM XAV939, $n = 460–676$ cells, and (**F**) 10 μM ITK6, $n = 301–593$ cells. (**G**) ADPr condensate formation was analyzed in cells pretreated with different doses of ITK7 for 1 h prior to 24-h IFNγ treatment, and compared to untreated control, $n = 163–1049$ cells. (**H**) PAR (Enzo life sciences, BML-SA216) and ADPr (Millipore, MABE1016) colocalization was assessed after 24-h treatment with or without IFNγ, $n = 48–112$ condensates. White arrows indicate the position of condensates, and white boxes represent the zoomed-in regions. The total area of condensates and Pearson correlation coefficient were analyzed by CellProfiler. Mean ± SEM, ns not significant, $*p < 0.05$, $**p < 0.01$, $***p < 0.001$, $****p < 0.0001$, unless otherwise stated. (**A–G**) Two-way ANOVA; (**H**) Unpaired *t*-test. All *p* values are provided in Dataset EV2. Data were representative of three biological replicates. Scale bar, 10 μm. Source data are available online for this figure.

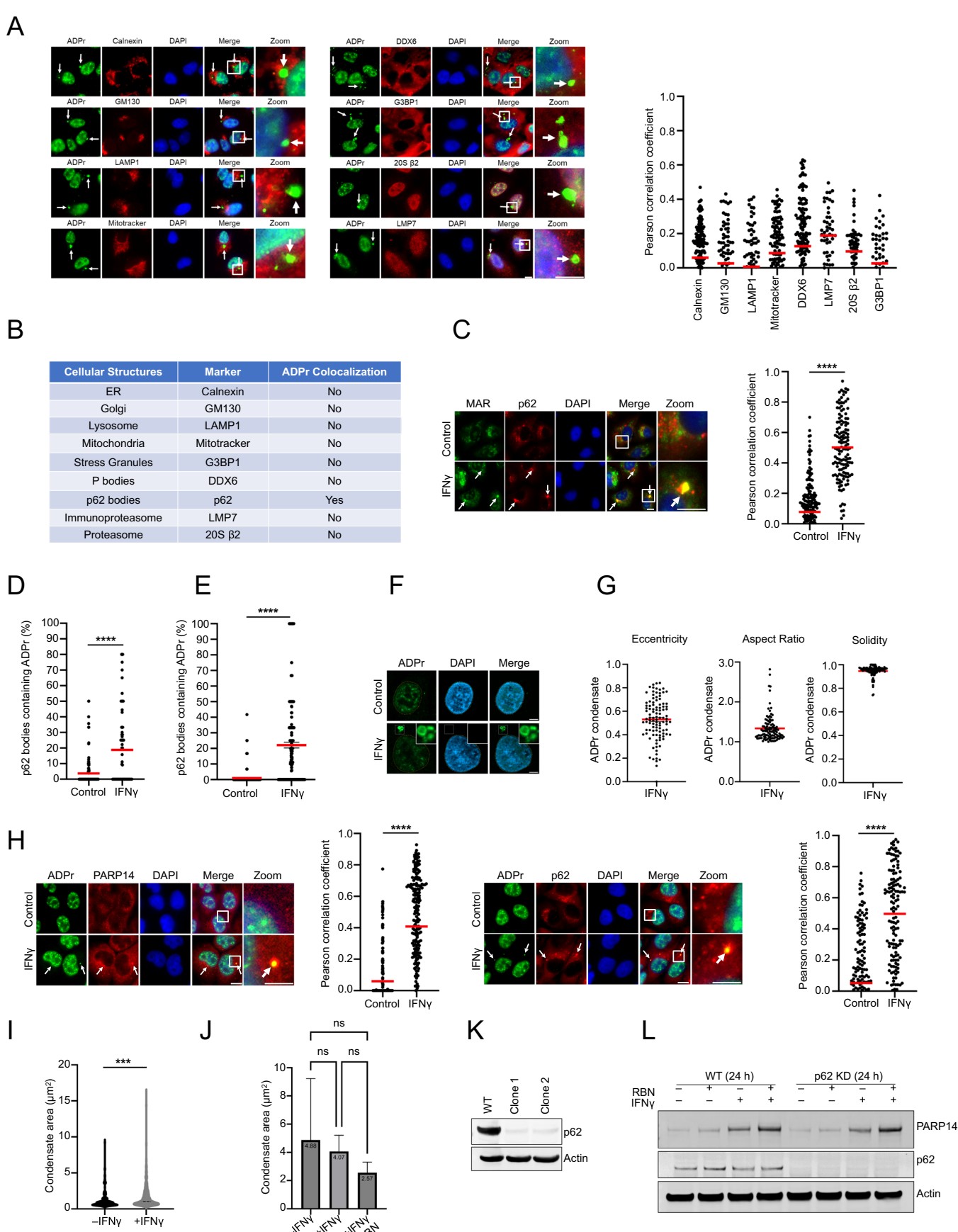

**Figure EV3. ADPr condensates colocalize with p62 and exhibit distinct morphometric properties upon IFNγ treatment.**

(A) A549 cells were treated with IFNγ for 24 h and stained for colocalization of different markers of cellular structures (red) and ADPr (green), $n = 52$–198 condensates. The colocalization of ADPr condensates with the cellular structures was calculated by CellProfiler software using the Pearson correlation coefficient. (B) Table summarizing the colocalization of different cellular structure markers with ADPr condensates in A549 cells. (C) Colocalization of p62 with MAR was analyzed in cells with or without 24-h IFNγ treatment, $n = 153$–198 condensates. (D) Quantification showing the percentage of p62 bodies positive for ADPr for Fig. 3A, obtained by using a Leica microscope, $n = 70$–112 cells. (E) Quantification showing the percentage of p62 bodies positive for ADPr for Fig. 3D, obtained by using Zeiss Airyscan detector, $n = 250$ cells. (F, G) Representative immunofluorescence images using an Airyscan detector (Zeiss). Cells were imaged after 24-h IFNγ treatment: Morphometry analysis of ADPr condensates was performed by calculating eccentricity, aspect ratio, and solidity measured using the CellProfiler size/shape measurement module. Aspect ratio calculated by the ratio between major axis length and minor axis length of each region of interest, $n = 94$–107 condensates. Scale bars, 5 μm. (H) A375 cells were treated with or without IFNγ for 24 h and analyzed for colocalization of ADPr with PARP14 (left panel, $n = 179$–188 condensates) or p62 (right panel, $n = 148$–182 condensates). (I) Violin plot showing size quantification of p62 bodies in A549 cells treated with or without IFNγ for 24 h by using imageJ "Max Entropy" for setting threshold and identifying condensates with size greater than 1 pixel using the "Analyze Particle, $n = 182$ condensates (–IFNγ) versus 467 condensates (+IFNγ) from three different fields in each group. (J) Size comparison of p62 bodies in FRAP analyses under different conditions for Fig. 3H,I. (K) p62 protein levels were measured in shp62 knockdown (KD) A549 cell clones generated by lentiviral transduction. (L) PARP14 and p62 protein levels in wild-type (WT) and p62 KD cells pretreated with RBN (10 μM) for 1 h followed by 24-h treatment with or without IFNγ. White arrows indicate the position of condensates, and white boxes represent the zoomed-in regions. The total area of condensates and Pearson correlation coefficient were analyzed by CellProfiler. Mean ± SEM, ns not significant, *$p < 0.05$, **$p < 0.01$, ***$p < 0.001$, ****$p < 0.0001$, unless otherwise stated. (C–E, H–I) Unpaired $t$-test; (J) One-way ANOVA. All $p$ values are provided in Dataset EV2. Data were representative of three biological replicates. Scale bar, 10 μm, unless otherwise indicated. Source data are available online for this figure.

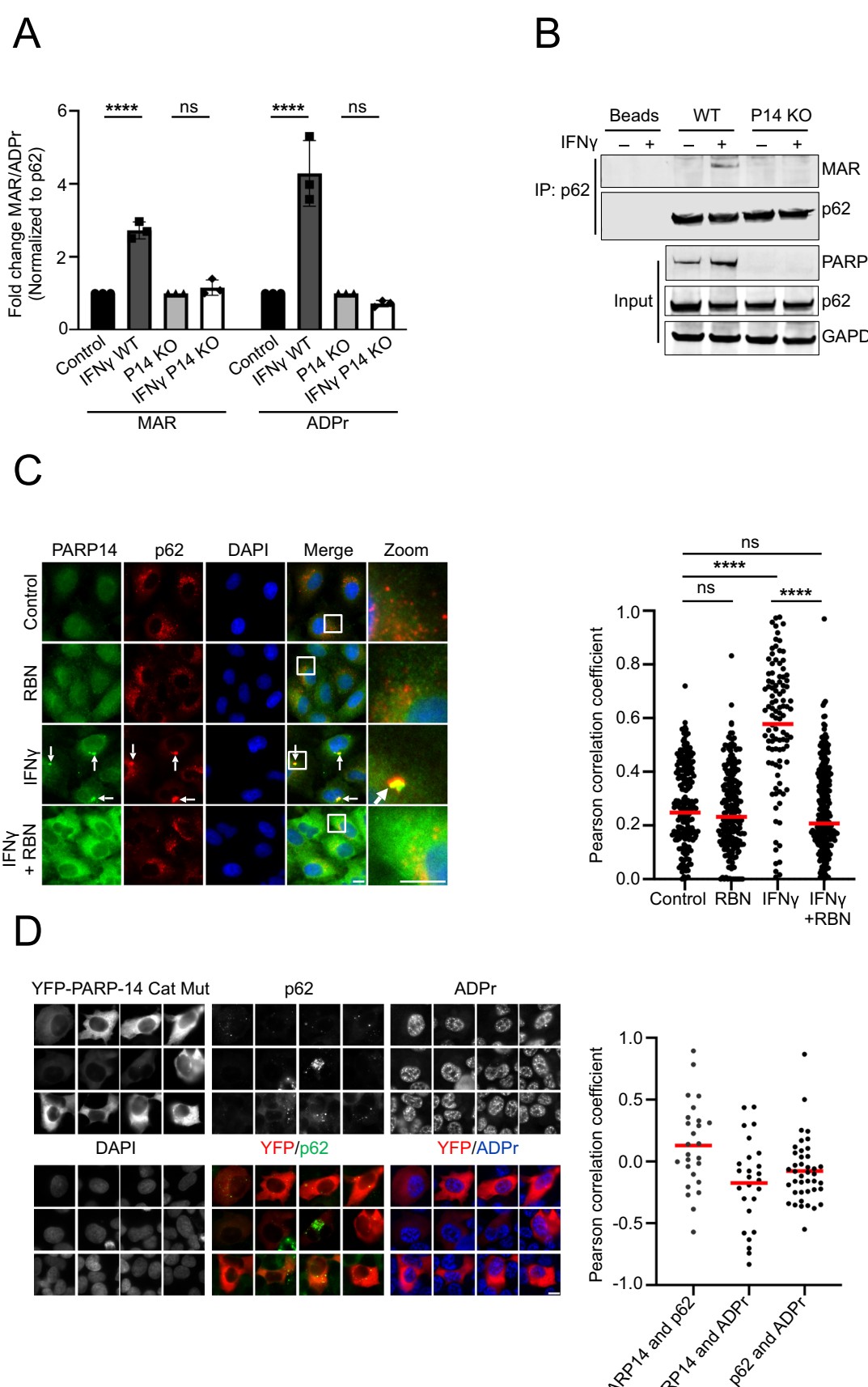

◄  **Figure EV4.  PARP14-mediated p62 co-condensation depends on PARP14 catalytic activity.**

(A) Quantification of MAR and ADPr signal on p62 immunoprecipitates, as shown in Fig. 4D, from wild-type (WT) and PARP14 knockout (KO) A549 cells after 24-h treatment with or without IFNγ. (B) Immunoprecipitates of p62 performed under denaturing conditions were probed for MAR (HCA-355) and p62 from WT and PARP14 KO A549 cells after 24-h treatment with or without IFNγ. (C) PARP14 (Abcam) and p62 colocalization was assessed in cells pretreated with or without RBN for 1 h, followed by 24-h treatment with or without IFNγ, $n = 106$–321 condensates. (D) U2OS cells were transiently transfected with YFP-PARP14 catalytic mutant and analyzed for ADPr condensate formation and PARP14 and p62 colocalization, $n = 27$–44 condensates. White arrows indicate the position of condensates, and white boxes represent the zoomed-in regions. The total area of condensates and Pearson correlation coefficient were analyzed by CellProfiler. Mean ± SEM, ns not significant, $*p < 0.05$, $**p < 0.01$, $***p < 0.001$, $****p < 0.0001$, unless otherwise stated. (A, C) Two-way ANOVA. All $p$ values are provided in Dataset EV2. Data were representative of three biological replicates. Scale bar, 10 μm. Source data are available online for this figure.

    

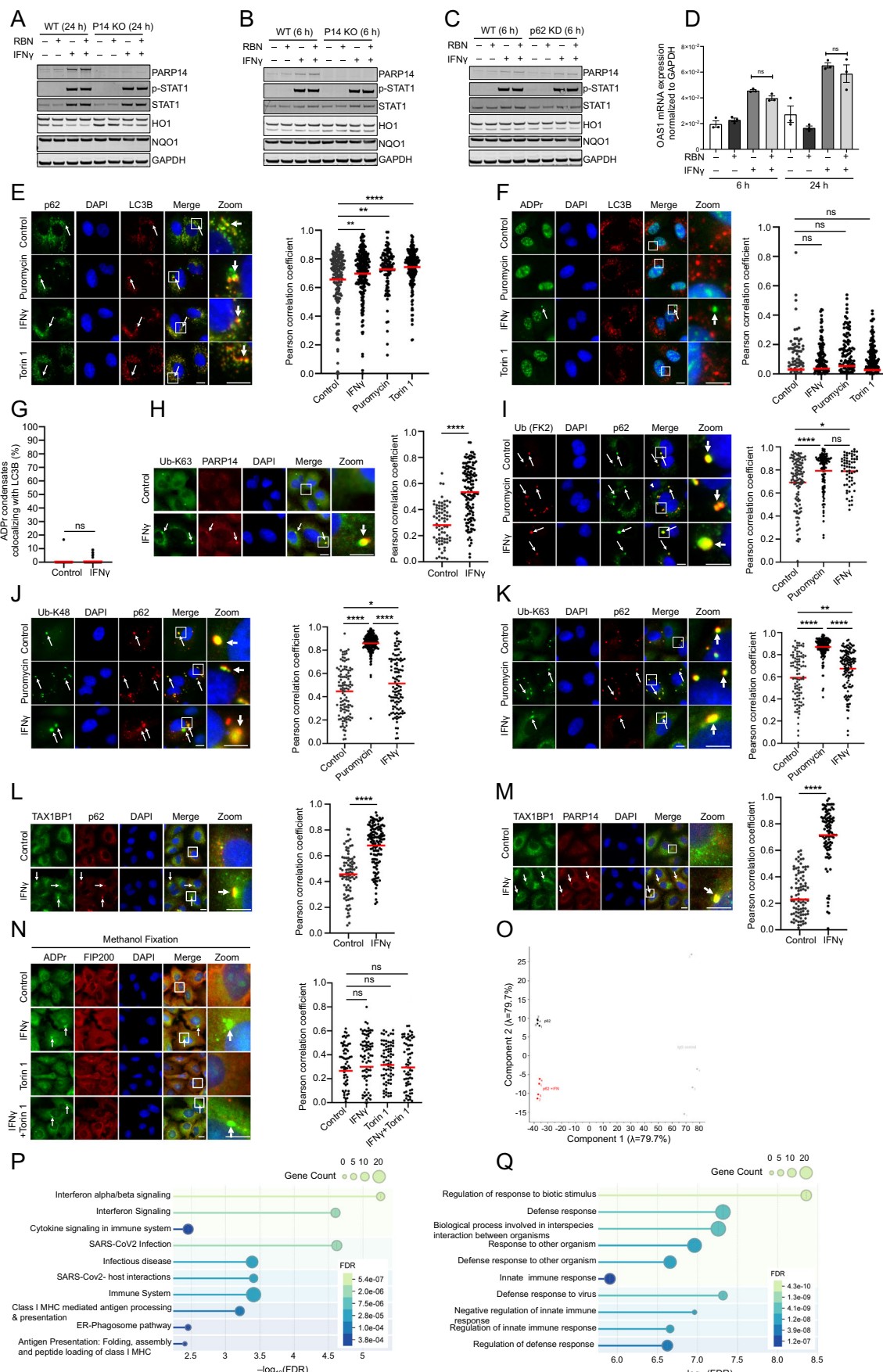

◀

**Figure EV5.  ADPr-enriched p62 bodies selectively contain components of canonical p62 bodies.**

(A, B) Western blot analyses of p-STAT1, STAT1, HO1, and NQO1 in A549 wild-type (WT) and PARP14 knockout (KO) cells treated with different combinations of 1-h RBN (10 µM) pretreatment and 6-h or 24-h IFNγ treatment. (C) Western blot analyses of p-STAT1, STAT1, HO1, and NQO1 in wild-type (WT) and p62 knockdown (KD) cells treated with different combinations of 1-h RBN (10 µM) pretreatment and 6-h IFNγ treatment. (D) RT-qPCR analyses of OAS1 mRNA levels in cells treated with different combinations of RBN (10 µM) pretreatment for 1 h, followed by IFNγ treatment for either 6 or 24 h. (E, F) Colocalization of LC3B with (E) p62, $n = 105$–221 condensates, and (F) ADPr, $n = 105$–165 condensates, was assessed in cells either treated with puromycin (5 µg/ml) for 3 h, IFNγ overnight or Torin 1 (1 µM) for 3 h, and compared to untreated control. Puromycin and Torin 1 treatment was given for 3 h before fixing cells. (G) Quantification showing the percentage of LC3B positive for ADPr for Fig. 5D, $n = 81$–87 condensates. (H) Colocalization of PARP14 and Ub-K63 was assessed in cells treated with or without IFNγ for 24 h, $n = 74$–135 condensates. (I–K) Colocalization of p62 with (I) Ubiquitin, $n = 64$–124 condensates, (J) Ub-K48, $n = 113$–210 condensates, and (K) Ub-K63, $n = 96$–177 condensates, was assessed in cells treated with or without IFNγ or puromycin. IFNγ was given overnight while puromycin (5 µg/ml) treatment was given for 3 h before fixation. (L) Colocalization of TAX1BP1 and p62 was assessed after 24-h treatment with or without IFNγ, same as Fig. 5H, 86–131 condensates. (M) Colocalization of TAX1BP1 and PARP14 was assessed after 24-h treatment with or without IFNγ, $n = 98$–101 condensates. (N) Colocalization of FIP200 and ADPr was assessed after treating A549 cells with IFNγ overnight, followed by either DMSO or Torin 1 (1 µM) treatment for 6-h using ice-cold methanol for fixation, $n = 55$–108 condensates. (O) Principal component analysis (PCA) plot of immunoprecipitated samples across four biological replicates: IgG Control (gray), p62 alone (black), and p62 in the presence of IFNγ (red). (P, Q) Term enrichment analysis showing (P) Reactome pathway, and (Q) gene ontology statistically enriched within the p62-specific network after IFNγ treatment. Statistical testing was conducted using the STRING database with default settings. White arrows indicate the position of condensates, and white boxes represent the zoomed-in regions. The total area of condensates and Pearson correlation coefficient were analyzed by CellProfiler. Mean ± SEM, ns not significant, $*p < 0.05$, $**p < 0.01$, $***p < 0.001$, $****p < 0.0001$, unless otherwise stated. (D) One-way ANOVA; (E, F, I–K, N) Two-way ANOVA; (G, H, L, M) Unpaired $t$-test. All $p$ values are provided in Dataset EV2. Data were representative of three biological replicates. Scale bar, 10 µm. Source data are available online for this figure.

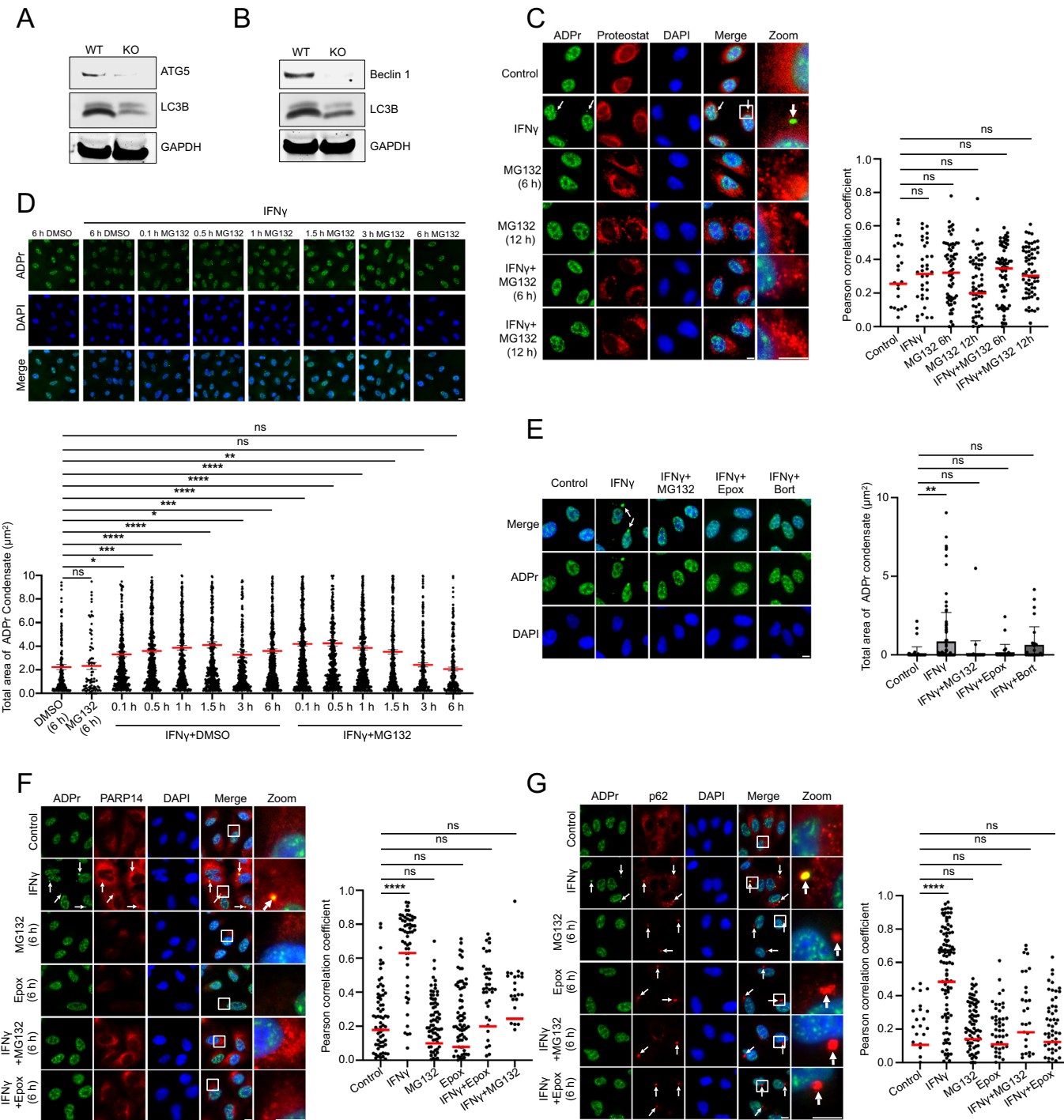

◀ **Figure EV6.  Cytoplasmic ADPr condensates do not colocalize with aggresomes but depend on proteasome activity.**

(A) Western blot showing ATG5 and LC3B protein levels in ATG5 knockout (KO) cells. (B) Western blot showing Beclin 1 and LC3B protein levels in Beclin 1 KO cells. (C) Colocalization of ADPr and aggresomes stained with PROTEOSTAT® dye was assessed in A549 cells treated with or without IFNγ overnight, followed by 10 μM MG132 treatment for either 6 or 12 h, $n = 26$–71 condensates. (D) ADPr condensates were monitored after cells were treated with IFNγ for 24 h, followed by 10 μM MG132 for different time periods as indicated, and compared to untreated control. The lower panel shows quantification of ADPr condensates in IFNγ-treated cells either in the presence of DMSO or 10 μM MG132, $n = 85$–535 cells. (E) ADPr condensate formation was assessed in cells treated with IFNγ overnight, followed by treatment with proteasome inhibitors 10 μM MG132 or 1 μM Epoxomicin (Epox) or Bortezomib (100 nM) for 6 h, $n = 88$–227 cells. (F) Colocalization of ADPr and PARP14 was analyzed in cells treated with IFNγ overnight, followed by the treatment with 10 μM MG132 or 1 μM Epoxomicin for 6 h, same as in Fig. 6D, and compared to untreated control, $n = 33$–106 condensates. (G) Colocalization of ADPr and p62 was analyzed in cells treated with MG132 or Epoxomicin, as in panel (F), $n = 33$–109 condensates. White arrows indicate the position of condensates, and white boxes represent the zoomed-in regions. The total area of condensates and Pearson correlation coefficient were analyzed by CellProfiler. Mean ± SEM, ns not significant, $*p < 0.05$, $**p < 0.01$, $***p < 0.001$, $****p < 0.0001$, unless otherwise stated. (C–G) One-way ANOVA. All $p$ values are provided in Dataset EV2. Data were representative of three biological replicates. Scale bar, 10 μm. Source data are available online for this figure.

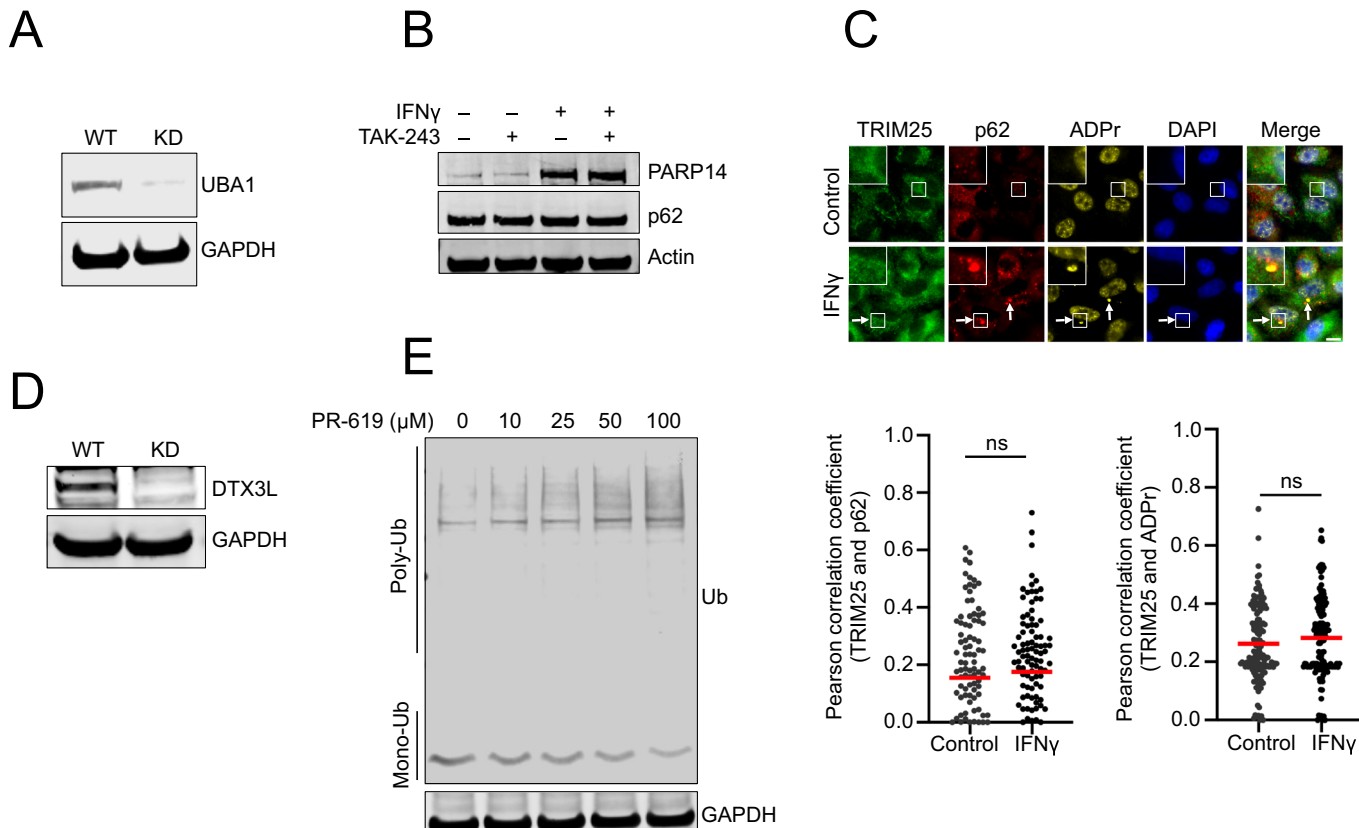

**Figure EV7. Cytoplasmic ADPr condensate formation depends on specific components of the ubiquitination pathway.**

(A) Western blot showing UBA1 protein levels in A549 wild-type (WT) and lentiviral shRNA-UBA1 knockdown (KD) cells. (B) PARP14 and p62 protein levels were analyzed from samples in Fig. 7B. (C) Colocalization of TRIM25 with p62 and ADPr was analyzed in A549 cells with or without 24-h IFNγ treatment, $n = 106–108$ condensates for the p62 panel and $n = 150$ condensates for the ADPr panel. (D) Western blot showing DTX3L protein levels in A549 WT and lentiviral shRNA-DTX3L KD cells. (E) The total ubiquitination profile was assessed in A549 cells treated with different doses of PR619 for 8 h. White arrows indicate the position of condensates, and white boxes represent the zoomed-in regions. The total area of condensates and Pearson correlation coefficient were analyzed by CellProfiler. Mean ± SEM, ns not significant. (C) Unpaired t-test. All p values are provided in Dataset EV2. Data were representative of three biological replicates. Scale bar, 10 μm. Source data are available online for this figure.

