## [Peer Review File · The EMBO Journal]

Interferon-Induced PARP14-Mediated ADP-Ribosylation in p62 Bodies Requires the Ubiquitin-Proteasome System

Rameez Raja, Banhi Biswas, Rachy Abraham, Yiran Wang, Che-Yuan Chang, Ivo Hendriks, Sara Larsen, Hongrui Liu, Xingyi Yang, Chenyao Wang, Hien Vu, Anne Hamacher-Brady, Danfeng Cai, and Anthony Leung

Corresponding author(s): Anthony Leung (anthony.leung@jhu.edu)

Review Timeline:

Transfer from Review Commons:	28th Jul 24
Editorial Decision:	17th Sep 24
Revision Received:	8th Dec 24
Editorial Decision:	14th Feb 25
Revision Received:	11th Mar 25
Accepted:	13th Mar 25

Editor: Hartmut Vodermaier

Transaction Report:

Review #1

1. Evidence, reproducibility and clarity:

Evidence, reproducibility and clarity (Required)

Summary

In this study, Raja et al. found cytoplasmic condensates formed by the treatment of INF γ , investigated components of these condensates and identified p62, NBR1 and PARP14 as their components. INF γ treatment induced PARP14 expression, and PARP14 inhibitor treatment inhibited condensation formation, suggesting that the amount of PARP14 and its enzymatic activity are important for the condensate formation. The ADPr-positive p62 condensates were independent of autophagic degradation, and proteasomal activity was required for their formation.

Major comment

1. The finding that the ubiquitin-proteasome, but not autophagy activity, is indispensable for the formation of p62 condensates is of interest. However, the molecular mechanism by which the ubiquitin-proteasome system (UPS) is involved in the regulation of the PARP14-p62 condensate is still unclear. Which step(s) is the UPS involved?
2. The p62 condensate serves as a scaffold for autophagosome formation through the assembling autophagy receptors including NBR1 and TAX1BP1, followed by recruiting ATG proteins such as FIP200. While ADPr-positive p62 condensates also contain NBR1 and polyubiquitinated proteins, they are unrelated to autophagic degradation. It is unclear what factors govern autophagy-independent function.
3. The authors claim that the amount of PARP14 and its MAR activity are essential for the condensate formation. However, all experiments were performed only with PARP14 inhibitors, and further validation is needed. If the importance of PARP14 activity is to be directly demonstrated, experiments in which an enzyme activity mutant is introduced into PARP14 KO cells are needed.
4. In Figure 2a, the heatmap alone is insufficient. Neither errors nor statistical comparisons are indicated.
5. The statistical analysis of Figure S2 is inappropriate; instead of t-tests, multiple comparisons should be used to compare three or more groups.

Minor comment

1. What percentage of p62 condensates upon INF γ treatment are ADPr positive? Are all p62 bodies seen with INF γ stimulation unrelated to autophagy?
 2. Is ADPr condensation a PARP14-specific phenomenon? PARP9 and PARP12 were also upregulated by INF γ treatment. Are these factors also involved in condensate formation?
 3. Figure 4D appears to be immunoprecipitation (IP) under non-denaturing conditions. If so, it is not possible to distinguish whether the MAR signal is derived from p62 or from the p62 interacting proteins (the associated ubiquitinated substrates). IP experiments should be performed under denaturing conditions.
 4. In Figure 5B, which band is HO1, the upper or lower?
 5. There is no image for ubiquitin in S5D.
- Right panel in Figure 4F shows only IF γ + RBN, which should show all data sets in the same panel.

2. Significance:

Significance (Required)

Liquid droplets, which have continuously being identified in cells, are a hot topic in cell biology. Droplet formation, structure, molecular dynamics, and degradation, as well as their abnormalities and disease development due to genetic mutation and stress, are of wide-ranging interest from basic to pathological aspects. Therefore, this research has the potential to attract interest from a wide range of fields.

General assessment

Overall, the data are clear and the phenomenon is of interest. However, the molecular mechanism and biological significance of the condensate formation is unknown; It is unclear why proteasome activity is required for the formation of PARP14-mediated ADP ribosylation. It is also unclear what the consequences are for the cell if the ADPr-positive condensates are not formed. The authors should address these general and important issues and provide the data if not all.

3. How much time do you estimate the authors will need to complete the suggested revisions:

Estimated time to Complete Revisions (Required)

(Decision Recommendation)

Between 3 and 6 months

4. Review Commons values the work of reviewers and encourages them to get credit for their work. Select 'Yes' below to register your reviewing activity at Web of Science Reviewer Recognition Service (formerly Publons); note that the content of your review will not be visible on Web of Science.

Yes

Review #2

1. Evidence, reproducibility and clarity:

Evidence, reproducibility and clarity (Required)

This manuscript investigates the formation of a novel cellular structure or condensate, similar to p62 bodies, that includes PARP14 and p62. The interferon-induced PARP14-mediated ADP-ribosylation of p62 in these condensates depends on an active ubiquitin-proteasome system. These condensates are characterized by the presence of PARP14 and ADPr and include some, but not all, components of the p62 bodies. Furthermore, their formation depends on both ubiquitin activation and proteasome activity, but it is unaffected by autophagy inhibition, unlike conventional p62 bodies.

The Introduction provides a well-delineated context of condensates, highlighting the importance of post-translational modifications in responding to environmental changes.

Although the manuscript is well-organized with apparent logical development, there are weaknesses that diminish the impact of the reported data. A more accurate review of the structures, both from a morphological and quantitative perspective, would strengthen the conclusions and the overall impact of this work. Additionally, while the authors have analyzed the contribution of PARP14 to condensate formation, the biological significance of these structures remains unclear. For instance, performing MS (mass spectrometry) analysis on the described structures could help identify their composition and functions.

The methodologies used in this study are standard for molecular and cellular biology research, including immunofluorescence assays, transient transfections, immunoprecipitations, and fluorescence recovery after photobleaching (FRAP) assays. These methods are described in detail and can be reproduced.

Below, please find a list of comments and suggestions to enhance the robustness of the data:

****Major Points****

1. The IF analyses are central to the conclusions reported and are employed for each of the inhibitors or other tools used to investigate the formation of these condensates. The quality of the IF images needs to be improved; the shape and contacts of the condensates should be analyzed using either super-resolution or EM microscopy, or preferably both. The lack of morphometry and quantification from cell populations needs to be addressed for all experiments. These analyses are needed to support the claim that the condensates presented in this study are indeed novel structures, rather than being transient aggregates of a different nature.
2. The claim that PARP14 is essential for the formation of condensates requires support by the analyses indicated above. Minor points regarding Fig. 1 are indicated below. I suggest performing KD of PARP9 and/or PARP12 (whose expression is increased upon IFN treatment) and checking ADPr condensates to validate the central role of PARP14.
3. According to the text, "PARP14 was pulled down by ADP-ribose binding Af1521 macrodomain following IFN γ treatment (Fig. 2H)", but the legend to the figure says otherwise. A Pan-ADPr binding reagent (MABE1016) is reported in the figure. Although the conclusion is similar for the results obtained with these two tools (but they must be described and reported properly), it is still insufficient to claim that PARP14 is ADP-ribosylated. This point should be at least discussed.
4. I have difficulties analyzing the colocalization with the different organelles, even enlarging the images as much as possible. In most cases, only one condensate per image is shown. Continuities with the nuclear envelope appear in some cases: has this been investigated?

****Minor Points****

1. Fig. 1A: The DAPI images at 3 and 6 hours are reversed. Additionally, for Fig. S1a and Fig. 1A, please include quantifications.
2. Fig. 1B: Check PARP14 levels (and other IFN-PARPs) under the same experimental conditions.
3. Fig. 1H: Explain why PARP14 IF staining is still visible upon RBN012811 treatment, while it is completely lost in WB analysis or upon PARP14 siRNA treatment (Fig. 1G). In addition,

please include IF quantifications.

4. Fig. 2C: Please include quantifications.

5. Fig. 2E: The RBN treatment time is not indicated. Please include this information in the figure legend.

6. Fig. 2G: I am not convinced about the PARP14 staining. IF images do not show an increase in PARP14 levels, while WB analysis shows a strong increase in PARP14 protein levels (see Fig. 2E). Moreover, the RBN treatment time was not indicated; please include it in the figure legend. Does RBN alone affect PARP14 localization? The reported picture shows only 2 cells, each with a different subcellular localization of PARP14. As previously suggested, quantifications are required.

7. Fig. 3B: Pearson's correlation coefficient (PCC) is reported for $n=3$. The images show one condensate per cell. Under these conditions, the number of cells analyzed should be at least 100 for each experiment. Additionally, the PCC between PARP14 and p62 at steady state is shown to be 60% (which is quite high). However, the IF pictures do not support this quantification. Can the authors provide higher-resolution pictures? Does PARP14 always co-localize with p62? Lines 207-208 state: "these findings suggest that PARP14 is localized to p62 bodies upon IFN γ treatment when ADP-ribosylation occurs." According to the PCC value, the two proteins co-localize even in the absence of IFN. Can the authors clarify this aspect?

8. Fig. S3B: Please include quantifications.

9. Fig. S3C: How was the condensate size quantified? It would be useful to show a quantification mask.

10. Fig. 3D: Does p62 KD affect PARP14 localization? The reported picture shows only 2 cells, each with different staining of PARP14.

11. Fig. 4A: Please quantify the PARP14 co-IP signal with p62, normalized to PARP14 total levels.

In the reported WB, it is difficult to see the interaction between PARP14 and p62 in untreated conditions. Please provide clearer WB.

Additionally, I would expect an increased interaction between PARP14 and p62 upon IFN treatment due to PARP14 recruitment to p62 condensates, not just because of increased PARP14 levels. Since the authors show that PARP14 is not recruited to ADPr condensates upon RBN treatment (Fig. 2G), why is the interaction between p62 and PARP14 so high under RBN treatment?

12. Fig. 4C: Please quantify WB signals of ADP-ribosylated p62 for the different conditions analyzed. ADP-ribosylation of p62 is still present in cells lacking PARP14. Are there other enzymes that can modify p62? Moreover, the authors state: "We observed an increased MARYlation of p62 upon IFN γ treatment" (line 230); is this dependent on the increase in PARP14 levels or the translocation of PARP14 to ADPr condensates? Quantifications

should help clarify this aspect.

13. Fig. 4G: Quantifications are required.

2. Significance:

Significance (Required)

The role of the PARP family in cellular processes is a very active and rapidly growing field. New information about the organization of PARPs in the nucleus, cytosol, or different types of bodies/structures is certainly relevant to the field.

However, the present study is too preliminary at the moment to be considered highly relevant. Both the data analysis and conclusions need to be carefully reviewed. After major revisions, the manuscript might be of general interest if well contextualized within the fields of post-translational modification and protein degradation processes. It would remain in any case interesting for the field of ADP ribosylation.

3. How much time do you estimate the authors will need to complete the suggested revisions:

Estimated time to Complete Revisions (Required)

(Decision Recommendation)

Between 3 and 6 months

No

Review #3

1. Evidence, reproducibility and clarity:

Evidence, reproducibility and clarity (Required)

In this manuscript, titled "Interferon-Induced PARP14-Mediated ADP-Ribosylation in p62 Bodies Requires an Active Ubiquitin-Proteasome System", Raja et al. perform fluorescence microscopy assays and molecular analyses on cultured cells *ex vivo* to further our understanding of ADP-ribose accumulations that form in the cytoplasm in response to IF γ stimulation. Guided by the operon-like linkage and co-expression patterns of PARP14, PARP9, and DTX3L and recent reports describing a SARS-CoV-2-dissolved cytoplasmic body induced by interferon-induced that is rich in ADP-ribose and the PARP9/DTX3L heterodimer form following IF γ stimulation, the authors provide clarity in this manuscript regarding two knowledge gaps - (i) what catalyzes mono(ADP-ribosyl)ation within these structures (as PARP9 lacks ADP-ribosyltransferase activity) and (ii) how these foci/condensates relate to similarly composed autophagy-associate "p62 bodies" that have been previously described. Using a combination of genetic depletion and inhibitor-based approaches, the authors show that these ADPr-condensates rely on the catalytic activity of PARP14. The authors also show that while these ADPr-condensates share componentry with "p62 bodies" like ubiquitin and p62 itself, these foci are distinct accumulations, as they lack both LC3B and sensitivity to autophagy inhibitors and require an active ubiquitin-mediated proteosomal degradation system.

While this report represents a more incremental advance in our understanding of these cell signaling structures, especially considering a pair of very recently published and similar reports (Kar et al., EMBO J [2024] and Chaves Ribiero et al EMBO J [2024]), the work here is well-written and reasoned and complements these works with some novelty and distinction to that reported literature. The experiments are definitive and of high quality and the authors interpretations/conclusions are largely well-supported by the results. Thus, it is my opinion that this work is appropriate for publication with predominantly minor revisions (outlined below) and a few more substantive experimental additions.

****Major Comment:****

A major claim and novelty reported here is that the ADPr condensates are distinct from "p62 bodies". The evidence to support this rely largely on differences in their sensitivity to pharmacological treatments as well as somewhat subtle differences in FRAP recovery in p62 condensates after IFN γ treatment. But, this claim would be better supported with more comprehensive mapping of differences in the componentry or functional outcomes of these condensates. The authors might consider:

- Mass spectrometry against p62 (a common component) in standard "p62 bodies" and ADPr Condensates, followed by IF to confirm significantly different composition in, what is

argued here, these distinct structures.

- Fine mapping of concentration dependence of components that give rise to these distinct condensates as has been demonstrated in papers like Riback et al Nature 2020 and others.
- Methodology of the author's choosing to decipher functional outcomes from these condensates followed by demonstration that components unique to ADPr condensates are dispensable for functioning "p62 bodies" and, vice versa, components unique to "p62 bodies" are dispensable for ADPr Condensate function.

****Minor Comments:****

Overall, the representative microscopy images are far too small. For the benefit of future readers, please consider enlarging these images.

More of the quantitation of microscopy images, with accompanying statistics, that are found in abundance in the supplemental material should find their way into the main figures of the manuscript. This will give room for larger and more reader-friendly representative microscopy images in the main figures/text as discussed briefly above.

Can the authors test whether or not the condensates are purely driven by mono(ADP-ribosyl)ation? Or does poly(ADP-ribose) co-occupy these condensates and play a substantive role?

The manuscript would benefit from discussing very recent and related reports (Kar et al., EMBO J [2024] and Chaves Ribeiro et al EMBO J [2024]), that I suspect were not available at the time of submission.

IFNalpha and IFNbeta, which are used in Figure S1, do not appear as reagents in Table 1.

On lines 113-114, it would seem more appropriate to describe the increase of PARP-14 as statistically significant and largest in magnitude. "most significant" would just mean lowest p-value, which I expect is different that the authors intend here.

In Figure S1, better care should be taken to crop and align the western blots.

On line 154, it may be more appropriate to describe ITK as a "weaker" inhibitor of PARP14 relative to PARP11. It certainly is effective as an inhibitor (Figs. S2A and S2G) and its unclear how the authors (or anyone would) define what qualities make it "weak".

The multiple bands for PARP14 in Figure 3E should be addressed. Why does this differ from other blots from the same cells?

2. Significance:

Significance (Required)

I expect the advances in this work will appeal more to specialists who are interested in ADP-ribosylation as a signaling molecule and to those engaged in biotechnological efforts to drug immunological responses.

The advances reported here are incremental. The ADPr condensates that form in response to IFN γ , the involvement of PARP9/DTX3L, and very recently the involvement of PARP14 and its MARYlation activity are all known. Less known is the notion that this condensate is distinct from other kinds of "bodies", which is a clear point of novelty, especially if buttressed by the authors as suggested in this review.

3. How much time do you estimate the authors will need to complete the suggested revisions:

Estimated time to Complete Revisions (Required)

(Decision Recommendation)

Between 1 and 3 months

No

Revision Plan

Manuscript number: RC-2024-02532

Corresponding author(s): Anthony K. L. Leung

1. General Statements

Our work addresses the formation of a novel class of interferon-induced ADP-ribosylation (ADPr)-containing condensates that can be reversed by SARS-CoV-2, which has been implicated as part of the antiviral responses. We have made significant advances by (1) revealing the identity of these condensates, (2) defining the responsible ADP-ribosyltransferase, and (3) determining the requirements for condensation.

We are encouraged by the reviewers' responses. Reviewer 1 acknowledged the novelty of our studies on the requirement of the ubiquitin-proteasome system in the formation of ADPr-containing PARP14/p62 condensates, recognizing that our condensate investigation is timely and important in cell biology, and noting that “data are clear and the phenomenon is of interest.” Reviewer 2 recognized the significance of our work in the rapidly growing field of PARP biology. Reviewer 3 noted that our work is significant for multiple fields, including ADP-ribosylation and immunology, and acknowledged our work as high quality and well-supported by the results. We appreciate that the reviewer deemed our work ready for publication with minor revisions.

Although our submission was made in May, two related studies were published in June (Kar et al., EMBO J, 2024; Ribiero et al., EMBO J, 2024) that reported on the responsible ADP-ribosyltransferase PARP14, leading Reviewer 3 to view our work as incremental. One of the strengths of Review Commons is its Scooping Protection Policy. Thus, these back-to-back EMBO J studies should not be perceived as diminishing our impact but rather as highlighting the importance of reporting the critical role of PARP14 in ADPr-containing condensate formation. Importantly, our study surpasses these studies in three key aspects: (1) the rigor of our approach, using multiple genetic manipulation techniques (siRNA, shRNA, and CRISPR-Cas9) and chemical inhibitors (catalytic and PROTAC) to demonstrate the specific requirement of PARP14; (2) identifying the condensate as structurally related to p62 bodies; and (3) establishing that an active ubiquitin-proteasome system is essential for ADPr condensation.

2. Description of the planned revisions

Reviewer #1 (Evidence, reproducibility and clarity (Required)):

Summary

In this study, Raja et al. found cytoplasmic condensates formed by the treatment of INF γ , investigated components of these condensates and identified p62, NBR1 and PARP14 as their components. INF γ treatment induced PARP14 expression, and PARP14 inhibitor treatment inhibited condensation formation, suggesting that the amount of PARP14 and its enzymatic activity

Revision Plan

are important for the condensate formation. The ADPr-positive p62 condensates were independent of autophagic degradation, and proteasomal activity was required for their formation.

Major comment

1. The finding that the ubiquitin-proteasome, but not autophagy activity, is indispensable for the formation of p62 condensates is of interest. However, the molecular mechanism by which the ubiquitin-proteasome system (UPS) is involved in the regulation of the PARP14-p62 condensate is still unclear. Which step(s) is the UPS involved?

We appreciate the reviewer's acknowledgment of the novelty of our studies on the requirement of the ubiquitin-proteasome system (UPS) in the formation of ADPr-containing PARP14/p62 condensates. We have demonstrated that condensate formation requires the first and last steps of the UPS using two distinct classes of inhibitors:

- (1) TAK243 inhibits the E1 enzyme by forming a covalent adduct with ubiquitin that mimics the ubiquitin-adenylate complex, thereby blocking the initial step of ubiquitin conjugation (Fig. 6F-G).
- (2) Three different proteasome inhibitors with varying degrees of selectivity—MG132, epoxomicin, and Bortezomib—block the final step of the UPS by inhibiting the 26S proteasome (Fig. 6B, S6D).

Given that blocking the early steps of the ubiquitin conjugation pathway or the late stages of the UPS inhibits the formation of ADPr condensates, we deduce that an active UPS is required.

The UPS involves three major steps of ubiquitination, performed by E1, E2, and E3 enzymes, followed by proteasomal degradation. Among these steps, E3 ubiquitin ligases play a crucial role in determining which proteins are ubiquitinated and thus marked for degradation. Previous studies have shown that the E3 ligase DTX3L, which shares the same genomic loci with PARP14, is co-expressed and colocalized at the condensate upon interferon stimulation (Russo et al., JBC, 2021; Kar et al., EMBO J, 2024; Ribiero et al., EMBO J, 2024). This E3 ligase is also critical for the formation of these condensates, as demonstrated by genetic depletion strategies. However, it remains unclear whether the E3 ligase activity is required.

To further narrow down which steps of the UPS are involved, our revision plan includes:

- (1) Exploration of the importance of DTX3L enzymatic activity. We will knock down this E3 ligase and reintroduce either the wild-type or a catalytically dead version by mutating four zinc-coordinating residues in the RING finger domain (C576S, H578S, C581S and C584S; Tessadori et al., Nature Genetics 2017).
- (2) Investigating condensate formation using DUB inhibitors, as deubiquitination also impacts the UPS.

Revision Plan

3. The authors claim that the amount of PARP14 and its MAR activity are essential for the condensate formation. However, all experiments were performed only with PARP14 inhibitors, and further validation is needed. If the importance of PARP14 activity is to be directly demonstrated, experiments in which an enzyme activity mutant is introduced into PARP14 KO cells are needed.

We would like to clarify that we have not only used PARP14 chemical inhibitors to reduce MAR activity but also employed PROTAC to reduce the amount of PARP14 (Fig. 1H). Both approaches demonstrated that the inhibition of either the amount or MAR activity of PARP14 is critical for condensate formation. Additionally, we demonstrated that condensate formation is reduced upon PARP14 knockdown using siRNA and shRNA, as well as CRISPR-mediated knockout (Fig. 1G and S1C-H).

Furthermore, we showed that transient transfection of a PARP14 mutant deficient in ADP-ribosylhydrolase activity into U2OS cells leads to the formation of ADPr condensates that colocalize with PARP14, independent of IFN γ treatment. Notably, a subset of condensates—particularly the larger ones—that contain both PARP14 and ADPr showed strong colocalization with p62 (Fig. 4G). Treatment with PARP14 MAR activity inhibitor under these conditions resulted in the disappearance of ADPr/PARP14 condensates while p62 bodies remained (Fig. 4H), further indicating that ADPr enrichment in p62 bodies depends on the MAR activity of PARP14.

To further confirm the dependence on MAR activity, we have now repeated the experiment using a PARP14 mutant deficient in MAR activity. PARP14 and ADPr condensates were not observed upon expression of this mutant, indicating that condensate formation depends on PARP14 MAR activity.

4. In Figure 2a, the heatmap alone is insufficient. Neither errors nor statistical comparisons are indicated.

We will incorporate our statistical data presented in Fig. S2A-D into Fig. 2A.

Revision Plan

5. The statistical analysis of Figure S2 is inappropriate; instead of t-tests, multiple comparisons should be used to compare three or more groups.

As suggested by the reviewer, we are providing additional statistical analyses for the data presented in Fig. S2A-F and Fig. S2G. Since we are revisiting our quantification on a per-cell basis rather than per-field (see Response to Reviewer 2), we will also perform the multiple comparison analyses, as suggested, once we complete our analyses.

Minor comment

1. What percentage of p62 condensates upon IFN γ treatment are ADPr positive? Are all p62 bodies seen with IFN γ stimulation unrelated to autophagy?

We will perform the quantification as suggested for ADPr and p62, ADPr and LC3B, and p62 and LC3B.

2. Is ADPr condensation a PARP14-specific phenomenon? PARP9 and PARP12 were also upregulated by IFN γ treatment. Are these factors also involved in condensate formation?

Amongst all catalytically active PARPs, ADPr condensation requires only PARP14. Russo et al., J Biol Chem 2021, have shown that genetic knockout of PARP9 affects the formation of ADPr condensates; however, PARP9 is catalytically inactive as an ADP-ribosyltransferase. Ribeiro et al., EMBO 2024, have further confirmed the requirement of PARP9 by siRNA knockdown and have also shown that condensate formation does not require PARP12. Based on the reviewers' comments, we will independently confirm this observation by performing knockdown experiments.

Revision Plan

3. Figure 4D appears to be immunoprecipitation (IP) under non-denaturing conditions. If so, it is not possible to distinguish whether the MAR signal is derived from p62 or from the p62 interacting proteins (the associated ubiquitinated substrates). IP experiments should be performed under denaturing conditions.

In collaboration with Drs. Sara Buch-Larsen and Ivo Hendriks (Novo Nordisk Foundation Center for Protein Research), we have identified a couple of high-confidence ADP-ribosylation sites on p62 in A549 cells treated with IFN γ . We will provide the mass spectra to support these findings. Additionally, we would like to note that following our submission, Kubon et al. reported in a bioRxiv preprint that p62 is ADP-ribosylated in a PARP14-dependent manner upon treatment with type I interferon, IFN β . This finding is consistent with our study involving type II interferon, IFN γ .

4. In Figure 5B, which band is HO1, the upper or lower?

Both bands are HO1, as shown by Biswas et al., J Biol Chem 2014. One band appears at 28 kDa and the other at 32 kDa. The 32-kDa isoform is predominantly constitutive in the cytoplasm, whereas the 28-kDa HO-1 is predominant and primarily localized to the nucleus.

5. There is no image for ubiquitin in S5D.

Our original statement, “However, when inhibited with the mTOR inhibitor Torin-1, autophagy is induced, leading to increased autophagosome formation marked by LC3B on the membranes, which facilitates the recruitment of p62 and ubiquitinated proteins (Fig. S5D),” contained a misplaced figure citation. The correct statement should be: “However, when inhibited with the mTOR inhibitor Torin-1, autophagy is induced, leading to increased autophagosome formation marked by LC3B on the membranes (Fig. S5D), which facilitates the recruitment of p62 and ubiquitinated proteins.” Our intention was to show that there are conditions, such as Torin-1 treatment, where p62 and LC3B colocalize.

Right panel in Figure 4F shows only IF γ + RBN, which should show all data sets in the same panel.

Given the complexity of the three conditions with extensive data points and error bars on the FRAP experiments, we aim to present the data clearly. Instead of merging the panel into one figure, we initially provided a summary table in Figure 4F. However, in response to the reviewer's comments, we will provide the composite image that includes all data sets in the same panel.

Reviewer #2 (Evidence, reproducibility and clarity (Required)):

This manuscript investigates the formation of a novel cellular structure or condensate, similar to p62 bodies, that includes PARP14 and p62. The interferon-induced PARP14-mediated ADP-ribosylation of p62 in these condensates depends on an active ubiquitin-proteasome system. These condensates are characterized by the presence of PARP14 and ADPr and include some, but not all, components of the p62 bodies. Furthermore, their formation depends on both ubiquitin activation and proteasome activity, but it is unaffected by autophagy inhibition, unlike conventional p62 bodies.

Revision Plan

The Introduction provides a well-delineated context of condensates, highlighting the importance of post-translational modifications in responding to environmental changes.

Although the manuscript is well-organized with apparent logical development, there are weaknesses that diminish the impact of the reported data. A more accurate review of the structures, both from a morphological and quantitative perspective, would strengthen the conclusions and the overall impact of this work. Additionally, while the authors have analyzed the contribution of PARP14 to condensate formation, the biological significance of these structures remains unclear. For instance, performing MS (mass spectrometry) analysis on the described structures could help identify their composition and functions.

The methodologies used in this study are standard for molecular and cellular biology research, including immunofluorescence assays, transient transfections, immunoprecipitations, and fluorescence recovery after photobleaching (FRAP) assays. These methods are described in detail and can be reproduced.

Below, please find a list of comments and suggestions to enhance the robustness of the data:

We thank the reviewer for acknowledging the logical progression of the manuscript and the detailed, reproducible methods. As detailed below, we will perform super-resolution microscopy experiments to examine morphology, improve data quantification, and conduct proteomics experiments to identify how the p62 interactome changes upon IFN γ treatment.

Major Points

1. The IF analyses are central to the conclusions reported and are employed for each of the inhibitors or other tools used to investigate the formation of these condensates. The quality of the IF images needs to be improved; the shape and contacts of the condensates should be analyzed using either super-resolution or EM microscopy, or preferably both. The lack of morphometry and quantification from cell populations needs to be addressed for all experiments. These analyses are needed to support the claim that the condensates presented in this study are indeed novel structures, rather than being transient aggregates of a different nature.

We believe the reviewer may be referring to the size of the images rather than their quality, as they are of high resolution when zoomed in. However, we agree that larger images would enhance clarity. We will be discreet in choosing the figures to present, given that we have over 550 panels in the main figures and over 250 in the supplemental figures. We will resize our figures to ensure the images are clearly visible.

Revision Plan

We will perform Airyscan imaging to provide super-resolution images for a better understanding of the morphology of these condensates. We will perform the morphometry quantification (circularity and ellipticity). For quantification, we would like to point out that nearly all of our experiments were analyzed from at least 4 fields, each containing 20-50 cells (depending on the magnification of 20x or 40x). In response to the reviewer's comment, we will also provide quantification on a per-cell basis.

Our data indicate that these are novel ADPr-containing condensates that colocalize with PARP14, p62, NBR1, and ubiquitin, but not LC3B (Fig. 1F, 3B, 5E-F, 5I). These structures are inducible by IFN γ treatment and can be inhibited by even 1 hour of PARP14 inhibition (Fig. 1A-B, 2D). They are dependent on PARP14 induction and its ADP-ribosyltransferase activity (Fig. 1G, 2B, 2D). These structures are not protein aggregates, as evidenced by their lack of staining with ProteoStat Dye (Fig. S6A), which stains for unfolded proteins.

2. The claim that PARP14 is essential for the formation of condensates requires support by the analyses indicated above. Minor points regarding Fig. 1 are indicated below. I suggest performing KD of PARP9 and/or PARP12 (whose expression is increased upon IFN treatment) and checking ADPr condensates to validate the central role of PARP14.

ADPr condensation requires PARP14, as demonstrated by multiple genetic depletion techniques (siRNA/shRNA/CRISPR; Fig. 1G, S1C-H) and chemical inhibitors (catalytic and PROTAC; Fig. 1H, 2B, 2D). We will provide additional image analyses to support the claim. In addition, Russo et al., J Biol Chem 2021, have shown that genetic knockout of PARP9 affects the formation of ADPr condensates; however, PARP9 is catalytically inactive as an ADP-ribosyltransferase. Ribeiro et al., EMBO 2024, have further confirmed the requirement of PARP9 by siRNA knockdown and have also shown that condensate formation does not require PARP12. Based on the reviewers' comments, we will independently confirm this observation by performing knockdown experiments.

3. According to the text, "PARP14 was pulled down by ADP-ribose binding Af1521 macrodomain following IFN γ treatment (Fig. 2H)", but the legend to the figure says otherwise. A Pan-ADPr binding reagent (MABE1016) is reported in the figure. Although the conclusion is similar for the results obtained with these two tools (but they must be described and reported properly), it is still insufficient to claim that PARP14 is ADP-ribosylated. This point should be at least discussed.

We apologize for the confusion. The Pan-ADPr binding reagent (MABE1016) is a His-tagged recombinant Af1521 macrodomain that binds to ADP-ribosylated proteins (Gibson et al., Biochemistry 2017). Therefore, we used Ni-NTA resin to pull down His-tagged AF1521 for the subsequent analysis of PARP14. We will revise the text and figure legend for Fig. 2H to clarify this. Additionally, in collaboration with Drs. Sara Buch-Larsen and Ivo Hendriks (Novo Nordisk

Foundation Center for Protein Research), we have identified multiple ADP-ribosylation sites on PARP14 in A549 cells treated with IFNγ. We will provide the mass spectra to confirm that PARP14 is indeed ADP-ribosylated in cells. Consistent with our studies, Kar et al. and Ribeiro et al. (EMBO J. 2024) also recently reported that PARP14 is ADP-ribosylated upon IFNγ treatment in A549 cells. Additionally, Higashi et al. (J. Proteome Res. 2019) reported PARP14 ADP-ribosylation in IFNγ-treated macrophage cells. We will include these references in our *Discussion*.

4. I have difficulties analyzing the colocalization with the different organelles, even enlarging the images as much as possible. In most cases, only one condensate per image is shown. Continuities with the nuclear envelope appear in some cases: has this been investigated?

We have provided images of at least two cells containing multiple cytoplasmic condensates in Figure S3A. We believe part of the confusion arises from the staining pattern of ADPr in the nucleus, which colocalizes with splicing speckles. However, this nuclear staining was not altered by IFNγ treatment. Therefore, we have focused on the cytoplasmic condensates in this study and will clarify this focus in the main text. However, in response to the reviewer's question, we will also visit the possibilities of enrichment of signals at the nuclear envelope. For each colocalization study, we have analyzed at least four fields, each containing 20-30 cells. To further strengthen our claim for colocalization analyses, we will now provide quantification on a per-cell basis.

Minor Points

1. Fig. 1A: The DAPI images at 3 and 6 hours are reversed. Additionally, for Fig. S1a and Fig. 1A, please include quantifications.

We apologize for the oversight and will provide quantification.

2. Fig. 1B: Check PARP14 levels (and other IFN-PARPs) under the same experimental conditions.

As suggested, we have assessed the PARP14 protein levels. Additionally, we will assess the levels of other IFNγ-induced PARPs, including PARP9, PARP12, and PARP14, using qPCR.

Revision Plan

3. Fig. 1H: Explain why PARP14 IF staining is still visible upon RBN012811 treatment, while it is completely lost in WB analysis or upon PARP14 siRNA treatment (Fig. 1G). In addition, please include IF quantifications.

To clarify, for Figure 1H, the images provided correspond to those from IFN γ -treated cells while the western blot data include both with and without IFN γ treatment. We would like to point out that RBN012811 treatment indeed shows a similar dose-dependent signal with increasing hours of treatment, comparable to the western blot results. Specifically, we observed a small amount of PARP14 remaining at the 1-hour timepoint on western blots and IF images, with the highest intensity observed compared to other timepoints. In addition, we believe part of the discrepancy is due to background staining by PARP14 antibodies. Therefore, we will examine the level of background staining in PARP14 KO cells and provide corresponding IF quantification.

4. Fig. 2C: Please include quantifications.

We will provide quantification.

5. Fig. 2E: The RBN treatment time is not indicated. Please include this information in the figure legend.

We apologize for the oversight and will add the treatment time (24 h) to the figure legend.

6. Fig. 2G: I am not convinced about the PARP14 staining. IF images do not show an increase in PARP14 levels, while WB analysis shows a strong increase in PARP14 protein levels (see Fig. 2E). Moreover, the RBN treatment time was not indicated; please include it in the figure legend. Does RBN alone affect PARP14 localization? The reported picture shows only 2 cells, each with a different subcellular localization of PARP14. As previously suggested, quantifications are required.

When presenting the data, our aim was to show the pattern rather than the relative intensity difference. Therefore, we used the autocontrast image function across different conditions, which resulted in an apparent change in pattern even with weak signals in control or RBN-treated samples. To address this, we will ensure the images presented across different conditions have the same exposure and are shown with consistent image contrast parameters. We will also include quantification of the condensates. Additionally, we apologize for the oversight and will add the RBN treatment time to the figure legend.

7. Fig. 3B: Pearson's correlation coefficient (PCC) is reported for $n=3$. The images show one condensate per cell. Under these conditions, the number of cells analyzed should be at least 100 for each experiment. Additionally, the PCC between PARP14 and p62 at steady state is shown to be 60% (which is quite high). However, the IF pictures do not support this quantification. Can the

Revision Plan

authors provide higher-resolution pictures? Does PARP14 always co-localize with p62? Lines 207-208 state: "these findings suggest that PARP14 is localized to p62 bodies upon IFN γ treatment when ADP-ribosylation occurs." According to the PCC value, the two proteins co-localize even in the absence of IFN. Can the authors clarify this aspect?

We apologize for the inaccurate description. The data should be n=4, representing different fields (as indicated by the number of dots in the original graph panels in Fig. 3B), each containing 20-30 cells. The Pearson's correlation coefficient was calculated across the cells, instead of focusing on the condensates—we will provide additional analyses on condensate colocalization analyses. We will also provide larger images and quantification to indicate the level of PARP14 colocalization with p62. For PARP14, we did not see a significant number of PARP14 condensates in control cells; PARP14 condensates were seen only after IFN γ treatment. A fraction of PARP14 condensates did not colocalize with p62. We will provide detailed quantification analyses.

8. Fig. S3B: Please include quantifications.

Quantification will be provided.

9. Fig. S3C: How was the condensate size quantified? It would be useful to show a quantification mask.

We apologize for the omission in the Method Section. The condensate size quantification was performed with ImageJ. The nuclei were first identified with DAPI staining and masked out from the ADPr channel. The image was then thresholded with the "Maximum Entropy" method from ImageJ and the "Analyze Particles" function was used to identify condensates with size larger than 1 pixel. Quantification mask will be provided.

10. Fig. 3D: Does p62 KD affect PARP14 localization? The reported picture shows only 2 cells, each with different staining of PARP14.

p62 KD reduced the number of PARP14 condensates but did not change their localization. We will provide a representative image with more cells to illustrate this effect more clearly and provide quantification on the change in number of condensates.

11. Fig. 4A: Please quantify the PARP14 co-IP signal with p62, normalized to PARP14 total levels. In the reported WB, it is difficult to see the interaction between PARP14 and p62 in untreated conditions. Please provide clearer WB.

As suggested, we will quantify the PARP14 co-IP signal by normalizing it with PARP14 input levels. Additionally, we will provide a clearer WB in the revised manuscript.

Revision Plan

Additionally, I would expect an increased interaction between PARP14 and p62 upon IFN treatment due to PARP14 recruitment to p62 condensates, not just because of increased PARP14 levels. Since the authors show that PARP14 is not recruited to ADPr condensates upon RBN treatment (Fig. 2G), why is the interaction between p62 and PARP14 so high under RBN treatment?

RBN treatment inhibits PARP14 catalytic activity but simultaneously increases PARP14 levels, as first described by Schenkel et al., Cell Chem Biol 2021. Western blot data indicate that the interaction between PARP14 and p62 is independent of this activity and instead depends on PARP14 protein levels. However, the formation of ADPr/PARP14-containing condensates requires catalytically active PARP14. Based on these data, we conclude that the colocalization of p62 and PARP14 depends on the catalytic activity of PARP14, which is reflected by its ADP-ribosylation.

12. Fig. 4C: Please quantify WB signals of ADP-ribosylated p62 for the different conditions analyzed. ADP-ribosylation of p62 is still present in cells lacking PARP14. Are there other enzymes that can modify p62? Moreover, the authors state: "We observed an increased MARylation of p62 upon IFN γ treatment" (line 230); is this dependent on the increase in PARP14 levels or the translocation of PARP14 to ADPr condensates? Quantifications should help clarify this aspect.

As suggested, WB signals has been quantified:

Revision Plan

We agree with the reviewer's observation regarding the presence of ADP-ribosylated p62 in PARP14 KO cells. The basal levels of ADP-ribosylated p62 may be due to other PARP enzymes. However, PARP14 is critical for the increase in ADP-ribosylation under IFN γ treatment, as the increase was not observed in PARP14 KO cells. Given that PARP14 inhibitors increase PARP14 levels, we interpret that the increase in p62 MARylation requires an increase in active PARP14 levels, not just its total level. Since PARP14 activity is crucial for the localization of PARP14 to p62 condensates and its enrichment of ADPr signals, it is possible, as suggested by the reviewer, that the increase in MARylation of p62 is dependent on the translocation of PARP14 to the structure. However, the field currently lacks the tools to disrupt p62 bodies without knocking down p62 to definitively test whether colocalization is required for the MARylation increase.

13. Fig. 4G: Quantifications are required.
Quantification will be provided

Reviewer #2 (Significance (Required)):

The role of the PARP family in cellular processes is a very active and rapidly growing field. New information about the organization of PARPs in the nucleus, cytosol, or different types of bodies/structures is certainly relevant to the field.

However, the present study is too preliminary at the moment to be considered highly relevant. Both the data analysis and conclusions need to be carefully reviewed. After major revisions, the manuscript might be of general interest if well contextualized within the fields of post-translational modification and protein degradation processes. It would remain in any case interesting for the field of ADP ribosylation.

We thank the reviewer for recognizing the significance of our work in the rapidly evolving field of PARP biology. We apologize for the lack of clarity that we indeed quantified over 100 cells across at least 4 fields of images for the data reported. To further address the concerns raised, we will provide additional cell-based quantification to strengthen our claims. Furthermore, we will enhance the contextualization of our findings within the broader frameworks of post-translational modification and protein degradation processes in the Introduction and Discussion sections.

Reviewer #3 (Evidence, reproducibility and clarity (Required)):

In this manuscript, titled "Interferon-Induced PARP14-Mediated ADP-Ribosylation in p62 Bodies Requires an Active Ubiquitin-Proteasome System", Raja et al. perform fluorescence microscopy assays and molecular analyses on cultured cells *ex vivo* to further our understanding of ADP-ribose accumulations that form in the cytoplasm in response to IF γ stimulation. Guided by the operon-like linkage and co-expression patterns of PARP14, PARP9, and DTX3L and recent reports describing a

Revision Plan

SARS-CoV-2-dissolved cytoplasmic body induced by interferon-induced that is rich in ADP-ribose and the PARP9/DTX3L heterodimer form following IF γ stimulation, the authors provide clarity in this manuscript regarding two knowledge gaps - (i) what catalyzes mono(ADP-ribosyl)ation within these structures (as PARP9 lacks ADP-ribosyltransferase activity) and (ii) how these foci/condensates relate to similarly composed autophagy-associate "p62 bodies" that have been previously described. Using a combination of genetic depletion and inhibitor-based approaches, the authors show that these ADPr-condensates rely on the catalytic activity of PARP14. The authors also show that while these ADPr-condensates share componentry with "p62 bodies" like ubiquitin and p62 itself, these foci are distinct accumulations, as they lack both LC3B and sensitivity to autophagy inhibitors and require an active ubiquitin-mediated proteosomal degradation system.

While this report represents a more incremental advance in our understanding of these cell signaling structures, especially considering a pair of very recently published and similar reports (Kar et al., EMBO J [2024] and Chaves Ribiero et al EMBO J [2024]), the work here is well-written and reasoned and complements these works with some novelty and distinction to that reported literature. The experiments are definitive and of high quality and the authors interpretations/conclusions are largely well-supported by the results. Thus, it is my opinion that this work is appropriate for publication with predominantly minor revisions (outlined below) and a few more substantive experimental additions.

We thank the reviewer for recognizing the high quality of our work and for acknowledging that our interpretations are well-supported by the results. We appreciate that the reviewer deemed our work ready for publication with minor revisions. We believe the reviewer's perception of our work as incremental arises because two related studies were published in June, after we submitted our work to Review Commons in May. According to the Scooping Protection Policy, our work should still be considered novel. The publication of these studies in EMBO J, which discovered the role of PARP14 in ADPr-containing condensate formation, highlights the significance of our research. However, we surpass their findings by demonstrating the critical role of PARP14 using multiple genetic manipulation techniques (siRNA, shRNA, and CRISPR-Cas9) and chemical inhibitors (catalytic and PROTAC), indicating the rigor of our studies. Moreover, we not only investigate PARP14 but also define the identity of these condensates related to p62 bodies and establish the requirement of an active ubiquitin-proteasome system for their formation.

Minor Comments:

Overall, the representative microscopy images are far too small. For the benefit of future readers, please consider enlarging these images.

More of the quantitation of microscopy images, with accompanying statistics, that are found in abundance in the supplemental material should find their way into the main figures of the

Revision Plan

manuscript. This will give room for larger and more reader-friendly representative microscopy images in the main figures/text as discussed briefly above.

We appreciate the reviewer's suggestions and rightly pointed out that our quantification and statistics were in supplementary materials. Given that we have provided over 800 image panels, we will restructure the manuscript so that the cell biology information is more readily available. We will move our quantification data and statistics currently in the supplementary materials to the main figures. Additionally, we will provide larger and more reader-friendly representative microscopy images in the main text as suggested.

Can the authors test whether or not the condensates are purely driven by mono(ADP-ribosyl)ation? Or does poly(ADP-ribose) co-occupy these condensates and play a substantive role?

We have tested for the presence of poly(ADP-ribose) in the condensates and found it is not present. We will provide the supporting data.

The manuscript would benefit from discussing very recent and related reports (Kar et al., EMBO J [2024] and Chaves Ribiero et al EMBO J [2024]), that I suspect were not available at the time of submission.

Yes, the reviewer is correct that our submission preceded the publication of these related reports. We will add a section to discuss their findings.

IFN α and IFN β , which are used in Figure S1, do not appear as reagents in Table 1.

We apologize for the oversight and will add the information on IFN α and IFN β to Table 1.

On lines 113-114, it would seem more appropriate to describe the increase of PARP-14 as statistically significant and largest in magnitude. "most significant" would just mean lowest p-value, which I expect is different that the authors intend here.

We thank the reviewer's suggestion will modify the text as follows:

"PARP9, PARP12, and PARP14 were statistically significantly upregulated at 6 and 24 hours post-treatment, with PARP14 showing the largest increase in mRNA expression levels (Fig. 1D and S1B)."

In Figure S1, better care should be taken to crop and align the western blots.

We thank the reviewer for pointing this out, and we will properly align and crop western blots in Figure S1.

Revision Plan

On line 154, it may be more appropriate to describe ITK as a "weaker" inhibitor of PARP14 relative to PARP11. It certainly is effective as an inhibitor (Figs. S2A and S2G) and its unclear how the authors (or anyone would) define what qualities make it "weak".

We thank the reviewer's suggestion and will modify the text as follows:

"...—specifically, RBN012579 (hereafter RBN) and ITK7 (a potent PARP11 inhibitor with inhibitory effects on PARP14 that are weaker than RBN)—..."

The multiple bands for PARP14 in Figure 3E should be addressed. Why does this differ from other blots from the same cells?

We believe that the multiple bands seen can be due to insufficient blocking and using a different lot of PARP14 antibodies. We have addressed this issue now by performing a new experiment with proper blocking conditions and using the same lot of PARP14 antibody as other blots. It should be noted that that variation is also observed in the reagent website: <https://www.scbt.com/p/parp-14-antibody-c-1>

Reviewer #3 (Significance (Required)):

I expect the advances in this work will appeal more to specialists who are interested in ADP-ribosylation as a signaling molecule and to those engaged in biotechnological efforts to drug immunological responses.

The advances reported here are incremental. The ADPr condensates that form in response to IFN γ , the involvement of PARP9/DTX3L, and very recently the involvement of PARP14 and its MARYlation activity are all known. Less known is the notion that this condensate is distinct from

other kinds of "bodies", which is a clear point of novelty, especially if buttressed by the authors as suggested in this review.

We agree with the reviewer that our work is significant for multiple fields, including ADP-ribosylation and immunology. The perception of our work as incremental likely stems from the publication of two related recent studies in June, after our submission in May. According to the Scooping Protection Policy in Review Commons, our work remains novel in editorial consideration. More importantly, the back-to-back EMBO J studies highlight the importance of reporting the critical role of PARP14 in ADPr-containing condensate formation. We went above and beyond in three aspects: (1) we rigorously demonstrate the critical role of PARP14 through multiple genetic techniques (siRNA, shRNA, and CRISPR-Cas9) and chemical inhibitors (catalytic and PROTAC), (2) we reveal the identity of these condensates as related to p62 bodies, and (3) we define their requirement for an active ubiquitin-proteasome system.

3. Description of the revisions that have already been incorporated in the transferred manuscript

We submit the original manuscript, but we have included new data and analyses above and provide a revision plan.

4. Description of analyses that authors prefer not to carry out

Please include a point-by-point response explaining why some of the requested data or additional analyses might not be necessary or cannot be provided within the scope of a revision. This can be due to time or resource limitations or in case of disagreement about the necessity of such additional data given the scope of the study. Please leave empty if not applicable.

Reviewer 1

2. The p62 condensate serves as a scaffold for autophagosome formation through the assembling autophagy receptors including NBR1 and TAX1BP1, followed by recruiting ATG proteins such as FIP200. While ADPr-positive p62 condensates also contain NBR1 and polyubiquitinated proteins, they are unrelated to autophagic degradation. It is unclear what factors govern autophagy-independent function.

As we have identified the requirement of active ubiquitin-proteasome system in regulating these condensates, determining the factors that govern autophagy-independent functions, though interesting, is beyond the scope of this manuscript. Our data indicate that the formation of ADPr-containing condensates, which include p62, other autophagy receptors such as NBR1, and polyubiquitinated proteins, but lack the autophagosome membrane protein LC3B (Fig. 3B, 5E-I, and S5D). Notably, this condensate formation is not inhibited by treatment with Bafilomycin A1 and chloroquine, which target the final step of autophagy involving lysosome interaction (Fig. 6A).

Revision Plan

Therefore, we aim to strengthen the understanding of the relationship between these condensates and autophagy factors, with the goal of identifying where these condensates diverge from the autophagy process.

In response to the reviewer's comments, we will further investigate whether these condensates also include other autophagy receptors, such as TAX1BP1, as well as the downstream autophagosome protein FIP200. Additionally, we will genetically deplete the critical autophagy factor ATG5 to confirm orthogonally that the formation of these condensates is indeed independent of autophagy.

Reviewer #1 (Significance (Required)):

Liquid droplets, which have continuously being identified in cells, are a hot topic in cell biology. Droplet formation, structure, molecular dynamics, and degradation, as well as their abnormalities and disease development due to genetic mutation and stress, are of wide-ranging interest from basic to pathological aspects. Therefore, this research has the potential to attract interest from a wide range of fields.

General assessment

Overall, the data are clear and the phenomenon is of interest. However, the molecular mechanism and biological significance of the condensate formation is unknown; It is unclear why proteasome activity is required for the formation of PARP14-mediated ADP ribosylation. It is also unclear what the consequences are for the cell if the ADPr-positive condensates are not formed. The authors should address these general and important issues and provide the data if not all.

We thank the reviewer for acknowledging that our condensate investigation is timely and important in cell biology and for recognizing that “*data are clear and the phenomenon is of interest*”. As mentioned in the *Discussion*, these condensates can be reversed by the SARS-CoV-2 macrodomain in lung A549 cells, whose activity to remove ADP-ribosylation is critical for viral replication and pathogenesis, indicating the biological significance of these condensates. In addition, similar IFN γ conditions can induce PARP14 expression in melanoma, where PARP14 inhibition resensitizes these cancers to immunotherapy. Given that these ADPr condensates are also observed in A375 melanoma cells beyond lung cells (Fig. S3B), this provides additional context to investigate their biological significance in the future.

We would like to note that we have already made significant advances by (1) revealing the identity of these condensates as related to p62 bodies (Fig. 3-5), (2) defining the responsible ADP-ribosyltransferase as PARP14 (Fig. 1-2), and (3) determining the requirements for condensation through ubiquitin-proteasome system (Fig. 6). The proposed exploration of the functional consequences and significance is beyond the scope of this manuscript. However, we will further define the mechanistic involvement of which step of ubiquitin-proteasome system.

Revision Plan

Reviewer #3

Major Comment:

A major claim and novelty reported here is that the ADPr condensates are distinct from "p62 bodies". The evidence to support this rely largely on differences in their sensitivity to pharmacological treatments as well as somewhat subtle differences in FRAP recovery in p62 condensates after IFN γ treatment. But, this claim would be better supported with more comprehensive mapping of differences in the componentry or functional outcomes of these condensates. The authors might consider:

-Mass spectrometry against p62 (a common component) in standard "p62 bodies" and ADPr Condensates, followed by IF to confirm significantly different composition in, what is argued here, these distinct structures.

-Fine mapping of concentration dependence of components that give rise to these distinct condensates as has been demonstrated in papers like Riback et al Nature 2020 and others.

-Methodology of the author's choosing to decipher functional outcomes from these condensates followed by demonstration that components unique to ADPr condensates are dispensable for functioning "p62 bodies" and, vice versa, components unique to "p62 bodies" are dispensable for ADPr Condensate function.

As rightly pointed out by the reviewer, our studies indicate the alteration in the composition and dynamics of p62 bodies upon IFN γ treatment. This was assessed using immunofluorescence against various known components of p62 bodies (Fig. 3B, 5E-I), quantification of the condensate size (Fig. S3C), and p62 mobility assessment by photokinetic experiments (Fig. 3C, 4F). In considering reviewer's suggestions, we will perform p62 interactome studies with and without IFN γ treatment to identify potential changes. Additionally, we will analyze the concentration dependence of ADPr for condensate formation. However, we believe that investigating the functional outcomes is beyond the scope of this manuscript.

Prof. Anthony K. L. Leung
Johns Hopkins University
Department of Biochemistry and Molecular Biology
Bloomberg School of Public Health
Baltimore, MD 21205

17th Sep 2024

Re: EMBOJ-2024-118598-T
Interferon-Induced PARP14-Mediated ADP-Ribosylation in p62 Bodies Requires an Active Ubiquitin-Proteasome System

Dear Anthony,

Thank you again for transferring your manuscript, together with a tentative point-by-point response, from Review Commons to The EMBO Journal, and my sincere apologies for the delay in getting back to you with a decision. I had invited all three original referees to look at your revision plan, and to comment on the whether the proposed changes would appear suitable for addressing their key concerns and for making this study a good candidate for an EMBO journal article. Unfortunately, due to some misunderstandings, combined with holiday-related absences at this time of the year, we have only now received the desired feedback from all of them.

The good thing is that all three Review Commons referees found your revision plans promising (see also some of their comments copied below for your information)! In this light, we shall therefore be happy to invite you to revise the manuscript along the lines suggested in your response letter. In particular, addressing referee 1's points 1 & 3 as proposed appears reasonable, while I agree that additional follow-up into the functional/physiological roles would well be beyond the scope of the present work. Similarly, I expect the additional super-resolution imaging & quantification, and autophagy marker should be helpful for addressing other major points raised in all reports. Finally, I highly appreciate the incorporation of proteomic analysis of p62 interactors, ideally with a certain (limited) amount of IF confirmation that some of these interactions would differ between conditions - happy to discuss further as the revision progresses. I hope that you would be able to complete these experiments and resubmit by the end of October, as estimated in your message.

When preparing a revised manuscript, please try to adhere to the guidelines listed below and in our Guide to Authors as closely as possible, as this should greatly facilitate our assessment at the time of resubmission - in particular regarding the completion of our author checklist and our new structured reagent table, the inclusion of editable text files and individual figures, and the conversion of "supplemental" material into Expanded View and/or Appendix content. Please also note that it is our policy to allow only a single round of (major) revision, making it important to comprehensively answer all criticisms at this point. Please do not hesitate to contact me should you have any further questions in this regard, and please do keep me updated about your resubmission timeline.

Thank you again for the opportunity to consider this study for The EMBO Journal. I look forward to receiving your revision.

With kind regards,

Hartmut

2) Each figure legend must specify
- size of the scale bars that are mandatory for all micrograph panels

- the statistical test used to generate error bars and P-values
- the type error bars (e.g., S.E.M., S.D.)
- the number (n) and nature (biological or technical replicate) of independent experiments underlying each data point
- Figures may not include error bars for experiments with $n < 3$; scatter plots showing individual data points should be used instead.

9) To facilitate reproducibility and cross-laboratory adoption of methodologies, please structure the Materials & Methods section as outlined in our guide to authors, including a completed Reagents and Tools Table that can be downloaded from our author guidelines as well (<https://www.embopress.org/page/journal/14602075/authorguide#structuredmethods>).

10) Digital image enhancement is acceptable practice, as long as it accurately represents the original data and conforms to community standards. If a figure has been subjected to significant electronic manipulation, this must be clearly noted in the figure legend and/or the 'Materials and Methods' section. The editors reserve the right to request original versions of figures and the original images that were used to assemble the figure. Finally, we generally encourage uploading of numerical as well as gel/blot image source data; for details see: embopress.org/page/journal/14602075/authorguide#sourcedata

At EMBO Press, we ask authors to provide source data for the main manuscript figures. Our source data coordinator will contact you to discuss which figure panels we would need source data for and will also provide you with helpful tips on how to upload and organize the files.

Please discuss the revision progress ahead of this time with the editor if you require more time to complete the revisions. Use the link below to submit your revision:

Link Not Available

Referee 1

I think that the authors adequately addressed the key points raised by the review and, that this work reaches the criteria to become acceptable for The EMBO Journal within a limited time frame.

In Major comment 1, I commented that the point of action of UPS is unclear.

The authors seem to be saying only that proteasome activity is important for the formation, since it accumulates with E1 inhibitors and proteasome inhibitors. (They seemed to simply assume that some of the proteasome substrates would be taken up.)

As per the proposed revision, the authors do not need to go into a detailed analysis of DTX3L, they could just make the text

clearer and show that K48 chains accumulate in the p62 body upon interferon treatment by immunostaining or other means.

Referee 2

I have now had the opportunity to analyze the revision proposed by the authors both on the basis of my comments and the comments of the other two reviewers.

Based on their report, the authors aim to satisfy all requests with new experiments, comments, quantitative analyses, use of super resolution and additional techniques, broadening of the discussion and so on.

It is clear that the manuscript that will result from this work will certainly be more complete, very different from the original, and of interest to a wider audience, therefore not only of interest to the ADP ribosylation field.

If all this is put into practice in the short time available, I think the manuscript can be considered by EMBO J or, if the revision will be partial, by Embo Rep.

Referee 3

I'd be happy to follow this manuscript to EMBO J. And the plan for revision seems thoughtful and appropriate.

Rev_Com_number: RC-2024-02532

New_manu_number: EMBOJ-2024-118598-T

Corr_author: Leung

Title: Interferon-Induced PARP14-Mediated ADP-Ribosylation in p62 Bodies Requires an Active Ubiquitin-Proteasome System

Reviewer #1 (Evidence, reproducibility and clarity (Required)):

Summary

In this study, Raja et al. found cytoplasmic condensates formed by the treatment of INF γ , investigated components of these condensates and identified p62, NBR1 and PARP14 as their components. INF γ treatment induced PARP14 expression, and PARP14 inhibitor treatment inhibited condensation formation, suggesting that the amount of PARP14 and its enzymatic activity are important for the condensate formation. The ADPr-positive p62 condensates were independent of autophagic degradation, and proteasomal activity was required for their formation.

Major comment

1. The finding that the ubiquitin-proteasome, but not autophagy activity, is indispensable for the formation of p62 condensates is of interest. However, the molecular mechanism by which the ubiquitin-proteasome system (UPS) is involved in the regulation of the PARP14-p62 condensate is still unclear. Which step(s) is the UPS involved?

We appreciate the reviewer's acknowledgment of the novelty of our studies on the requirement of the ubiquitin-proteasome system (UPS) in the formation of ADPr-containing PARP14/p62 condensates. To clarify the UPS steps involved, we demonstrated that both the initial and final stages of the UPS are essential for forming ADPr-containing condensates, as shown by using two distinct classes of inhibitors:

- (1) TAK243 inhibited the E1 enzyme by forming a covalent adduct with ubiquitin that mimics the ubiquitin-adenylate complex, thereby blocking the initial step of ubiquitin conjugation (Fig. 7B and EV7B).
- (2) Three different proteasome inhibitors with varying degrees of selectivity—MG132, epoxomicin, and Bortezomib—blocked the final step of the UPS by inhibiting the 26S proteasome (Fig. 6D and EV6E).

The inhibition of either the early ubiquitin conjugation or late proteasomal degradation steps disrupted ADPr condensate formation, indicating that an active UPS is necessary.

The UPS involves three major steps of ubiquitination, performed by E1, E2, and E3 enzymes, followed by proteasomal degradation. Among these steps, E3 ubiquitin ligases play a crucial role in determining which proteins are ubiquitinated and thus marked for degradation. Of note, prior studies identify the E3 ligase DTX3L, co-expressed and colocalized with PARP14 in interferon-stimulated cells, as a critical component in condensate formation (Russo et al., JBC, 2021; Kar et al., EMBO J, 2024; Ribiero et al., EMBO J, 2024). Our results align with these findings, as DTX3L depletion disrupts ADPr condensate formation (Fig. 7D and EV7D). Similarly, the knockdown of the E1 enzyme UBA1 resulted in condensate loss (Fig. 7A and EV7A), further validating the requirement for active UPS.

To assess whether ubiquitin recycling is necessary, we applied the deubiquitinating enzyme (DUB) inhibitor PR619. Treatment with PR619 increased steady-state levels of ubiquitinated proteins without disrupting condensate formation. Instead, the number of ADPr condensates increased (Fig. 7E and EV7E), further suggesting that ubiquitination is critical for ADPr condensate formation. Collectively, these data indicate that an active ubiquitin-proteasome system is required for the condensation of PARP14 and ADPr in p62 bodies.

2. The p62 condensate serves as a scaffold for autophagosome formation through the assembling autophagy receptors including NBR1 and TAX1BP1, followed by recruiting ATG proteins such as FIP200. While ADPr-positive p62 condensates also contain NBR1 and polyubiquitinated proteins, they are unrelated to autophagic degradation. It is unclear what factors govern autophagy-independent function.

As our data indicate, the formation of these condensates requires an active UPS but is unrelated to autophagic degradation. While identifying factors governing their autophagy-independent functions lies beyond this manuscript's scope, we provide substantial evidence differentiating these condensates from autophagy-driven structures.

Our data indicated that ADPr-containing condensates include p62, NBR1, and polyubiquitinated proteins, but lack the autophagosome membrane protein LC3B (Fig. 3A, 5D-G, EV5E-F, and EV5H). Notably, this condensate formation was not inhibited by treatment with Bafilomycin A1 and chloroquine, which target the final step of autophagy involving lysosome interaction (Fig. 6A).

In response to the reviewer's suggestion, we further explored whether additional autophagy receptor and autophagosome component are present in these condensates. We observed colocalization with the receptor TAX1BP1 (Fig. 5H and EV5L-M) but not with the downstream autophagy initiation component FIP200 (Fig. 5I and EV5N), suggesting that these condensates selectively associate with certain autophagy receptors while not recruiting autophagosome formation machinery. Additionally, genetic depletion of essential autophagy factors Beclin 1 and ATG5 did not alter ADPr condensate formation (Fig. 6B-C and EV6A-B), providing further confirmation that these condensates operate independently of autophagy.

Our proteomics analyses further revealed that the association of p62 with NBR1 and TAX1BP1 remains unchanged upon IFN γ treatment. However, p62 shows increased association with PARP14 and two additional proteins, PARP9 and DTX3L, both of which are known to co-express with PARP14 (Fig. 5J-K), consistent with our imaging analyses.

These results reinforce the distinct regulatory pathways governing ADPr-positive condensates, distinguishing them from canonical autophagy-related structures.

3. The authors claim that the amount of PARP14 and its MAR activity are essential for the condensate formation. However, all experiments were performed only with PARP14 inhibitors, and further validation is needed. If the importance of PARP14 activity is to be directly demonstrated, experiments in which an enzyme activity mutant is introduced into PARP14 KO cells are needed.

We would like to clarify that we have not only used PARP14 chemical inhibitors to reduce MAR activity but also employed PROTAC to reduce the amount of PARP14 (Fig. 1G-H). Both approaches demonstrated that the inhibition of either the amount or MAR activity of PARP14 is critical for condensate formation. Additionally, we demonstrated that condensate formation is reduced upon PARP14 knockdown using siRNA and shRNA, as well as CRISPR-mediated knockout (Fig. 1E-F and EV1F-K).

Furthermore, we showed that transient transfection of a PARP14 mutant deficient in ADP-ribosylhydrolase activity into U2OS cells leads to the formation of ADPr condensates that colocalize with PARP14, independent of IFN γ treatment. Notably, a subset of condensates—particularly the larger ones—that contain both PARP14 and ADPr showed strong colocalization with p62 (Fig. 4H). Treatment with PARP14 MAR activity inhibitor under these conditions resulted in the disappearance of ADPr/PARP14 condensates while p62 bodies remained (Fig. 4I), further indicating that ADPr enrichment in p62 bodies depends on the MAR activity of PARP14.

To further confirm the dependence on MAR activity, we have now repeated the experiment using a PARP14 mutant deficient in MAR activity. PARP14 and ADPr condensates were not observed upon expression of this mutant, indicating that condensate formation depends on PARP14 MAR activity (Fig. EV4D).

4. In Figure 2a, the heatmap alone is insufficient. Neither errors nor statistical comparisons are indicated.

We have incorporated our statistical comparisons presented in Fig. 2A and replaced the heatmap in the revised manuscript.

5. The statistical analysis of Figure S2 is inappropriate; instead of t-tests, multiple comparisons should be used to compare three or more groups.

As suggested by the reviewer, we have added statistical analyses for the data presented in Fig. EV2A-G, using Two-Way ANOVA, Tukey's multiple comparisons test.

In addition, we have revisited our quantification on a per-cell basis rather than per-field (see also Response to Reviewer 2).

Minor comment

1. What percentage of p62 condensates upon IFN γ treatment are ADPr positive? Are all p62 bodies seen with IFN γ stimulation unrelated to autophagy?

We have quantified the co-localization of ADPr with p62 as well as ADPr with LC3B, using LC3B as a proxy for autophagy. Specifically, for the percentage of p62 condensates positive for ADPr, we analyzed data corresponding to Fig. 3A and 3D, using a Pearson correlation coefficient threshold of > 0.6 . Upon IFN γ treatment, ~20% of p62 condensates were ADPr-positive, indicating a significant increase. This quantification is now provided as Fig. EV3D-E.

For LC3B-ADPr co-localization, we analyzed data from experiments associated with Fig. 5D. We did not observe any significant changes, which aligns with our hypothesis that ADPr condensates are devoid of LC3B (Fig. EV5G). This conclusion is further supported by experiments using pharmacological inhibitors such as Chloroquine, Bafomycin A1, as well as cell lines deficient in autophagy-related genes (ATG5 and Beclin 1; Fig. 6A-C and EV6A-B).

These findings collectively demonstrate that IFN γ induces ADPr modification in a subset of p62 bodies, which lack LC3B and are independent of autophagy.

2. Is ADPr condensation a PARP14-specific phenomenon? PARP9 and PARP12 were also upregulated by IFN γ treatment. Are these factors also involved in condensate formation?

Among all catalytically active PARPs, ADPr condensation requires only PARP14. Russo et al. (J Biol Chem, 2021) demonstrated that a genetic knockout of PARP9 affects the formation of ADPr condensates, despite PARP9 being catalytically inactive as an ADP-ribosyltransferase. Ribeiro et al. (EMBO, 2024) further confirmed the involvement of PARP9 through siRNA knockdown but also showed that condensate formation does not require PARP12. In contrast, Kar et al. (EMBO, 2024) reported that siRNA knockdown of PARP9 is not required for condensate formation in the same A549 cell line. Consistent with the latter findings, our data show no reduction in ADPr condensate numbers upon knockdown of either PARP9 or PARP12 in A549 cells, confirming that PARP14 is the major catalytically active ADP-ribosyltransferase that drives this condensation phenomenon (Fig. EV1L-O).

3. Figure 4D appears to be immunoprecipitation (IP) under non-denaturing conditions. If so, it is not possible to distinguish whether the MAR signal is derived from p62 or from the p62 interacting proteins (the associated ubiquitinated substrates). IP experiments should be performed under denaturing conditions.

As suggested by the reviewer, we have now conducted IPs under denaturing conditions and confirmed that p62 is MARylated upon IFN γ treatment in wild-type cells, but not in PARP14 knockout cells, indicating that p62 MARylation depends on PARP14 (Fig. EV4B). Additionally, we would like to note that following our submission, Kubon et al. reported in a bioRxiv preprint that p62 is ADP-ribosylated in a PARP14-dependent manner upon treatment with type I interferon, IFN β . This finding is consistent with our study involving type II interferon, IFN γ .

4. In Figure 5B, which band is HO1, the upper or lower?

Both bands are HO1, as shown by Biswas et al., J Biol Chem 2014. One band appears at 28 kDa and the other at 32 kDa. The 32-kDa isoform is predominantly constitutive in the cytoplasm, whereas the 28-kDa HO-1 is predominant and primarily localized to the nucleus.

5. There is no image for ubiquitin in S5D.

Our original statement, “However, when inhibited with the mTOR inhibitor Torin-1, autophagy is induced, leading to increased autophagosome formation marked by LC3B on the membranes, which facilitates the recruitment of p62 and ubiquitinated proteins (Fig. S5D),” contained a misplaced figure citation. The correct statement should be:

“However, when inhibited with the mTOR inhibitor Torin 1, autophagy is induced, leading to increased autophagosome formation marked by LC3B on the membranes (Fig. EV5E-F), which facilitates the recruitment of p62 and ubiquitinated proteins.”

Our intention was to show that there are conditions, such as Torin 1 treatment, where p62 and LC3B colocalize.

Right panel in Figure 4F shows only IF γ + RBN, which should show all data sets in the same panel.

Given the complexity of the three conditions with extensive data points and error bars on the FRAP experiments, we aim to present the data clearly. Instead of merging the panel into one figure, we initially provided a summary table in Fig. 4F. However, in response to the reviewer's comments, we have now provided the composite image that includes all data sets in the same panel as Fig. 3H-I.

	Recovery	Statistics	T _{1/2} (s)	Statistics
-IFN γ	0.24 ± 0.02]***] ns	763 ± 150]ns] ns
+IFN γ	0.48 ± 0.06		768 ± 192	
+IFN γ +RBN	0.30 ± 0.04		536 ± 225	

Reviewer #1 (Significance (Required)):

Liquid droplets, which have continuously being identified in cells, are a hot topic in cell biology. Droplet formation, structure, molecular dynamics, and degradation, as well as their abnormalities and disease development due to genetic mutation and stress, are of wide-ranging interest from basic to pathological aspects. Therefore, this research has the potential to attract interest from a wide range of fields.

General assessment

Overall, the data are clear and the phenomenon is of interest. However, the molecular mechanism and biological significance of the condensate formation is unknown; It is unclear why proteasome activity is required for the formation of PARP14-mediated ADP ribosylation. It is also unclear what the

consequences are for the cell if the ADPr-positive condensates are not formed. The authors should address these general and important issues and provide the data if not all.

We thank the reviewer for acknowledging that our condensate investigation is timely and important in cell biology and for recognizing that “*data are clear and the phenomenon is of interest*”. As mentioned in the *Discussion*, these condensates can be reversed by the SARS-CoV-2 macrodomain in lung A549 cells, whose activity to remove ADP-ribosylation is critical for viral replication and pathogenesis, indicating the biological significance of these condensates. In addition, similar IFN γ conditions can induce PARP14 expression in melanoma, where PARP14 inhibition resensitizes these cancers to immunotherapy. Given that these ADPr condensates are also observed in A375 melanoma cells beyond lung cells (Fig. EV3H), this provides additional context to investigate their biological significance in the future.

We would like to note that we have already made significant advances by (1) revealing the identity of these condensates as related to p62 bodies (Fig. 3-5), (2) defining the responsible ADP-ribosyltransferase as PARP14 (Fig. 1-2), and (3) determining the requirements for condensation through ubiquitin-proteasome system (Fig. 6-7). The proposed exploration of the functional consequences and significance is beyond the scope of this manuscript. However, we have now further defined the mechanistic involvement of which step of the ubiquitin-proteasome system.

The ubiquitin-proteasome system involves ubiquitination by E1, E2, and E3 enzymes, followed by proteasomal degradation. Disruption of ubiquitination, either through the E1 inhibitor TAK-243 or genetic depletion of the E1 enzyme UBA1, inhibited ADPr condensate formation (Fig. 7A-B). Both E3 ligases tested, DTX3L and TRIM25, showed increased association with p62; however, only DTX3L colocalized with p62 and ADPr, and its knockdown inhibited ADPr condensate formation (Fig. 5K, 7C-D, and EV7C-D). Inhibition of the final step of the ubiquitin-proteasome system using three distinct proteasome inhibitors—MG132, epoxomicin, and bortezomib—also blocked ADPr condensation (Fig. 6D and EV6E). Conversely, inhibiting deubiquitinating enzymes with PR619 increased ubiquitinated protein levels and increased ADPr condensate formation, underscoring the critical role of ubiquitination (Fig. 7E and EV7E). These findings collectively demonstrate that an active ubiquitin-proteasome system is critical for PARP14 and ADPr condensation within p62 bodies.

Reviewer #2 (Evidence, reproducibility and clarity (Required)):

This manuscript investigates the formation of a novel cellular structure or condensate, similar to p62 bodies, that includes PARP14 and p62. The interferon-induced PARP14-mediated ADP-ribosylation of p62 in these condensates depends on an active ubiquitin-proteasome system. These condensates are characterized by the presence of PARP14 and ADPr and include some, but not all, components of the p62 bodies. Furthermore, their formation depends on both ubiquitin activation and proteasome activity, but it is unaffected by autophagy inhibition, unlike conventional p62 bodies.

The Introduction provides a well-delineated context of condensates, highlighting the importance of post-translational modifications in responding to environmental changes.

Although the manuscript is well-organized with apparent logical development, there are weaknesses that diminish the impact of the reported data. A more accurate review of the structures, both from a morphological and quantitative perspective, would strengthen the conclusions and the overall impact of this work. Additionally, while the authors have analyzed the contribution of PARP14 to condensate formation, the biological significance of these structures remains unclear. For instance, performing MS

(mass spectrometry) analysis on the described structures could help identify their composition and functions.

The methodologies used in this study are standard for molecular and cellular biology research, including immunofluorescence assays, transient transfections, immunoprecipitations, and fluorescence recovery after photobleaching (FRAP) assays. These methods are described in detail and can be reproduced.

Below, please find a list of comments and suggestions to enhance the robustness of the data:

We thank the reviewer for acknowledging the logical progression of the manuscript and the detailed, reproducible methods. As suggested by the reviewer, we have now performed super-resolution microscopy experiments to examine morphology, improved data quantification, and conducted proteomics experiments to identify how the p62 interactome changes upon IFN γ treatment.

Major Points

1. The IF analyses are central to the conclusions reported and are employed for each of the inhibitors or other tools used to investigate the formation of these condensates. The quality of the IF images needs to be improved; the shape and contacts of the condensates should be analyzed using either super-resolution or EM microscopy, or preferably both. The lack of morphometry and quantification from cell populations needs to be addressed for all experiments. These analyses are needed to support the claim that the condensates presented in this study are indeed novel structures, rather than being transient aggregates of a different nature.

We believe the reviewer may be referring to the size of the images rather than their quality, as they are of high resolution when zoomed in. However, we agree that larger images would enhance clarity. To address this, we have included insets that display a quarter of the cell, allowing for a clearer visualization of the condensate shape and contacts.

We also performed Airyscan imaging to provide super-resolution images for a better understanding of the morphology of these condensates (e.g., circularity, ellipticity, solidity; Fig. 3G and EV3F-G). Additionally, we rendered the data using 3D modeling and provided a video for examining the colocalization between ADPr, PARP14, and p62 (Fig. 3F and Movie EV1).

For quantification, we would like to point out that nearly all our experiments were analyzed from at least 3 biological replicates, each containing 20-50 cells (depending on the magnification of 20x or 40x). In response to the reviewer's comment, we have now provided quantification on a per-cell basis.

Our data indicate that these are novel ADPr-containing condensates that colocalize with PARP14, p62, NBR1, and ubiquitin, but not LC3B (Fig. 1D, 3A, 5D-G). These structures are inducible by IFN γ treatment and can be inhibited by even 1 hour of PARP14 inhibition (Fig. 1A, 2D, and EV1B). They are dependent on PARP14 induction and its ADP-ribosyltransferase activity (Fig. 1F, 2B, 2D). These structures are not protein aggregates, as evidenced by their lack of staining with ProteoStat Dye (Fig. EV6C), which stains for unfolded proteins.

2. The claim that PARP14 is essential for the formation of condensates requires support by the analyses indicated above. Minor points regarding Fig. 1 are indicated below. I suggest performing KD of PARP9 and/or PARP12 (whose expression is increased upon IFN treatment) and checking ADPr condensates to validate the central role of PARP14.

ADPr condensation requires PARP14, as demonstrated by multiple genetic depletion techniques (siRNA/shRNA/CRISPR; Fig. 1F, EV11-K) and chemical inhibitors (catalytic and PROTAC; Fig. 1G-H, 2B, and 2D). Russo et al. (J Biol Chem, 2021) demonstrated that a genetic knockout of PARP9 affects the formation of ADPr condensates, despite PARP9 being catalytically inactive as an ADP-ribosyltransferase. Ribeiro et al. (EMBO, 2024) further confirmed the involvement of PARP9 through siRNA knockdown but also showed that condensate formation does not require PARP12. In contrast, Kar et al. (EMBO, 2024)

reported that siRNA knockdown of PARP9 is not required for condensate formation in the same A549 cell line. Consistent with the latter findings, our data show no reduction in ADPr condensate numbers upon knockdown of either PARP9 or PARP12 in A549 cells (Fig. EV1L-O), confirming that PARP14 is the major catalytically active ADP-ribosyltransferase that drives this condensation phenomenon.

3. According to the text, "PARP14 was pulled down by ADP-ribose binding Af1521 macrodomain following IFN γ treatment (Fig. 2H)", but the legend to the figure says otherwise. A Pan-ADPr binding reagent (MABE1016) is reported in the figure. Although the conclusion is similar for the results obtained with these two tools (but they must be described and reported properly), it is still insufficient to claim that PARP14 is ADP-ribosylated. This point should be at least discussed.

We apologize for the confusion. The Pan-ADPr binding reagent (MABE1016) is a His-tagged recombinant Af1521 macrodomain that binds to ADP-ribosylated proteins (Gibson et al., Biochemistry 2017). Therefore, we used Ni-NTA resin to pull down His-tagged AF1521 for the subsequent analysis of PARP14. We have revised the figure legend for Fig. 2H to clarify this.

"(H) A549 WT and PARP14 KO cells were treated with or without IFN γ for 24 h and subjected to immunoprecipitation using the pan-ADPr binding reagent (MABE1016) overnight. This reagent is a His-tagged recombinant Af1521 macrodomain that binds to ADP-ribosylated proteins. Ni-NTA resin was used to pull down His-tagged MABE1016, followed by western blot analyses."

Consistent with our studies, Kar et al. and Ribeiro et al. (EMBO J. 2024) also recently reported that PARP14 is ADP-ribosylated upon IFN γ treatment in A549 cells. We have now included these references in our *Discussion*. Additionally, Higashi et al. (J. Proteome Res. 2019) reported PARP14 ADP-ribosylation in IFN γ -treated macrophage cells.

4. I have difficulties analyzing the colocalization with the different organelles, even enlarging the images as much as possible. In most cases, only one condensate per image is shown. Continuities with the nuclear envelope appear in some cases: has this been investigated?

We have provided images of at least two cells containing multiple cytoplasmic condensates in Fig. EV3A. We believe part of the confusion arises from the staining pattern of ADPr in the nucleus, which

colocalizes with splicing speckles. However, this nuclear staining was not altered by IFN γ treatment. Therefore, we have focused on the cytoplasmic condensates in this study.

In addition, we have included insets that display a quarter of the cell, allowing for a clearer visualization of the condensate colocalization with other structure. For each colocalization study, we have analyzed 52-198 condensates from at least 3 biological replicates, each containing 20-30 cells. To further strengthen our claim for colocalization analyses, we have now provided quantification on a per-cell basis.

Minor Points

1. Fig. 1A: The DAPI images at 3 and 6 hours are reversed. Additionally, for Fig. S1a and Fig. 1A, please include quantifications.

We apologize for the oversight, which we have corrected. All quantifications have now been included.

2. Fig. 1B: Check PARP14 levels (and other IFN-PARPs) under the same experimental conditions.

As suggested, we have assessed the PARP14 protein levels as well as the levels of other IFN γ -induced PARPs, including PARP9, PARP12, and PARP14, using RT-qPCR (Fig. EV1E)

3. Fig. 1H: Explain why PARP14 IF staining is still visible upon RBN012811 treatment, while it is completely lost in WB analysis or upon PARP14 siRNA treatment (Fig. 1G). In addition, please include IF quantifications.

To clarify, for Fig. 1H (now Fig. 1G), the images provided correspond to those from IFN γ -treated cells, while the western blot data (Fig. 1H) represent both untreated and IFN γ -treated conditions. Previously, when presenting the data, our aim was to highlight the condensate pattern rather than the relative intensity differences, and we used the autocontrast image function across different conditions. We have now ensured that images across all conditions are presented with the same exposure and consistent image parameters. Under these conditions, we note that RBN012811 treatment shows a dose-dependent signal over increasing treatment durations, consistent between IF images and western blot results. Notably, a small amount of PARP14 remains detectable at the 1-hour timepoint in both western blots and IF images.

4. Fig. 2C: Please include quantifications. The quantifications are now included

5. Fig. 2E: The RBN treatment time is not indicated. Please include this information in the figure legend.

We apologize for the oversight, which we have now clarified as below:

“(E) PARP14 protein levels were measured in cells pretreated with RBN (10 μ M) for 1 h with or without 24-h IFN γ treatment. RBN was maintained throughout the experiment.”

6. Fig. 2G: I am not convinced about the PARP14 staining. IF images do not show an increase in PARP14 levels, while WB analysis shows a strong increase in PARP14 protein levels (see Fig. 2E). Moreover, the RBN treatment time was not indicated; please include it in the figure legend. Does RBN alone affect PARP14 localization? The reported picture shows only 2 cells, each with a different subcellular localization of PARP14. As previously suggested, quantifications are required.

We appreciate the reviewer’s detailed observations regarding the PARP14 staining in Fig. 2G. Our initial intention was to demonstrate the pattern of PARP14 distribution rather than the relative intensity differences. To achieve this, we applied the autocontrast image function uniformly across conditions. However, we recognize that this may have inadvertently led to inconsistencies in visual interpretation. To address this concern, we have now reprocessed the images to ensure consistent exposure and image contrast parameters across all conditions. This adjustment confirms that the observed changes in PARP14 are attributable to increased expression levels following RBN treatment, not changes in subcellular localization.

Additionally, we have quantified the condensates and observed that PARP14 no longer colocalizes with ADPr upon RBN treatment, further supporting our findings. These quantifications have been included in the revised manuscript.

Regarding the RBN treatment time, we apologize for the oversight. We have updated the figure legend to include this critical detail, as follows:

“(G) Colocalization of ADPr and PARP14 was assessed in cells pretreated with either DMSO control or RBN (10 μ M) for 1 h, with or without 24-h IFN γ treatment. RBN was maintained throughout the experiment, $n = 45-88$ condensates.”

7. Fig. 3B: Pearson's correlation coefficient (PCC) is reported for $n=3$. The images show one condensate per cell. Under these conditions, the number of cells analyzed should be at least 100 for each experiment. Additionally, the PCC between PARP14 and p62 at steady state is shown to be 60% (which is quite high). However, the IF pictures do not support this quantification. Can the authors provide higher-resolution pictures? Does PARP14 always co-localize with p62? Lines 207-208 state: "these findings

suggest that PARP14 is localized to p62 bodies upon IFN γ treatment when ADP-ribosylation occurs." According to the PCC value, the two proteins co-localize even in the absence of IFN. Can the authors clarify this aspect?

We apologize for the inaccurate description. The data should be n = 3 biological replicates, representing different fields (as indicated by the number of dots in the original graph panels in Fig. 3B), each containing 20-30 cells. The Pearson's correlation coefficient was calculated across the cells, instead of focusing on the condensates. We have added the revised our analysis at condensate level (Fig. 3A-B and EV3C).

We have also now provided zoom-in insets and quantification to indicate the level of PARP14 colocalization with p62.

For PARP14, we did not see a significant number of PARP14 condensates in control cells; PARP14 condensates were seen only after IFN γ treatment. A fraction of PARP14 condensates did not colocalize with p62, as indicated by quantification analyses.

8. Fig. S3B: Please include quantifications.

We have added the quantifications for S3B (now presented as Fig. EV3H)

9. Fig. S3C: How was the condensate size quantified? It would be useful to show a quantification mask.

We apologize for the omission in the Method Section. The condensate size quantification was performed with ImageJ. The nuclei were first identified with DAPI staining and masked out from the ADPr channel. The image was then thresholded with the “Maximum Entropy” method from ImageJ, and the “Analyze Particles” function was used to identify condensates with size larger than 1 pixel. Representative quantification masks are now included below for reviewer’s evaluation.

Control

IFN γ

10. Fig. 3D: Does p62 KD affect PARP14 localization? The reported picture shows only 2 cells, each with different staining of PARP14.

p62 KD reduced the number of PARP14 condensates but did not apparently change their localization. We have now provided a representative image with more cells to illustrate this effect more clearly, with the effect on the number of condensate areas quantified at population level (now Fig. 3K).

11. Fig. 4A: Please quantify the PARP14 co-IP signal with p62, normalized to PARP14 total levels. In the reported WB, it is difficult to see the interaction between PARP14 and p62 in untreated conditions. Please provide clearer WB.

As suggested, we quantified the PARP14 co-IP signal by normalizing it with PARP14 input levels. Additionally, we have provided a clearer WB in the revised manuscript.

Additionally, I would expect an increased interaction between PARP14 and p62 upon IFN treatment due to PARP14 recruitment to p62 condensates, not just because of increased PARP14 levels. Since the authors show that PARP14 is not recruited to ADPr condensates upon RBN treatment (Fig. 2G), why is the interaction between p62 and PARP14 so high under RBN treatment?

RBN treatment inhibits PARP14 catalytic activity but simultaneously increases PARP14 levels, as first described by Schenkel et al., Cell Chem Biol 2021. Our western blot data indicated that the interaction between PARP14 and p62 is independent of this activity and instead depends on PARP14 protein levels. However, the formation of ADPr/PARP14-containing condensates required catalytically active PARP14. Based on these data, we conclude that the colocalization of p62 and PARP14 depends on the catalytic activity of PARP14, which is reflected by its ADP-ribosylation.

12. Fig. 4C: Please quantify WB signals of ADP-ribosylated p62 for the different conditions analyzed. ADP-ribosylation of p62 is still present in cells lacking PARP14. Are there other enzymes that can modify p62? Moreover, the authors state: "We observed an increased MARYlation of p62 upon IFN γ treatment" (line 230); is this dependent on the increase in PARP14 levels or the translocation of PARP14 to ADPr condensates? Quantifications should help clarify this aspect.

As suggested, WB signals have been quantified (Fig. EV4A). We agree with the reviewer's observation regarding the presence of ADP-ribosylated p62 in PARP14 KO cells. The basal levels of ADP-ribosylated p62 may be due to other PARP enzymes. However, PARP14 is critical for the increase in ADP-ribosylation under IFN γ treatment, as the increase was not observed in PARP14 KO cells. Given that PARP14 inhibitors increase PARP14 levels, we interpret that the increase in p62 MARYlation requires an increase in *active* PARP14 levels, not just its total level. Since PARP14 activity is crucial for the localization of PARP14 to p62 condensates and its enrichment of ADPr signals, it is possible, as suggested by the reviewer, that the increase in MARYlation of p62 is dependent on the translocation of

PARP14 to the structure. However, the field currently lacks the tools to disrupt p62 bodies without knocking down p62 to definitively test whether colocalization is required for the MARYlation increase.

13. Fig. 4G: Quantifications are required.

Quantification has now been included (now Fig. 4H).

Reviewer #2 (Significance (Required)):

The role of the PARP family in cellular processes is a very active and rapidly growing field. New information about the organization of PARPs in the nucleus, cytosol, or different types of bodies/structures is certainly relevant to the field.

However, the present study is too preliminary at the moment to be considered highly relevant. Both the data analysis and conclusions need to be carefully reviewed. After major revisions, the manuscript might be of general interest if well contextualized within the fields of post-translational modification and protein degradation processes. It would remain in any case interesting for the field of ADP ribosylation.

We thank the reviewer for recognizing the significance of our work in the rapidly evolving field of PARP biology. We apologize for the lack of clarity that we indeed quantified over 100 cells across at least 3 biological replicates for the data reported. To further address the concerns raised, we have now provided additional cell-based quantification to strengthen our claims. Furthermore, we have enhanced the contextualization of our findings within the broader frameworks of post-translational modification and protein degradation processes in the *Discussion* sections:

“The requirement for an active ubiquitin-proteasome system led us to explore existing data on the crosstalk between ubiquitination and ADP-ribosylation. Recent data show that PARP9 and DTX3L bind to PARP14 and regulate its stability (Saleh et al, 2024). This stabilization, critical for increasing PARP14 levels, is independent of the proteasome (Saleh et al, 2024), implying it is unrelated to the IFN-induced ADPr condensate regulation.

Instead, the IFN-induced ADP-ribosylation depends on the ubiquitination of proteins that are destined for proteasome degradation, consistent with the enrichment of K48-linked polyubiquitin chains in these ADPr-containing p62 bodies. A potential enzyme responsible for this modification is the E3 ligase DTX3L, given its physical association and co-localization with PARP14 (Saleh et al, 2024). Consistent with this premise, genetic depletion of DTX3L results in the loss of IFN-induced ADPr (Russo et al, 2021). Our proteasome inhibitor data now further suggest that not only the physical presence but also the enzymatic activity of DTX3L or other E3 ligases is critical, particularly their ability to ubiquitinate proteins with K48 linkages, which targets these poly-ubiquitinated proteins for degradation (Dikic & Schulman, 2023). Notably, inhibiting ubiquitination results in the loss of ADPr condensates, whereas increasing ubiquitination promotes ADPr condensate formation, suggesting that ADPr may play a downstream role in ubiquitin signaling within the proteasomal degradation pathway. Further investigation is warranted to identify these ubiquitinated protein substrates for proteasome degradation and to clarify ADPr's role in protein degradation during IFN γ treatment.”

Reviewer #3 (Evidence, reproducibility and clarity (Required)):

In this manuscript, titled "Interferon-Induced PARP14-Mediated ADP-Ribosylation in p62 Bodies Requires an Active Ubiquitin-Proteasome System", Raja et al. perform fluorescence microscopy assays and molecular analyses on cultured cells *ex vivo* to further our understanding of ADP-ribose accumulations that form in the cytoplasm in response to IF γ stimulation. Guided by the operon-like linkage and co-expression patterns of PARP14, PARP9, and DTX3L and recent reports describing a SARS-CoV-2-dissolved cytoplasmic body induced by interferon-induced that is rich in ADP-ribose and the PARP9/DTX3L heterodimer form following IF γ stimulation, the authors provide clarity in this manuscript regarding two knowledge gaps - (i) what catalyzes mono(ADP-ribosyl)ation within these structures (as

PARP9 lacks ADP-ribosyltransferase activity) and (ii) how these foci/condensates relate to similarly composed autophagy-associate "p62 bodies" that have been previously described. Using a combination of genetic depletion and inhibitor-based approaches, the authors show that these ADPr-condensates rely on the catalytic activity of PARP14. The authors also show that while these ADPr-condensates share componentry with "p62 bodies" like ubiquitin and p62 itself, these foci are distinct accumulations, as they lack both LC3B and sensitivity to autophagy inhibitors and require an active ubiquitin-mediated proteosomal degradation system.

While this report represents a more incremental advance in our understanding of these cell signaling structures, especially considering a pair of very recently published and similar reports (Kar et al., *EMBO J* [2024] and Chaves Ribiero et al *EMBO J* [2024]), the work here is well-written and reasoned and complements these works with some novelty and distinction to that reported literature. The experiments are definitive and of high quality and the authors interpretations/conclusions are largely well-supported by the results. Thus, it is my opinion that this work is appropriate for publication with predominantly minor revisions (outlined below) and a few more substantive experimental additions.

We thank the reviewer for recognizing the high quality of our work and for acknowledging that our interpretations are well-supported by the results. We appreciate that the reviewer deemed our work ready for publication with minor revisions. We believe the reviewer's perception of our work as incremental arises because two related studies were published in June, after we submitted our work to Review Commons in May. According to the Scooping Protection Policy, our work should still be considered novel. The publication of these studies in *EMBO J*, which discovered the role of PARP14 in ADPr-containing condensate formation, highlights the significance of our research. However, we surpass their findings by demonstrating the critical role of PARP14 using multiple genetic manipulation techniques (siRNA, shRNA, and CRISPR-Cas9) and chemical inhibitors (catalytic and PROTAC), indicating the rigor of our studies. Moreover, we not only investigate PARP14 but also define the identity of these condensates related to p62 bodies and establish the requirement of an active ubiquitin-proteasome system for their formation.

Major Comment:

A major claim and novelty reported here is that the ADPr condensates are distinct from "p62 bodies". The evidence to support this rely largely on differences in their sensitivity to pharmacological treatments as well as somewhat subtle differences in FRAP recovery in p62 condensates after IFN γ treatment. But, this claim would be better supported with more comprehensive mapping of differences in the componentry or functional outcomes of these condensates. The authors might consider:

-Mass spectrometry against p62 (a common component) in standard "p62 bodies" and ADPr Condensates, followed by IF to confirm significantly different composition in, what is argued here, these distinct structures.

We appreciate the reviewer's insightful suggestions to strengthen the evidence distinguishing ADPr condensates from "p62 bodies" by exploring differences in their composition and functional outcomes.

In our initial manuscript, we characterized ADPr condensates through immunofluorescence for established p62 body components (Fig. 3A-B, 5E-I), condensate size quantification (Fig. EV3I), and FRAP analysis of p62 mobility (Fig. 3H-I), all of which revealed significant compositional and dynamic changes upon IFN γ treatment.

In response to the reviewer's suggestions, we conducted additional proteomic analyses of the p62 interactome under untreated and IFN γ -treated conditions to identify shifts in condensate components.

Proteomics revealed interactions of p62 with PARP14, DTX3L, NBR1, and TAX1BP1 (Fig. 5J-K). While imaging data for PARP14 and NBR1 were included in the original manuscript, we expanded our analysis to assess colocalization of p62 with TAX1BP1 and the E3 ligases DTX3L and TRIM25.

Our results show that TAX1BP1, similar to NBR1, consistently colocalizes with p62, confirming its role as a core component of canonical p62 bodies (Fig. 5H and EV5L). In contrast, DTX3L and TRIM25 displayed increased association with p62 upon IFN γ treatment (Fig. 5K). Notably, DTX3L formed condensates that colocalized with p62 and ADPr, whereas TRIM25 did not (Fig. 7C and EV7C). Furthermore, DTX3L knockdown blocked ADPr condensate formation (Fig. 7D).

Additionally, proteomics identified ~20 other proteins that associate with p62 more strongly upon IFN γ treatment (\log_2 fold change >1.5, p-value <0.001; Fig. 5K). While these proteins may be additional components of ADPr condensates, further investigation is warranted.

-Fine mapping of concentration dependence of components that give rise to these distinct condensates as has been demonstrated in papers like Riback et al Nature 2020 and others.

In response to the reviewer's suggestion, we have explored the concentration dependence in our system. However, precise quantitative assessments proved challenging due to the nature of our ADPr staining protocol, which relies on signal amplification through primary and secondary antibodies. This amplification is prone to introduce nonlinearity in signal intensity, making it less reliable for quantitative comparisons.

In contrast, methodologies such as those employed by Riback et al. (Nature, 2020) utilize GFP-tagged components, which provide an inherently linear relationship between protein copy number and signal intensity. These methods are more suitable for fine mapping of concentration dependence but are not directly applicable to our experimental setup, which focuses on the critical post-translational modification staining of ADPr.

-Methodology of the author's choosing to decipher functional outcomes from these condensates followed by demonstration that components unique to ADPr condensates are dispensable for functioning "p62 bodies" and, vice versa, components unique to "p62 bodies" are dispensable for ADPr Condensate function.

While a full analysis of functional outcomes is beyond the scope of this manuscript, we further examined the composition of these condensates by assessing additional autophagy receptors and autophagosome components. We quantified the co-localization of ADPr with p62, as well as ADPr with LC3B, using LC3B as a proxy for autophagy-related canonical p62 bodies. Upon IFN γ treatment, ~20% of p62 condensates were ADPr-positive (Fig. EV3D-E). For co-localization between LC3B and ADPr (Fig.5D), we observed no significant changes upon IFN γ treatment, consistent with our hypothesis that these ADPr condensates are devoid of LC3B (Fig. EV5G).

Our findings further showed colocalization of p62 with the autophagy receptor NBR1 and TAX1BP1, but not with the autophagy initiation component FIP200 (Fig. 5G-I and EV5L-N), indicating a selective association with specific autophagy receptors while lacking recruitment of autophagosome machinery. Additionally, genetic depletion of autophagy factors ATG5 and Beclin 1 did not alter ADPr condensate formation (Fig. 6B-C), consistent with our autophagy inhibitor data (Fig. 6A), supporting their independence from canonical autophagy pathways.

We hope that these additional experiments and insights will substantiate the distinct nature of ADPr condensates in comparison to p62 bodies associated with autophagy.

Minor Comments:

Overall, the representative microscopy images are far too small. For the benefit of future readers, please consider enlarging these images.

More of the quantitation of microscopy images, with accompanying statistics, that are found in abundance in the supplemental material should find their way into the main figures of the manuscript. This will give room for larger and more reader-friendly representative microscopy images in the main figures/text as discussed briefly above.

We appreciate the reviewer’s suggestions and rightly pointed out that our quantification and statistics were in supplementary materials. Given that we have provided over 800 image panels, we have now restructured the manuscript so that the cell biology information is more readily available. We have moved many of our quantification data and statistics previously in the supplementary materials to the main figures. Additionally, we have included insets that display a quarter of the cell, allowing for a clearer visualization of the condensate, as suggested.

Can the authors test whether or not the condensates are purely driven by mono(ADP-ribosylation)? Or does poly(ADP-ribose) co-occupy these condensates and play a substantive role?

We have tested for the presence of poly(ADP-ribose) in the condensates and found it is not detectable with the antibodies against poly(ADP-ribose) (Fig. EV2H). In addition, inhibitors against PAR-making ADP-ribosyltransferases (Olaparib for PARPs 1 and 2 and XAV939 for PARPs 5a and 5b) did not affect these condensate formation (Fig. 2A, EV2C, and EV2E), indicating that these condensates are predominantly, if not purely, driven by mono(ADP-ribosylation).

The manuscript would benefit from discussing very recent and related reports (Kar et al., EMBO J [2024] and Chaves Ribiero et al EMBO J [2024]), that I suspect were not available at the time of submission.

Yes, the reviewer is correct that our submission preceded the publication of these related reports. We have now added the following in our *Discussion*:

“Consistent with our findings, recent studies reported after our submission confirm that ADPr condensates depend on PARP14 and have identified PARP14 and p62 as ADP-ribosylated in a PARP14-dependent manner (Kar et al, 2024; Ribeiro et al; Kubon et al, 2024).”

IFNalpha and IFNbeta, which are used in Figure S1, do not appear as reagents in Table 1.

We apologize for the oversight. We have added this information on IFNalpha and IFNbeta to the Reagents and tools table.

On lines 113-114, it would seem more appropriate to describe the increase of PARP-14 as statistically significant and largest in magnitude. "most significant" would just mean lowest p-value, which I expect is different that the authors intend here.

We thank the reviewer's suggestion and have modified the text as follows:

“PARP9, PARP12, and PARP14 were significantly upregulated at 6 and 24 h post-treatment, with PARP14 showing the largest increase in mRNA expression levels (Fig. EV1C-D).”

In Figure S1, better care should be taken to crop and align the western blots.

We thank the reviewer for pointing this out, and we have now properly aligned and cropped western blots in Fig. EV1.

On line 154, it may be more appropriate to describe ITK as a "weaker" inhibitor of PARP14 relative to PARP11. It certainly is effective as an inhibitor (Figs. S2A and S2G) and its unclear how the authors (or anyone would) define what qualities make it "weak".

We thank the reviewer's suggestion and will modify the text as follows:

"...—specifically, *RBN012579* (hereafter *RBN*) and *ITK7* (a potent *PARP11* inhibitor with inhibitory effects on *PARP14* that are weaker than *RBN*) (Schenkel et al, 2021; Kirby et al, 2018)—..."

The multiple bands for PARP14 in Figure 3E should be addressed. Why does this differ from other blots from the same cells?

We believe that the multiple bands seen can be due to insufficient blocking and using a different lot of PARP14 antibodies. We have addressed this issue by performing a new experiment with proper blocking conditions and using the same lot of PARP14 antibody as other blots (now Fig. EV3L). It should be noted that that variation is also observed in the reagent website: <https://www.scbt.com/p/parp-14-antibody-c-1>

Reviewer #3 (Significance (Required)):

I expect the advances in this work will appeal more to specialists who are interested in ADP-ribosylation as a signaling molecule and to those engaged in biotechnological efforts to drug immunological responses.

The advances reported here are incremental. The ADPr condensates that form in response to IFNγ, the involvement of PARP9/DTX3L, and very recently the involvement of PARP14 and its MARYlation activity are all known. Less known is the notion that this condensate is distinct from other kinds of "bodies", which is a clear point of novelty, especially if buttressed by the authors as suggested in this review.

We agree with the reviewer that our work holds particular significance for specialists in ADP-ribosylation signaling and immunology, as well as those exploring biotechnological approaches to modulate immunological responses. We understand the perception of our findings as incremental may arise due

to the publication of two related studies in June, shortly after our submission in May. However, as noted, the Scooping Protection Policy in Review Commons ensures that our work remains novel for editorial consideration. More importantly, the publication of these *EMBO J* studies underscores the growing recognition of the critical role of PARP14 in ADPr condensate formation. In this context, our work advances the field in several key ways:

- (1) **Rigorous Validation of PARP14's Role:** We provide comprehensive evidence of PARP14's critical role in ADPr condensate formation through multiple approaches, including siRNA, shRNA, CRISPR-Cas9, and chemical inhibition using catalytic and PROTAC inhibitors.
- (2) **Identification of ADPr Condensates as Distinct from Canonical p62 Bodies:** We present clear evidence distinguishing ADPr condensates from other p62-containing structures, expanding our understanding of their unique identity and function.
- (3) **Demonstration of Ubiquitin-Proteasome System (UPS) Involvement:** We define the active ubiquitin-proteasome system as essential for the formation of these condensates.

We further detail the mechanistic role of the UPS (Fig. 5-7), showing that disruption of ubiquitination, either through the E1 inhibitor TAK-243 or genetic depletion of UBA1, inhibits ADPr condensate formation. Among the tested E3 ligases, both DTX3L and TRIM25 associate with p62, but only DTX3L colocalizes with p62 and ADPr. Importantly, DTX3L knockdown impairs ADPr condensate formation. We also show that proteasome inhibitors (MG132, epoxomicin, and bortezomib) block condensation, whereas inhibition of deubiquitinating enzymes (with PR619) increases ubiquitinated protein levels and enhances ADPr condensate formation. These findings emphasize the requirement of active ubiquitination and proteasome function for PARP14-driven ADPr condensate formation within p62 bodies.

Additionally, our new proteomics data, super-resolution imaging, and quantification provide a more comprehensive analysis of these condensates. With this expanded evidence, we believe our work offers significant insights into a new class of condensates, demonstrating their regulation, composition, and distinction from canonical p62 bodies. Importantly, we show their dependence on the ubiquitination-proteasome system rather than autophagy.

We hope the reviewer will agree that these findings substantiate the novelty and impact of our study.

Prof. Anthony K. L. Leung
 Johns Hopkins University
 Department of Biochemistry and Molecular Biology
 Bloomberg School of Public Health
 Baltimore, MD 21205

14th Feb 2025

Re: EMBOJ-2024-118598R
 IFN-Induced PARP14-Mediated ADP-Ribosylation in p62 Bodies Requires Ubiquitin-Proteasome System

Dear Anthony,

Thank you for your patience with the re-review of your revised manuscript. It has now been seen again by all three Review Commons referees, and I am pleased to say that they were broadly satisfied with your responses and revisions. Referee 1 still notes a few minor comments (see below), which I would invite you to incorporate during a final round of minor revision. At this stage, please also take care of a few remaining editorial issues:

- Please remove the duplicated manuscript title on the second page.
- Please carefully go through the reference list, in which many citations are currently still incomplete, lacking e.g. page numbers, publication year, etc. Also, please adjust the format for citation of preprints as specified in our author guidelines: The citation in the text should be: "(preprint: NAME1 et al, YEAR)"
 The citation in the reference list: "Author NAME1, Author NAME2, ... (YEAR) article title. bioRxiv doi: XXX"
- As we are switching from a free-text author contribution statement towards a more formal statement based on Contributor Role Taxonomy (CRediT) terms, please remove the present Author Contribution section and instead specify each author's contribution(s) directly in the Author Information page of our submission system during upload of the final manuscript. See <https://casrai.org/credit/> for more information.
- Please upload the source data according to a scheme that separates main and EV figure source data. There should be one archive for each of the main figures; and on single archive combining the source data for all EV figure.
- Finally, during routine pre-acceptance image checks, we noted that some images shown in Figure 6B are also shown in Figure EV1M. Please carefully check and clarify/explain. While I notice that both sub-panels show identical treatments and may be repeated solely for comparison, this would nevertheless need to be explicitly indicated in both respective figure legends.

I am therefore returning the manuscript to you for a final round of revision, to allow you to make these modifications and upload the revised files. Once we will have received them, we should hopefully be able to proceed with formal acceptance and production of the manuscript.

With kind regards,

Hartmut

9) To facilitate reproducibility and cross-laboratory adoption of methodologies, please structure the Materials & Methods section as outlined in our guide to authors, including a completed Reagents and Tools Table that can be downloaded from our author guidelines as well (<https://www.embopress.org/page/journal/14602075/authorguide#structuredmethods>).

10) Digital image enhancement is acceptable practice, as long as it accurately represents the original data and conforms to community standards. If a figure has been subjected to significant electronic manipulation, this must be clearly noted in the figure legend and/or the 'Materials and Methods' section. The editors reserve the right to request original versions of figures and the original images that were used to assemble the figure. Finally, we generally encourage uploading of numerical as well as gel/blot image source data; for details see: embopress.org/page/journal/14602075/authorguide#sourcedata

At EMBO Press, we ask authors to provide source data for the main manuscript figures. Our source data coordinator will contact you to discuss which figure panels we would need source data for and will also provide you with helpful tips on how to upload and organize the files.

In the interest of ensuring the conceptual advance provided by the work, we recommend submitting a revision within 3 months (15th May 2025). Please discuss the revision progress ahead of this time with the editor if you require more time to complete the revisions. Use the link below to submit your revision:

Link Not Available

Referee #1:

In their revised manuscript, Raja et al. have addressed my initial concerns by incorporating new experiments, providing strong evidence for ADP-ribosylation in p62 body formation with the addition of quantitative imaging data. While the authors acknowledge that the molecular mechanism by which the ubiquitin-proteasome system regulates PARP14-p62 condensates remains unclear, the presented data are highly reliable. Minor comments for further improvement are listed below.
Minor comments

1. In Figure 4 and EV4B, the authors show the MARylation band of p62. Molecular weight markers should be added.
2. In Figure 3i, the graphs and tables do not match. The authors should check the labels on the data.

Referee #2:

The manuscript by Raja et al. has been revised with the addition of new data, morphological analysis, improved statistics and a more detailed discussion. The claim is that the formation of a novel class of condensates depends on an active ubiquitin-proteasome system and involves the interferon-induced PARP14-mediated ADP-ribosylation of p62. The initial hypothesis, as presented in its present form, is better substantiated and reported.

As previously noted, the manuscript may be of general interest within the fields of post-translational modification and ADP-ribosylation.

Referee #3:

The questions from my prior review have all been sufficiently addressed by the authors and I recommend this manuscript for publication.

Point-by-point responses for Raja et al.

Reviewer Comments

Referee #1:

In their revised manuscript, Raja et al. have addressed my initial concerns by incorporating new experiments, providing strong evidence for ADP-ribosylation in p62 body formation with the addition of quantitative imaging data. While the authors acknowledge that the molecular mechanism by which the ubiquitin-proteasome system regulates PARP14-p62 condensates remains unclear, the presented data are highly reliable. Minor comments for further improvement are listed below.

Response: We sincerely appreciate the reviewer's time and effort in providing valuable feedback to improve our manuscript.

Minor comments

1. In Figure 4 and EV4B, the authors show the MARYlation band of p62. Molecular weight markers should be added.

Response: We have provided molecular weight markers for all Western blots in the source data file. For reviewer's reference, we are providing them here again. Furthermore, for Figure 4D, the IP was performed against endogenous p62 while for EV4B IP was performed against GFP-p62.

Figure 4D

EV4B

2. In Figure 3i, the graphs and tables do not match. The authors should check the labels on the data.

Response: We appreciate the reviewer's careful attention to Figure 3i. The discrepancy likely arises because the recovery values in the table represent the normalized intensity change from the lowest to highest point of each curve, rather than the absolute values.

To clarify this, we have revised the graph by normalizing the lowest point to 0, ensuring that the recovery curves visually correspond to the recovery values in the table. Additionally, we have verified that the labels in both the table and graph are consistent. We thank the reviewer for their valuable feedback in improving clarity.

Referee #2:

The manuscript by Raja et al. has been revised with the addition of new data, morphological analysis, improved statistics and a more detailed discussion. The claim is that the formation of a novel class of condensates depends on an active ubiquitin-proteasome system and involves the interferon-induced PARP14-mediated ADP-ribosylation of p62. The initial hypothesis, as presented in its present form, is better substantiated and reported.

As previously noted, the manuscript may be of general interest within the fields of post-translational modification and ADP-ribosylation.

Response: We appreciate the reviewer's feedback and comments, which have significantly improved our manuscript. We also thank the reviewer for recognizing its general interest and noting that the data is now better substantiated and reported.

Referee #3:

The questions from my prior review have all been sufficiently addressed by the authors and I recommend this manuscript for publication.

Response: We sincerely thank the reviewer for their constructive feedback and for recommending our manuscript for publication.

Prof. Anthony K. L. Leung
Johns Hopkins University
Department of Biochemistry and Molecular Biology
Bloomberg School of Public Health
Baltimore, MD 21205

13th Mar 2025

Re: EMBOJ-2024-118598R1
Interferon-Induced PARP14-Mediated ADP-Ribosylation in p62 Bodies Requires the Ubiquitin-Proteasome System

Dear Anthony,

Thank you for submitting your final revised manuscript for our consideration. I am pleased to inform you that we have now accepted it for publication in The EMBO Journal.

With kind regards,

Hartmut

Rev_Com_number: RC-2024-02532

New_manu_number: EMBOJ-2024-118598R1

Corr_author: Leung

Title: Interferon-Induced PARP14-Mediated ADP-Ribosylation in p62 Bodies Requires the Ubiquitin-Proteasome System